# REM transcription factors and GDE1 shape the DNA methylation landscape through the recruitment of RNA polymerase IV transcription complexes

Zhongshou Wu[1,2,3], Yan Xue[2,4], Shuya Wang[2], Yuan-Hsin Shih[2,5], Zhenhui Zhong [2,6], Suhua Feng[2,7], Jonathan Draper [2], Allen Lu[2], Carsten A. Hoeke [2], Jihui Sha [8], Lu Li[8], James Wohlschlegel[8], Keqiang Wu [5] & Steven E. Jacobsen [2,3,7,8] ✉

In plants, the maintenance of DNA methylation is controlled by several self-reinforcing loops involving histone methylation and non-coding RNAs. However, how methylation is initially patterned at specific genomic loci is largely unknown. Here we describe four *Arabidopsis* REM transcription factors, VDD, VAL, REM12 and REM13, that recognize specific sequence regions and, together with the protein GENETICS DETERMINES EPIGENETICS1 (GDE1), recruit RNA polymerase IV transcription complexes. This targeted recruitment leads to the production of 24-nucleotide small interfering RNAs that guide DNA methylation to specific genomic sites in plant female reproductive tissues. In the absence of *GDE1*, polymerase IV transcription complexes are directed to loci bound by an alternative transcription factor, REM8, highlighting the role of REM transcription factors and GDE1 proteins as positional cues for epigenetic modulation. These findings establish a direct connection between sequence-specific transcription factors and the spatial regulation of siRNA production and DNA methylation, offering new insights into the genetic control of epigenetic patterning.

Cytosine DNA methylation and histone modifications are two deeply conserved epigenetic marks that together maintain genome stability and regulate gene expression in eukaryotes. These marks are critical for silencing transposable elements and controlling gene activity, thereby shaping development, physiology and genome integrity in a wide range of organisms[1–3]. In plants, DNA methylation occurs in three different sequence contexts: CG, CHG or CHH (where H is A, T or C) and is established primarily through the plant-specific RNA-directed DNA methylation (RdDM) pathway[4,5]. This process begins with RNA polymerase IV (Pol IV)-mediated transcription of 25–40-nucleotide

small interfering RNA (siRNA) precursors[6–8]. The recruitment of Pol IV is orchestrated by the CLASSY (CLSY) chromatin remodelling factors, CLSY1–4, which act in a tissue- and locus-specific manner to regulate 24-nt siRNA production[9–15].

In leaves, CLSY1 and CLSY2 dominate siRNA biogenesis[9], while in ovules, CLSY3 and CLSY4 are highly expressed and direct Pol IV complexes to generate siRNAs at specific loci called siren (small-interfering RNA in the endosperm) loci[11]. These siren-derived siRNAs promote DNA methylation in *cis* at their originating loci and also in *trans* at certain protein-coding genes, modifying gene expression and influencing

reproductive processes such as hybrid seed failure in *Capsella* and seed abortion in *Brassica rapa*[12,16,17]. A similar mechanism is observed in male germ lines, where the CLSY3 CLSY4-dependent hypermethylated transposable element (hyperTE) loci in tapetal nurse cells produce siRNAs that move into microspore mother cells, directing DNA methylation and gene regulation[10,12].

Repressive chromatin states are stabilized through self-reinforcing loops between DNA methylation and histone modifications, particularly H3K9 methylation[18,19]. For instance, the SU(VAR)3-9 HOMOLOG 4 (SUVH4), SUVH5 and SUVH6 methyltransferases bind methylated DNA and deposit H3K9 methylation in heterochromatin[20–24]. Conversely, CHROMOMETHYLTRANSFERASE 2 (CMT2) and CMT3 read H3K9 methylation to add CHH and CHG methylation to nearby DNA, reinforcing repressive chromatin[25–28]. In addition, the CLSY1 and CLSY2 remodellers facilitate the interaction between H3K9 methylation readers, such as SAWADEE DOMAIN HOMOLOG 1 (SHH1), and Pol IV, ensuring efficient siRNA production and DNA methylation[29–31]. The chromatin remodeller ZINC FINGER, MOUSE DOUBLE-MINUTE/SWITCHING COMPLEX B, PLUS-3 Protein (ZMP) supports siRNA biogenesis independently of SHH1 by confining Pol IV to chromatin regions devoid of H3K4me, preventing interference with actively expressed genes[32]. Together, these interconnected layers have supported a model in which epigenetic patterns are shaped predominantly by chromatin-based features, rather than by the underlying DNA sequence itself.

However, recent observations suggest that this dogma may be incomplete. At siren loci, which rely on CLSY3 and CLSY4, siRNA production and DNA methylation occur independently of H3K9 methylation[9,11]. Instead, these processes are associated with a conserved DNA motif, indicating that sequence-specific cues can shape the epigenetic landscape[11]. Here, we report that reproductive meristem (REM) transcription factors recognize defined DNA sequences and, with the assistance of GENETICS DETERMINES EPIGENETICS1 (GDE1), recruit the Pol IV complex to produce siRNAs and direct DNA methylation in *Arabidopsis* ovule tissue. Our findings establish a direct link between genetic elements and epigenetic regulation, expanding our understanding of how DNA methylation patterns are determined.

## Results

### *GDE1* encodes an uncharacterized protein at RdDM sites
To identify additional players in the RdDM pathway, we used cross-linked immunoprecipitation–mass spectrometry (IP–MS) with the RdDM-associated protein MORC7[33,34], and discovered a previously uncharacterized protein, AT1G77270, hereafter named GDE1. To determine the genomic localization of GDE1, we performed chromatin immunoprecipitation sequencing (ChIP-seq) with flower tissues from *pGDE1::GDE1-3FLAG* (*GDE1–3FLAG*). GDE1 largely co-localized with MORC7 across the genome (Extended Data Fig. 1a,b), as well as with key components of the RdDM pathway, including Pol IV and Pol V (Fig. 1a–c and Extended Data Fig. 1b). Genome-wide correlation analysis indicates that GDE1 is more closely associated with the Pol IV arm than the Pol V arm of the RdDM pathway (Extended Data Fig. 1b). These results suggest that GDE1 is involved in RdDM function, with a stronger connection to the Pol IV arm.

Motif enrichment analysis of GDE1 ChIP peaks with a fold enrichment greater than 5 revealed the presence of a highly conserved motif (hereafter named CLSY3 CLSY4 motif 1), which the RdDM component CLSY3 was previously shown to favour as well[11] (Fig. 1d). We found that CLSY4 ChIP signals exhibited strong enrichment at CLSY3 binding sites, with over 92% of CLSY3 peaks overlapping with CLSY4 peaks (Extended Data Fig. 1c,d). Furthermore, GDE1 was highly enriched at both CLSY3 and CLSY4 binding sites (Fig. 1e,f). Similar to the unique expression pattern of CLSY3 and CLSY4[11], GDE1 was highly expressed in flower tissues (Extended Data Fig. 1e) based on the ePlant database[35]. Taken together, these results suggest that GDE1 might collaborate with CLSY3 and CLSY4 (CLSY3/4).

### GDE1 directs CLSY3/4-Pol IV to modulate siRNA production
CLSY3/4 have been reported to be required for Pol IV activity and siRNA generation in ovules[9,11]. Our analysis identified 753 siRNA sites that were reduced in the *clsy3 clsy4* double mutant in ovule tissues (Fig. 2a). To determine whether GDE1 affect siRNA biogenesis, the levels of siRNA in *gde1-1* (SALKseq_10069.1) ovule tissue were profiled genome wide. Strikingly, 57% of the CLSY3 CLSY4-dependent siRNA loci showed reduced siRNAs in *gde1-1*, while 35% exhibited increased siRNAs in *gde1-1* (Fig. 2a–c and Extended Data Fig. 2a), suggestive of the overlapping and differential siRNA modulation function of GDE1 with CLSY3 and CLSY4. Within these CLSY3/4-dependent siRNA loci, we defined GDE1-dependent loci as group 1 sites, while loci showing increased siRNA in *gde1-1* were defined as group 2 sites and the rest of the loci were defined as group 3 sites (Fig. 2a).

To test whether the effect of siRNA production in *gde1-1* mutant relies on CLSY3/4 and Pol IV distribution, ChIP-seq of CLSY3/4 and Pol IV in Col-0 (wild type, WT) and *gde1-1* mutant was performed. CLSY3/4 and Pol IV displayed strong enrichment at group 1 in WT, but this enrichment was largely lost in the *gde1-1* background (Fig. 2d), mirroring the changes in siRNA abundance. Consistent with GDE1 playing a role in recruiting siRNA biogenesis to Group 1 sites, we found that GDE1 was strongly localized to group 1 sites (Fig. 2e), and GDE1–3FLAG fully complemented the *gde1-1* siRNA phenotype (Fig. 2b,c). By contrast, CLSY3/4 and Pol IV were barely enriched at Group 2 in WT, but more ChIP signals were observed in the *gde1-1* background (Fig. 2d and Extended Data Fig. 2a). These findings suggest that GDE1 plays a pivotal role in recruiting the CLSY3/4–Pol IV complex to generate siRNAs at a subset of CLSY3 CLSY4-dependent siRNA loci and that, in its absence, these complexes redistribute to other loci. Interestingly, the majority of group 2 sites were found in heterochromatic regions, while group 1 sites were located predominantly in euchromatic regions (Fig. 2f). This suggests that GDE1 primarily regulates siRNA production in euchromatic regions and competes with Pol IV recruitment to heterochromatic regions.

### GDE1 colocalizes with REM transcription factors
To understand how GDE1 influences siRNA production, we performed IP–MS with *GDE1–3FLAG* transgenic plants to uncover interacting proteins. Unexpectedly, none of the CLSY3/4–Pol IV components was identified in the IP–MS experiments (Fig. 3a and Supplementary Table 1). To further assess potential interactions using a more sensitive approach, we conducted co-immunoprecipitation (co-IP) assays using F2 transgenic plants expressing GDE1–3FLAG with CLSY3–9myc or Pol IV–9myc. GDE1 proteins were scarcely detectable in input samples, probably due to their restricted expression in flower tissues. However, in the immunoprecipitated samples, GDE1 successfully pulled down CLSY3 and Pol IV (Extended Data Fig. 2b), suggesting that GDE1 forms a complex with CLSY3 and Pol IV.

Interestingly, multiple members from the REM transcription factor family were pulled down by GDE1 in the IP–MS assay (Fig. 3a and Supplementary Table 1). To examine whether REM transcription factors co-localize with GDE1–CLSY3–Pol IV complexes, we performed ChIP-seq with 9myc-tagged transgenic lines. Two of these REM proteins, VDD and VAL, showed strong ChIP signals over the same set of CLSY3 CLSY4-dependent siRNA loci in group 1 (Fig. 3b–e and Extended Data Fig. 3a). In addition, a similar DNA motif with CLSY3 CLSY4 motif 1 was found in VDD ChIP-seq data (Extended Data Fig. 3b). Another factor, REM22, was enriched at a distinct subset of CLSY3 CLSY4-dependent siRNA loci within group 1, but also showed some colocalization with VDD and VAL (Fig. 3b–e and Extended Data Fig. 3a,c).

To further explore transcription factor–GDE1 complexes, IP–MS was performed by using *VDD–3FLAG* transgenic lines. VDD successfully pulled down GDE1, CLSY3, Pol IV subunits[36], RDR2 and RDM4, an IWR-type transcription factor known to interact with Pol IV[1] (Fig. 3f and Extended Data Fig. 2b), indicating that VDD forms a complex with GDE1

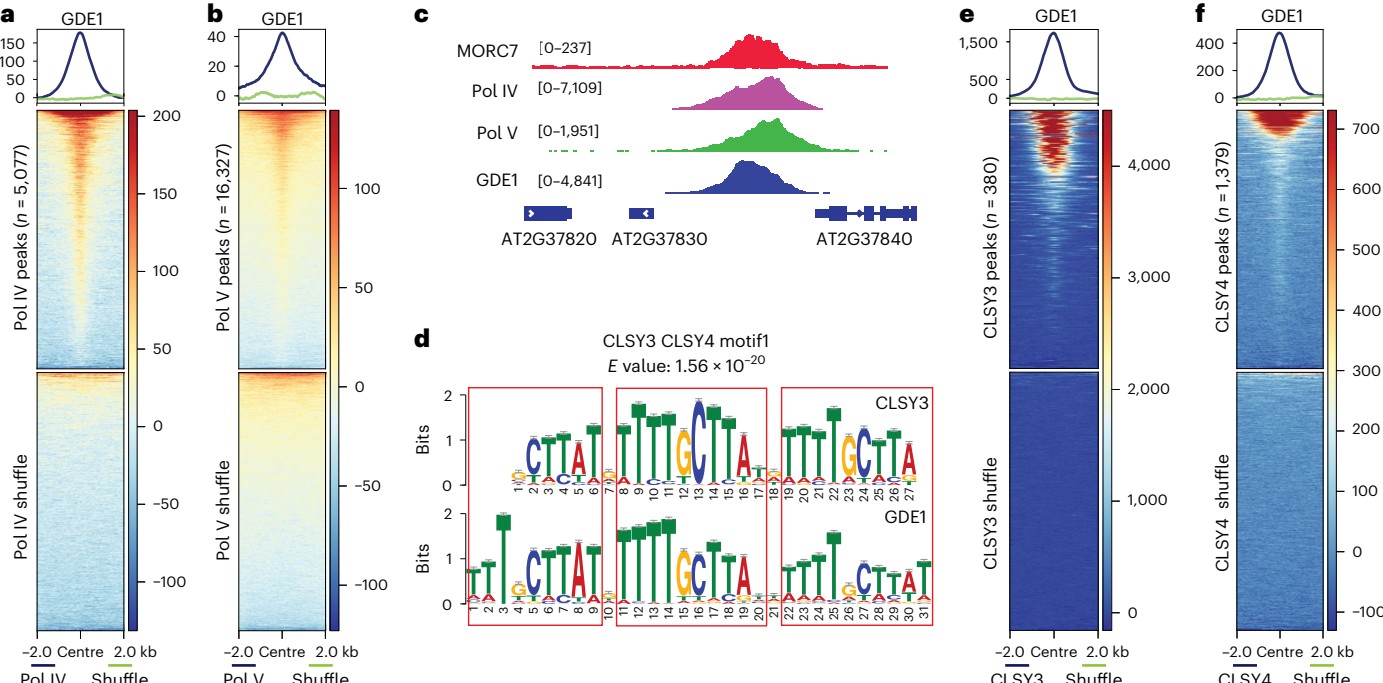

**Fig. 1 | GDE1 is a RdDM protein and co-localizes with Pol IV recruiter CLSY3/4.**
**a,b**, Metaplot and heatmap showing enrichment of GDE1 ChIP-seq signal over
Pol IV (*n* = 5,077) (**a**) and Pol V (*n* = 16,327) (**b**). **c**, A screenshot of MORC7, Pol IV,
Pol V and GDE1 ChIP-seq at a representative locus. Square brackets indicate the
range on a bar graph. **d**, TomTom analysis showing the similarities of CLSY3 and

GDE1 binding motifs. Three repeats are boxed in red. *E* value is the expected
number of false positives in the matches up to this point. **e,f**, Metaplot and
heatmap showing enrichment of GDE1 ChIP-seq signal over CLSY3 (*n* = 380) (**e**)
and CLSY4 (*n* = 1,368) (**f**).

and Pol IV complex components. In addition, we detected VAL, REM46
and additional members of the REM transcription factor family (Fig. 3f),
suggestive of the in vivo interactions among these transcription factors
family. Consistent with this observation, VAL was previously reported
to interact with itself and VDD by yeast two-hybrid and a bimolecular
fluorescence complementation assay[37]. In addition, most of these REM
transcription factors showed expression patterns similar to those of
CLSY3/4 and GDE1, exhibiting high expression in ovules in comparison
with other tissues (Extended Data Fig. 3d). Taken together, these results
suggest that multiple REM transcription factors may be involved in
the recognition of the CLSY3 CLSY4-dependent siRNA loci and their
association with GDE1 probably recruits CLSY3/4–Pol IV to generate
siRNAs for locus-specific DNA methylation.

To search for additional REM transcription factors that might
recognize the CLSY3 CLSY4-dependent siRNA loci, we performed
ChIP-seq using 9myc-tagged REM transcription factors identified
in the IP–MS data (Fig. 3a,f and Supplementary Tables 1 and 2) and/
or co-expressed with GDE1 by ATTED-II RNA coexpression analysis[38]
and/or highly expressed in ovules (Extended Data Fig. 3d). We found
that REM13 exhibited a localization and motif pattern similar to VDD
and VAL, demonstrating strong signals over the same set of CLSY3
CLSY4-dependent siRNA loci in group 1 (Fig. 3b–e and Extended Data
Fig. 3a,b). REM12 was identified from *VDD–3FLAG* IP–MS and highly
expressed in ovule; however, we failed to observe ChIP signal using
transgenic lines. Nevertheless, publicly available DNA affinity purifi-
cation sequencing (DAP-seq) data[39] showed that the binding sites of
REM12 highly overlapped with the same set of CLSY3 CLSY4-dependent
siRNA loci bound by VDD and VAL (Fig. 3g). In addition, a similar motif
enrichment was found in REM12 DAP-seq as the CLSY3 CLSY4 motif 1
(Extended Data Fig. 3b). For REM19, 80% of its ChIP-seq signals were
enriched at group 1 CLSY3 CLSY4-dependent siRNA loci, with a rela-
tively lower intensity, while 20% extended to group 2 regions (Fig. 3b–
e,h and Extended Data Fig. 3a). By contrast, 86% of REM8 ChIP-seq

peaks localized to group 2 of CLSY3 CLSY4-dependent siRNA loci
instead, showing a preference for *gde1-1* upregulated siRNA regions
(Fig. 3b–d,h and Extended Data Fig. 3a,e).

In summary, REM transcription factors VDD, VAL, REM12, REM13
and REM19, along with REM22 and REM8 to a lesser extent, were
enriched at group 1 CLSY3 CLSY4-dependent siRNA loci (Fig. 3b–e,g,h
and Extended Data Fig. 3a,c,e), suggesting functional collaboration in
targeting these loci. Meanwhile, REM8, together with REM19, localized
to a subset of group 2 of CLSY3 CLSY4-dependent siRNA loci (Fig. 3b–d,h
and Extended Data Fig. 3a,e), where the CLSY3/4–Pol IV complex
became enriched in the absence of *GDE1*. Collectively, the transcription
factors described here can bind over 28% of CLSY3 CLSY4-dependent
siRNA loci (Fig. 3i).

## GDE1 controls siRNA production and DNA methylation at siren sites

CLSY3/4 are required for siRNA production at siren sites, the predomi-
nate siRNA loci in female reproductive tissues (ovules)[11]. Notably, 86
out of 133 siren loci contained the same binding motif as CLSY3 CLSY4
motif 1 (Extended Data Fig. 4a), which was also exhibited by the VDD,
REM12 and REM13 (Extended Data Fig. 3b). These transcription factors
were enriched at more than 77% of the siren loci (Fig. 4a and Extended
Data Fig. 4b), and the strongly bound subset of group 1 sites primar-
ily corresponds to the siren loci (Fig. 4b). Moreover, REM19, REM22
and REM8 were found at siren loci with somewhat lower enrichment
(Extended Data Fig. 4b). Furthermore, GDE1 and the Pol IV recruiters,
CLSY3 and CLSY4, displayed substantial enrichment at siren loci (Fig. 4b
and Extended Data Fig. 4c). These results suggest that REM–GDE1 com-
plexes may recruit CLSY3/4 to siren loci to regulate siRNA levels.

To test the function of GDE1 at siren loci, 24-nt siRNA were
measured and we found that, in *gde1-1* ovules, 24-nt siRNA exhibited
a strong reduction, phenocopying the *clsy3 clsy4* double mutant
(Fig. 4c and Extended Data Fig. 4d). This dramatic loss of 24-nt siRNA

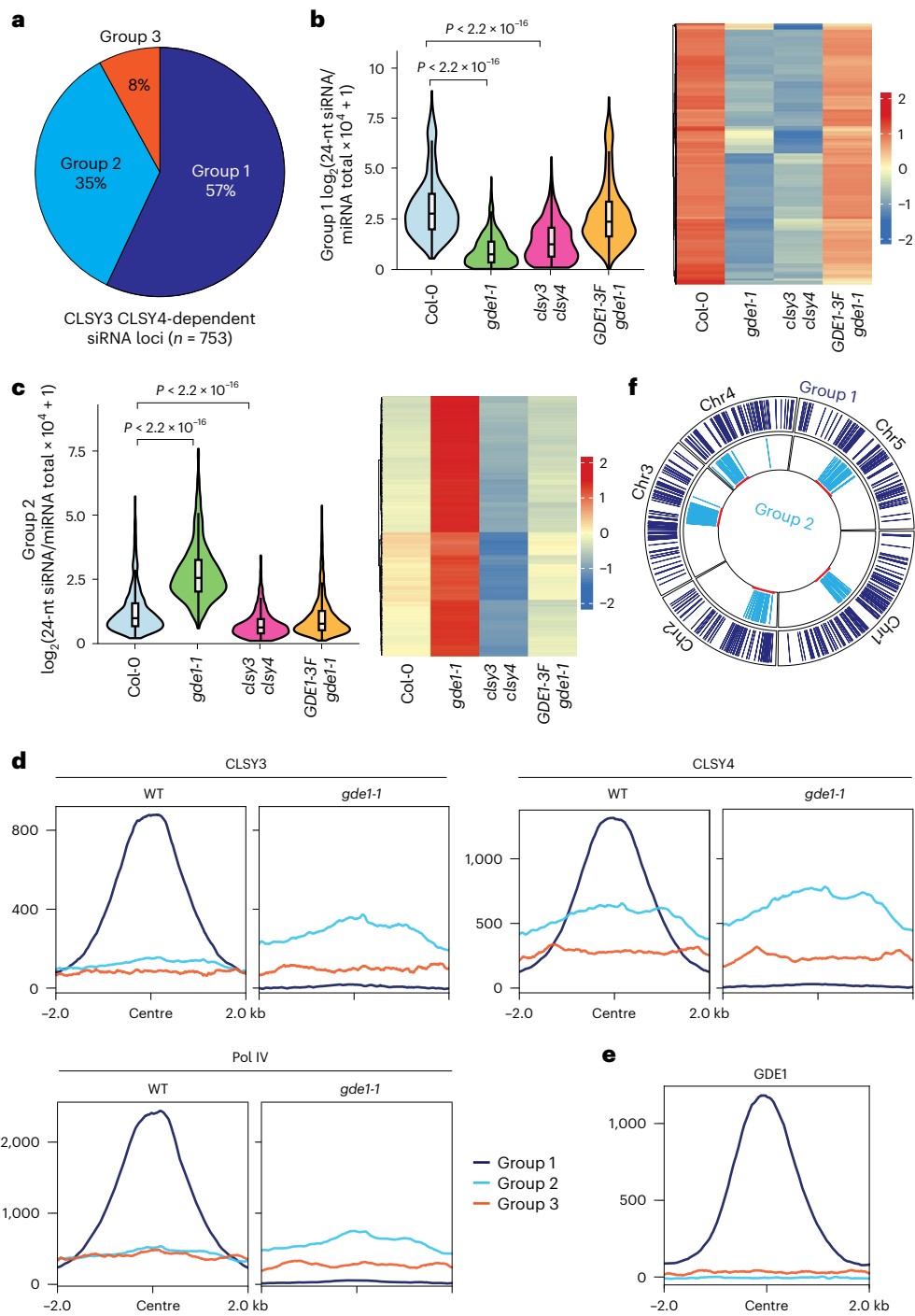

**Fig. 2 | GDE1 localizes at a subset of CLSY3 CLSY4-dependent siRNA loci for siRNA production and the loss of GDE1 redistributes the CLSY3/4–Pol IV complex to other loci. a**, A pie chart showing the percentage of three groups of CLSY3 CLSY4-dependent siRNA sites. **b,c**, Violin plots and heatmaps showing 24-nt siRNA levels in group 1 (**b**) (n = 429) and group 2 (**c**) (n = 262) of CLSY3 CLSY4-dependent sites. Pairwise t-tests showed $P = 4.4805 \times 10^{-123}$ (Col-0 versus gde1-1) and $P = 1.4105 \times 10^{-114}$ (Col-0 versus clsy3 clsy4) for group 1, and $P = 2.71776 \times 10^{-52}$ (Col-0 versus gde1-1) and $P = 2.81529 \times 10^{-48}$ (Col-0 versus clsy3 clsy4) for group 2. The line in the centre of each violin plot represents the

median. The thick grey bar in the centre represents the interquartile range. The whiskers represent the rest of the distribution. **d**, Metaplots showing enrichment of CLSY3/4 and Pol IV ChIP-seq signal over three groups of CLSY3 CLSY4-dependent sites in WT and gde1-1 backgrounds. The region is −2 kb to 2 kb from the centre. Same regions in gde1-1 were used for all metaplots. **e**, A metaplot showing enrichment of GDE1 ChIP-seq signal over three groups of CLSY3 CLSY4-dependent sites. **f**, A circular genome view showing the enrichment of groups 1 and 2 of CLSY3 CLSY4-dependent siRNA across all five chromosomes, with the pericentromeric heterochromatin marked in red along the inner circle.

is probably due to the significant loss of CLSY3/4 and Pol IV at siren loci in gde1-1, which did not occur at non-siren loci (Extended Data Fig. 4e–g). Consistent with the fact that CHH and CHG methylation levels are affected in siRNA biogenesis mutants[40], we found that these types of methylation were significantly reduced at siren loci in

gde1-1 ovules, resembling the pattern observed in the clsy3 clsy4 double mutant (Fig. 4d and Extended Data Fig. 4h). Taken together, these results indicate that GDE1 is required for CLSY3/4 and Pol IV recruitment, siRNA production and DNA methylation at siren loci in ovules.

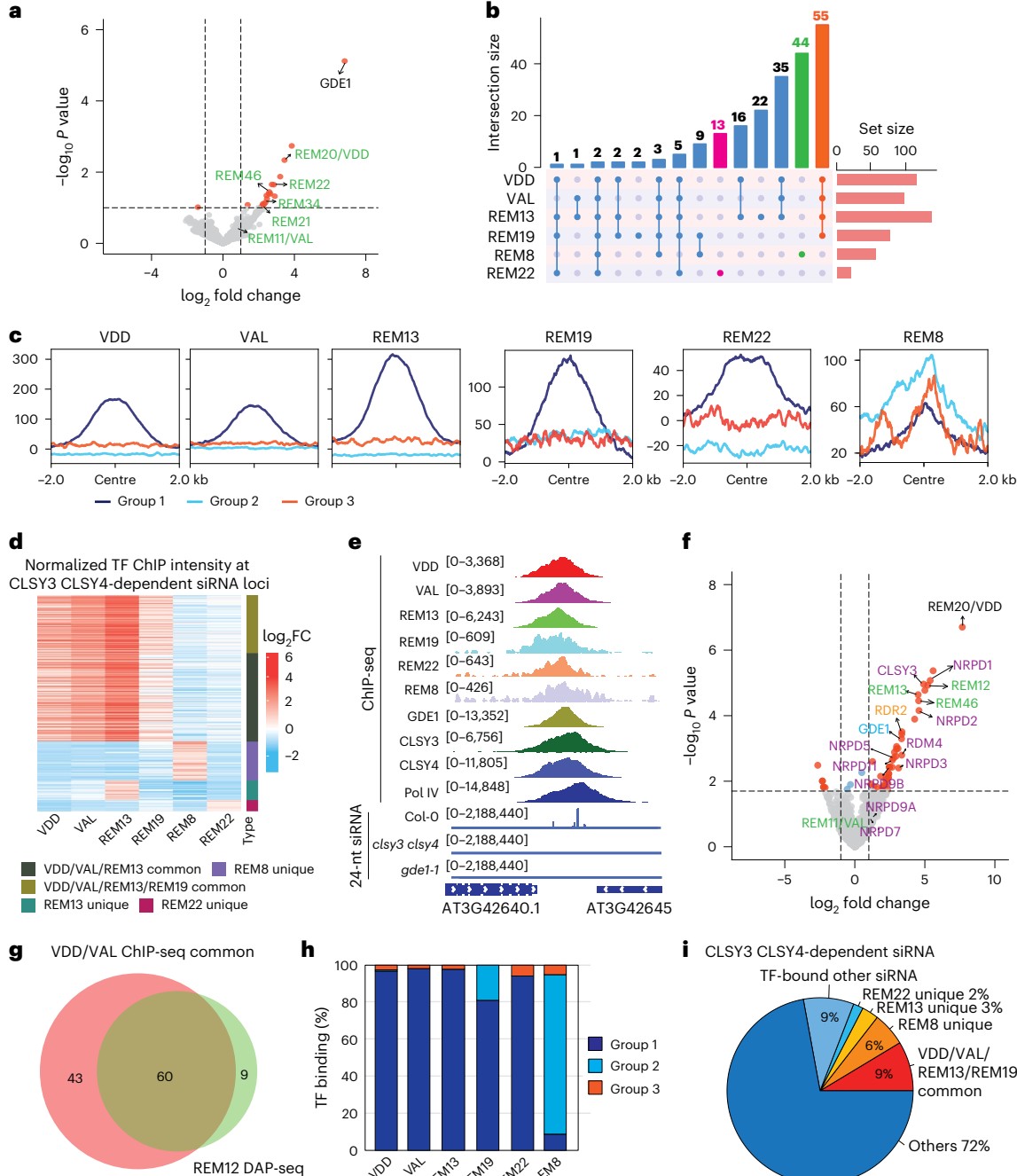

**Fig. 3 | REM transcription factors associate with GDE1 and Pol IV complex at CLSY3 CLSY4-dependent siRNA loci. a**, A volcano plot showing proteins that have significant interactions with GDE1 as detected by IP–MS, with REM type transcription factors labelled with green letters. The two-sided empirical Bayes test performed by LIMMA was used for statistical analysis. **b**, An upset plot displaying the comparative analysis among all tested transcription factors bound to CLSY3 CLSY4-dependent siRNA loci. VDD, VAL, REM13 and REM19 common targets are labelled with orange, REM8 unique target is labelled with green and REM22 unique target is labelled with magenta. **c**, A metaplot showing enrichment of VDD, VAL, REM13, REM19, REM22 and REM8 ChIP-seq signal over three groups of CLSY3 CLSY4-dependent sites. The region is −2 kb to 2 kb from the centre. Same regions were used for all metaplots. **d**, A heatmap showing normalized transcription factor ChIP-seq intensity at CLSY3 CLSY4-dependent siRNA loci. **e**, A screenshot of VDD,

VAL, REM13, REM19, REM22, REM8, GDE1, CLSY3, CLSY4 and Pol IV ChIP-seq at a representative group 1 of CLSY3 CLSY4-dependent siRNA sites. Square brackets indicate the range on a bar graph. **f**, A volcano plot showing proteins that have significant interactions with VDD as detected by IP–MS, with REM type transcription factors labelled with green letters, GDE1 labelled with cyan letters and CLSY3, RDM4 and Pol IV subunits labelled with purple letters. The two-sided empirical Bayes test performed by LIMMA was used for statistical analysis. **g**, A Venn diagram showing the similarities between VDD and VAL ChIP-seq common targeted siRNA loci with REM12 DAP-seq targeted siRNA loci. **h**, A bar plot showing the percentage of each REM transcription factor binding to three distinct groups of CLSY3 CLSY4-dependent siRNA loci. **i**, A pie chart displaying the percentage of transcription factors binding to CLSY3 CLSY4-dependent siRNA loci.

## REM transcription factors promote the biogenesis of siRNA

VDD has previously been implicated in controlling the death of the receptive synergid cell during fertilization[37]. Loss-of-function mutants exhibit strong female gametophytic defects[37], and VDD–9myc fully restored the lethality phenotype (Extended Data Fig. 5a). The genetic lethality of knockout mutants and the multitude of these REM transcription factors make mutant analysis challenging. VAL, REM12 and a VDD interactor, REM46, form a gene cluster in the genome. To create partial loss-of-function mutants of the siren targeting REM transcription factors, we utilized the CRISPR–Cas9 editing system to simultaneously delete all three genes (Extended Data Fig. 5b). The 24-nt siRNA levels at siren loci were significantly reduced in two independent *rem46 val rem12 triple* mutants (Fig. 4e,f), indicating the involvement of these REM transcription factors in regulating siRNA levels at siren loci. Furthermore, the siRNA level at siren loci in *rem46 val rem12 gde1-1* or *rem46 val rem12 clsy3 clsy4* mutants resembled those of their respective parents *gde1-1* or *clsy3 clsy4*, further suggesting that REM transcription factors function in the same pathway with GDE1 and CLSY3/4 at siren loci (Fig. 4e). In addition, around 30% of CLSY3 CLSY4-dependent siRNA loci showed reduced siRNA levels in the *rem46 val rem12* mutants, although not to the same extent as in the *clsy3 clsy4* mutant (Fig. 4g and Extended Data Fig. 5c). All of these reduced siRNA in the *rem46 val rem12* mutants belonged to group 1 of CLSY3 CLSY4-dependent sites, implying broader effects of these REM transcription factors on regulating siRNA production than those revealed by ChIP-seq.

We also utilized a gain-of-function approach by fusing the REM transcription factor VDD with the artificial zinc finger 108 (ZF), which target hundreds of binding sites throughout the *Arabidopsis* genome[41,42]. A VDD–ZF fusion was transformed into the WT background, and ChIP-seq was performed to identify the binding sites. We identified 397 clear peaks with ZF binding to the genome which were highly enriched for the known ZF binding motif (Extended Data Fig. 5d). Small RNA sequencing analysis of the VDD–ZF transgenic plants revealed that 24-nt siRNAs were produced de novo at many of the ZF fusion targeted regions in ovule tissues (Fig. 4h–i), indicating the sufficiency of REM transcription factors to promote siRNA biogenesis.

To investigate whether the localization of REM transcription factors was affected by GDE1 or CLSY3/4. We performed ChIP-seq using flower tissues collected from VDD–9myc *gde1-1* and VDD–3FLAG *clsy3 clsy4* and found that the ChIP-seq signals of VDD were reduced significantly at siren loci in both *gde1-1* and *clsy3 clsy4* mutants (Fig. 4j). These results suggest that GDE1 and CLSY3/4 are both involved in the stabilization of REM transcription factors to siren loci.

## REM factors drive siRNA biogenesis independently of H3K9me

H3K9 methylation has been reported to recruit Pol IV for siRNA biogenesis[9,30]. To test the relationship between REM transcription factors and H3K9 methylation-initiated RdDM pathways, siRNA levels were measured in ovule tissues of *suvh4 suvh5 suvh6* triple mutant, where H3K9 methylation is largely lost[27]. H3K9 methylation-dependent siRNAs largely overlapped with CLSY1 CLSY2-dependent loci, while there was little overlap with REM transcription factors- dependent or CLSY3 CLSY4-dependent loci (Extended Data Fig. 6a,b). Furthermore, no significant reduction at siren loci was observed in either *clsy1 clsy2* double or *suvh4 suvh5 suvh6* triple mutants (Extended Data Fig. 6c), suggestive of independence of REM transcription factor-initiated siRNA production with H3K9 methylation. Collectively, in ovule tissue, 20.25% of Pol IV-dependent 24nt-siRNAs reside in clusters reduced in the *clsy3 clsy4* double mutant and REM transcription factor-GDE1 recruitment appears to be the dominant mechanism operating at these sites (Extended Data Fig. 6d). By contrast, 9.68% depended on CLSY1 CLSY2, where H3K9 methylation plays a crucial role (Extended Data Fig. 6d).

## Essential role of the RBHG domain in GDE1 function

All transcription factors (VDD, VAL, REM12 and REM13) capable of recognizing the CLSY3 CLSY4 motif 1 feature two B3 domains, located at both ends of the proteins (Extended Data Fig. 7a). AlphaFold3[43] confidently predicted dimerization of these factors through their N-terminal B3 domains, while their C-terminal B3 domains were predicted to be responsible for DNA recognition across almost all combinations (Fig. 5a,b and Extended Data Fig. 7a). Notably, an α-helix of GDE1 (amino acids 249–261, hereafter named RBHG (REM-binding helix of GDE1) domain) was predicted to fit into a pocket formed by the transcription factor dimers, establishing extensive electrostatic and hydrophobic interactions (Fig. 5a–c and Extended Data Fig. 7a). To experimentally assess the significance of the α-helix of GDE1's predicted association with REM transcription factor dimers, a triple mutant (E251A/Y252A/Y255A) was generated and introduced into the *gde1-1* mutant. Even though this mutant produced GDE1 transcript levels comparable to the WT GDE1, it did not fully rescue the siRNA deficiency of the *gde1-1* mutant (Fig. 5d and Extended Data Fig. 7b). These results suggest that this α-helix of GDE1 is critical for function, consistent with its proposed role in interacting with REM transcription factors.

## REM–GDE1 recruit CLSY3/4–Pol IV for unidirectional transcription

The CLSY3 CLSY4 motif 1 exhibits two or three tandem TTTTGCTTAT sequences with a single nucleotide spaced between them (Fig. 1d and Extended Data Fig. 3b). The REM proteins belong to the B3 DNA-binding domain superfamily, and some family members, the auxin response factors, possess the ability to form dimers that recognize sequences with a spacer of a specific length[44,45]. Throughout the genome, the occurrence of the three repeats ($n = 38$) and two repeats ($n = 48$) is rare, with 37,456 loci having a single repeat. ChIP-seq signals for all components (VDD, VAL, REM13, GDE1, CLSY3, CLSY4 and Pol IV) showed strong enrichment at all triple-repeat sites and most double-repeat sites, but very minor enrichment at the single-repeat sites (Fig. 6a and Extended Data Fig. 8a). These findings suggest that at least double repeats are required for robust recruitment of the REM transcription factors–GDE1–CLSY3/4–Pol IV transcription complex.

It was notable that there was always a one-nucleotide space between repeats with strong binding, leading us to speculate that this space might allow proper spatial dimerization of transcription factors. To examine this further, we identified 27 loci in the genome that contained double repeats but with a two-nucleotide space. Interestingly, we found that no factors in the Pol IV transcription complex were enriched at these loci (Extended Data Fig. 8b). In addition, we performed DAP-seq[46] by incubating Halo-tagged recombinant proteins with DNA extracted from WT unopened flower buds to test the ability of the VDD–VAL–GDE1 complex to bind *Arabidopsis* genomic DNA in vitro. All sequences that bound the VDD–VAL–GDE1 complex showed a one-nucleotide spacer (Fig. 6b). Overall, these results strongly support that the REM–GDE1 complex can directly bind the CLSY3 CLSY4 motif 1 and that the single-nucleotide spacer is important.

We found that REM transcription factors and GDE1 ChIP-seq peak summits, but not CLSY3/4–Pol IV summits, localized to the centre of motifs (Fig. 6a,c). On average, CLSY3/4 and Pol IV were located approximately 200 or 300 base pairs away from the centre of the motif, respectively (Fig. 6c). In addition, CLSY3/4 and Pol IV summits were found on either one side or the other side of the centre of the motif, but not on both sides (Fig. 6a,c). To further characterize the direction of CLSY3/4 and Pol IV distribution relative to the sequence motif, we separated motifs into two groups based on the DNA strand that contained the TTTTGCTTATNTTTTGCTTAT motif sequence in the genome. We found that the distribution of CLSY3/4 and Pol IV signals was still localized to either one side or the other side of the motif centres (Extended Data Fig. 8c), indicating that the direction of the motif alone does not

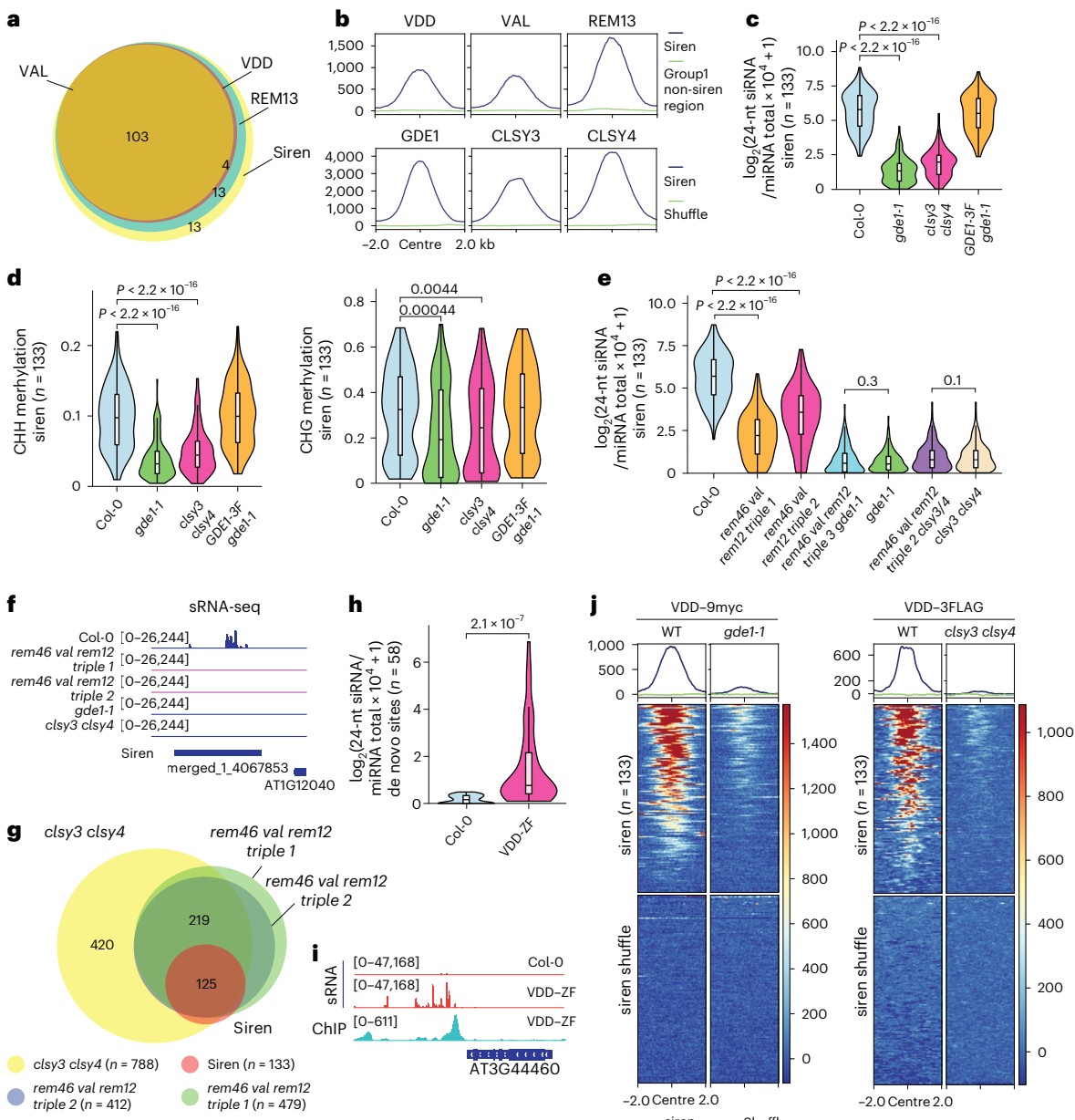

**Fig. 4 | VDD/VAL/REM13–GDE1–CLSY3/4 complex localizes to siren loci for siRNA production. a**, A Venn diagram showing the relationship between siren loci and VDD, VAL and REM13 ChIP-seq targets. **b**, Top: metaplots showing enrichment of VDD, VAL and REM13 ChIP-seq signal over siren and group 1 non-siren region. Bottom: metaplots showing enrichment of GDE1, and CLSY3/4 ChIP-seq signal over siren loci. The region is −2 kb to 2 kb from the centre. Same regions were used for all metaplots. **c**, A violin plot showing the 24-nt siRNA levels at siren loci (n = 133) in the indicated genotypes. Pairwise t-tests showed P = 2.64555 × 10^{−80} (Col-0 vs gde1-1) and P = 9.98545 × 10^{−70} (Col-0 versus clsy3 clsy4). The line in the centre of each violin plot represents the median. The thick black bar in the centre represents the interquartile range. The whiskers represent the rest of the distribution. **d**, A violin plot showing the CHH and CHG methylation levels at siren loci (n = 133) in the indicated genotypes. P values calculated by pairwise t-tests are indicated. Pairwise t-tests showed P = 7.02463 × 10^{−29} (Col-0 versus gde1-1) and P = 2.52025 × 10^{−20} (Col-0 versus clsy3 clsy4) for CHH methylation. The line in the centre of each violin plot represents the median. The thick black bar in the centre represents the interquartile range. The whiskers represent the rest of the distribution. **e**, A violin plot showing the 24-nt siRNA levels at siren loci (n = 133) in the

indicated genotypes. P values calculated by pairwise t-tests are indicated. Pairwise t-tests showed P = 2.41619 × 10^{−56} (Col-0ov versus rem46 val rem12 triple 1 ov) and P = 3.18267 × 10^{−22} (Col-0 ov versus rem46 val rem12 triple 2 ov). The line in the centre of each violin plot represents the median. The thick black bar in the centre represents the interquartile range. The whiskers represent the rest of the distribution. **f**, A screenshot of 24-nt siRNA levels at a representative siren site. Square brackets indicate the range on a bar graph. **g**, A Venn diagram showing the relationship of downregulated siRNA loci in each background with siren loci. **h**, A violin plot showing the 24-nt siRNA levels at de novo ZF fusion targeted sites (n = 58) in the indicated genotypes. P values calculated by pairwise t-tests are indicated. The line in the centre of each violin plot represents the median. The thick black bar in the centre represents the interquartile range. The whiskers represent the rest of the distribution. **i**, A screenshot of 24-nt siRNA and VDD–ZF ChIP-seq at a de novo ZF fusion loci. Square brackets indicate the range on a bar graph. **j**, Metaplot and heatmap showing enrichment of VDD ChIP-seq signal over siren loci in Col-0, gde1-1 and clsy3 clsy4 double backgrounds. The region is −2 kb to 2 kb from the centre. Same regions were used for all metaplots.

**a**

| | GDE1 sites | TF | AA sites | Types of interaction |
|---|---|---|---|---|
| VDD–VAL–GDE1 motif 1 | E251 | VAL | K17 | Hydrogen bound |
| | Y252 | VDD | N94 | Hydrogen bound |
| | Y252 | VAL | F16 | Pi bound |
| | D249 | VAL | K17 | Hydrogen bound |
| | D249 | VDD | A114 | Hydrogen bound |
| | K254 | VAL | E23 | Hydrogen bound |
| | Y255 | VAL | G88 | Hydrogen bound |
| REM12–REM13–GDE1 motif 1 | E251 | REM13 | T32 | Hydrogen bound |
| | Y252 | REM12 | G89 | Hydrogen bound |
| VDD–REM13–GDE1 motif 1 | E251 | REM13 | T32 | Hydrogen bound |
| | Y252 | VDD | N94 | Hydrogen bound |
| | Y255 | REM13 | F96 | Pi bound |
| | L256 | VDD | R25 | Hydrogen bound |
| VDD–REM12–GDE1 motif 1 | E251 | REM12 | K16 | Hydrogen bound |
| | Y252 | REM12 | F15 | Hydrogen bound |
| | K254 | REM12 | E22 | Hydrogen bound |
| | Y255 | REM12 | G89 | Hydrogen bound |
| | H258 | REM12 | H109 | Pi bound |
| | R263 | VDD | M34 | Hydrogen bound |
| | R263 | VDD | M33 | Hydrogen bound |

| | GDE1 sites | TF | AA sites | Types of interaction |
|---|---|---|---|---|
| VAL–REM13–GDE1 motif 1 | E251 | VAL | L115 | Hydrogen bound |
| | Y255 | REM13 | F101 | Hydrogen bound |
| | Y494 | REM13 | P94 | Hydrogen bound |
| VAL–REM12–GDE1 motif 1 | Y252 | VAL | F16 | Pi bound |
| | Y252 | REM12 | G89 | Hydrogen bound |
| | K254 | VAL | E23 | Hydrogen bound |
| | Y255 | VAL | G88 | Hydrogen bound |
| VDD–VDD–GDE1 motif 1 | Y255 | VDD_1 | F8 | Hydrogen bound |
| | H258 | VDD_2 | K5 | Hydrogen bound |
| | H258 | VDD_2 | K5 | Pi bound |
| REM13–REM13–GDE1 motif 1 | E251 | REM13_1 | T32 | Hydrogen bound |
| | S260 | REM13_2 | Q31 | Hydrogen bound |
| | T262 | REM13_1 | Y114 | Hydrogen bound |
| | R263 | REM13_2 | T32 | Hydrogen bound |
| REM12–REM12–GDE1 motif 1 | E251 | REM12_2 | S23 | Hydrogen bound |
| | Y252 | REM12_1 | G89 | Hydrogen bound |
| | Y252 | REM12_2 | F15 | Pi bound |
| VAL–VAL–GDE1 motif 1 | E251 | VAL_1 | K17 | Hydrogen bound |
| | Y252 | VAL_2 | F16 | Pi bound |
| | K254 | VAL_1 | E23 | Hydrogen bound |
| | Y255 | VAL_2 | G88 | Hydrogen bound |

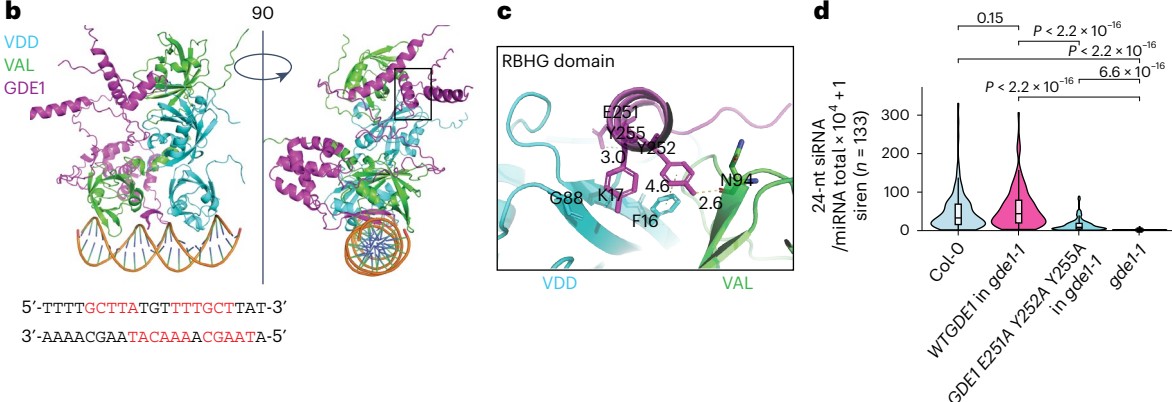

**b**

VDD
VAL
GDE1

90

5′-TTTT**GCTTA**TGT**TTTGCT**TAT-3′
3′-AAAACGAA**TACAAA**A**CGAAT**A-5′

**c**

RBHG domain

E251
Y255
Y252
3.0
G88
4.6
K17
F16
N94
2.6

VDD   VAL

**d**

0.15
$P < 2.2 \times 10^{-16}$
$P < 2.2 \times 10^{-16}$
$6.6 \times 10^{-16}$
$P < 2.2 \times 10^{-16}$

24-nt siRNA /miRNA total × $10^4$ + 1 siren ($n$ = 133)

Col-0 | WTGDE1 in *gde1-1* | GDE1 E251A Y252A Y255A in *gde1-1* | *gde1-1*

**Fig. 5 | The RBHG domain of GDE1 is critical for siRNA production. a**, A table showing the residues and types of interaction formed between RBHG domain of GDE1 and transcription factor dimers. AA, amino acid. **b**, AlphaFold3 predicted the structure of VDD–VAL–GDE1 bound to dsDNA containing TTTTGCTTATGTTTTGCTTAT (sequence below, high-affinity nucleotide acids red bolded). The DNA is shown as a ribbon representation. **c**, Interaction of the RBHG domain of GDE1 with VDD–VAL. Interacting residues are highlighted with sticks. Hydrogen bonds and pi bonds are highlighted with dashed lines. **d**, A violin plot showing the 24-nt siRNA levels at siren loci ($n$ = 133) in the indicated genotypes. $P$ values calculated by pairwise $t$-tests are indicated. Pairwise $t$-tests showed $P = 2.6169 \times 10^{-24}$ (WTGDE1 in *gde1-1*ov versus *gde1-1*ov), $P = 9.5561 \times 10^{-17}$ (WTGDE1 in *gde1-1*ov versus GDE1 E251A Y252A y255A in *gde1-1*ov), and $P = 1.39057 \times 10^{-19}$ (Col-0ov versus *gde1-1*ov). The line in the centre of each violin plot represents the median. The thick black bar in the centre represents the interquartile range. The whiskers represent the rest of the distribution.

determine the direction of Pol IV transcription. An alternative explanation is that the directionality of Pol IV transcription may be influenced by the chromatin environment of the regions. To explore this, we divided the motifs into two additional groups based on the relative positions of Pol IV summits and observed that the direction of the Pol IV shift closely aligned with more nucleosome-occupied regions (Fig. 6d). It is not clear, however, if the higher nucleosome density is a cause or a consequence of Pol IV transcription. In any case, these results suggest that CLSL3/4 and Pol IV are recruited to the CLSY3 CLSY4 motif 1 by the REM transcription factor–GDE1 complex, after which they engage in unidirectional transcription (to one side of the motif or the other) to produce Pol IV transcripts required to produce siRNAs. Consistent with this model, the accumulation levels of 24-nt siRNA transcripts aligned with the localization pattern of the CLSY3/4–Pol IV complex at the CLSY3 CLSY4 motif 1 (Fig. 6d).

**REM8 recognizes CLSL3 CLSY4 motif 2 for siRNA production**

As opposed to the above-described REM transcription factors that appear to mainly drive CLSY3/4–Pol IV complex to group 1 loci of CLSY3 CLSY4-dependent siRNA, we found that REM8 exhibited unique binding sites within group 2 sites (Fig. 3b–d,h and Extended Data Fig. 3a,e).

A REM8 deletion mutant was generated using CRISPR–Cas9 (Extended Data Fig. 9a), but this mutation did not affect siRNA production at REM8 binding sites (Extended Data Fig. 9b). Given the genetic redundancy of REM transcription factors at the CLSY3 CLSY4 motif 1, it is likely that additional factors contribute to the recognition of motif 2. In line with upregulation of 24-nt siRNA in *gde1-1* at group 2 loci (Fig. 2a,c), we found that more 24-nt siRNAs were detectable at REM8-binding non-siren regions in *gde1-1* (Fig. 7a), which is correlated with the enhanced enrichment of CLSY3/4 and Pol IV enrichment at these sites in the *gde1-1* background (Extended Data Fig. 9c). This also shows that siRNAs accumulate at these sites in a GDE1-independent manner and that REM8 acts via a different mechanism to recruit Pol IV complexes.

Motif analysis of REM8 peaks revealed three repeats of the AAGCGGAT sequence with one-nucleotide spaced between them (Fig. 7b) (hereafter named the CLSY3 CLSY4 motif 2). Fifty CLSY3 CLSY4 motif 2 sites were identified, matching to 22 24-nt siRNA loci throughout the genome, and REM8 and CLSY3/4 were enriched at most of them (Extended Data Fig. 9d). siRNA levels at CLSY3 CLSY4 motif 2 loci decreased in the *clsy3 clsy4* mutant but increased in the *gde1-1* mutant (Fig. 7c). These results suggest that REM8 may act together with other unidentified factors to recruit the CLSY3/4–Pol IV complex to CLSY3

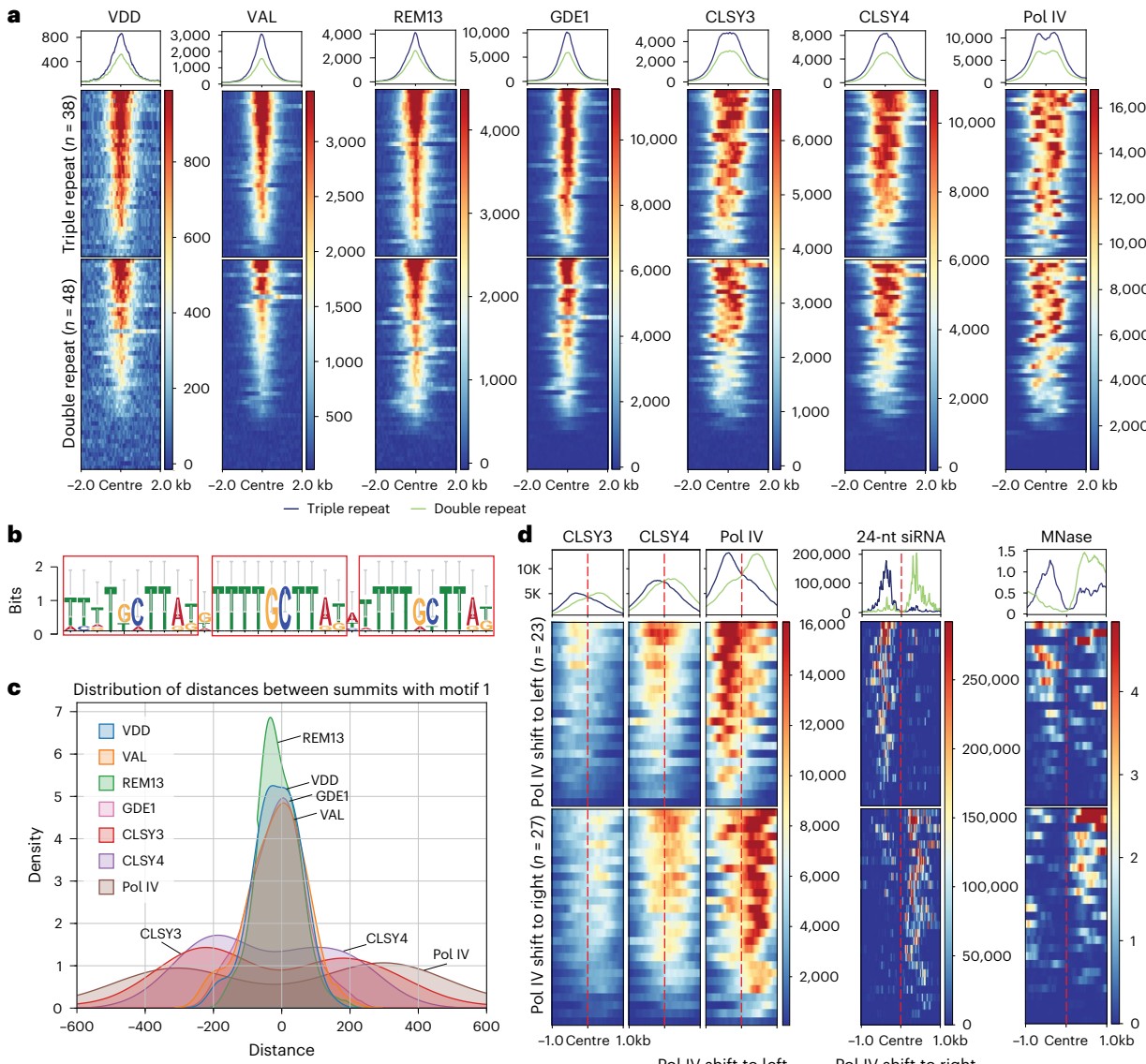

**Fig. 6 | Pol IV transcription complex recognizes CLSY3 CLSY4 motif 1 and engages in a unidirectional transcription. a**, Metaplot and heatmap showing enrichment of VDD, VAL, REM13, GDE1, CLSY3, CLSY4 and Pol IV ChIP-seq signal over triple-repeat sites (*n* = 38) and double-repeat sites (*n* = 48). **b**, Sequence logo assessment of DNA sequences identified by VDD–VAL–GDE1 DAP-seq. **c**, The position of VDD, VAL, REM13, GDE1, CLSY3, CLSY4 and Pol IV ChIP-seq summits to the closest CLSY3 CLSY4 motif 1 centre. **d**, Metaplot and heatmap showing enrichment of CLSY3, CLSY4, Pol IV ChIP-seq, MNase-seq signal and 24-nt siRNA signal over Pol IV left shift (*n* = 23) or Pol IV right shift (*n* = 27). The region is −1 kb to 1 kb from the centre. Same regions were used for all metaplots.

CLSY4 motif 2 for 24-nt siRNA production and that this recruitment is more robust in the *gde1-1* mutant, where CLSY3/4–Pol IV are released from the siren loci.

### GDE1 is required for anther siRNA production

CLSY3 is also required for the biogenesis of siRNAs at hyperTE loci in male meiocyte cells[10]. Notably, there is a limited overlap of only 12 shared loci between the maternal siren loci and paternal hyperTE loci[12], suggesting distinct siRNA biogenesis mechanisms. In tapetum cells, most REM transcription factors, with the exception of REM46 and REM16, are very lowly expressed (Extended Data Fig. 3d). Intriguingly, GDE1 stands out as showing very high expression levels in tapetum cells (Extended Data Fig. 3d), suggestive of a potential role in meiocyte cells. 449 siRNA loci showed reduction in the anthers of the *clsy3 clsy4* mutant (Fig. 7d), and over half of these loci exhibited similar levels of reduction in the anthers of the *gde1-1* mutant (Fig. 7e,f). However, hardly any differentially expressed siRNAs were found in *rem46 val rem12*

*triple* mutants (Fig. 7d). Consistently, only GDE1, CLSY3/4 and Pol IV, but none of the tested REM transcription factors, showed enrichment at anther CLSY3 CLSY4-dependent siRNA loci (Extended Data Fig. 10). Together, these results suggest that GDE1 is also required for anther siRNA production, but probably via a mechanism different from that seen in ovules.

### Discussion

Our study uncovers distinct mechanisms by which CLSY proteins recruit Pol IV complexes to direct siRNA production, highlighting an alternative paradigm in the regulation of DNA methylation and siRNA biogenesis. While CLSY1 and CLSY2 operate in association with SHH1, an H3K9 methylation reader, to recruit Pol IV to chromatin via histone marks and other epigenetic modifications[9,30,31,47,48], we demonstrate that CLSY3 and CLSY4 recruit Pol IV in a fundamentally different manner. In ovules, CLSY3 and CLSY4 rely on GDE1, which acts in concert with REM transcription factors to direct siRNA production to specific

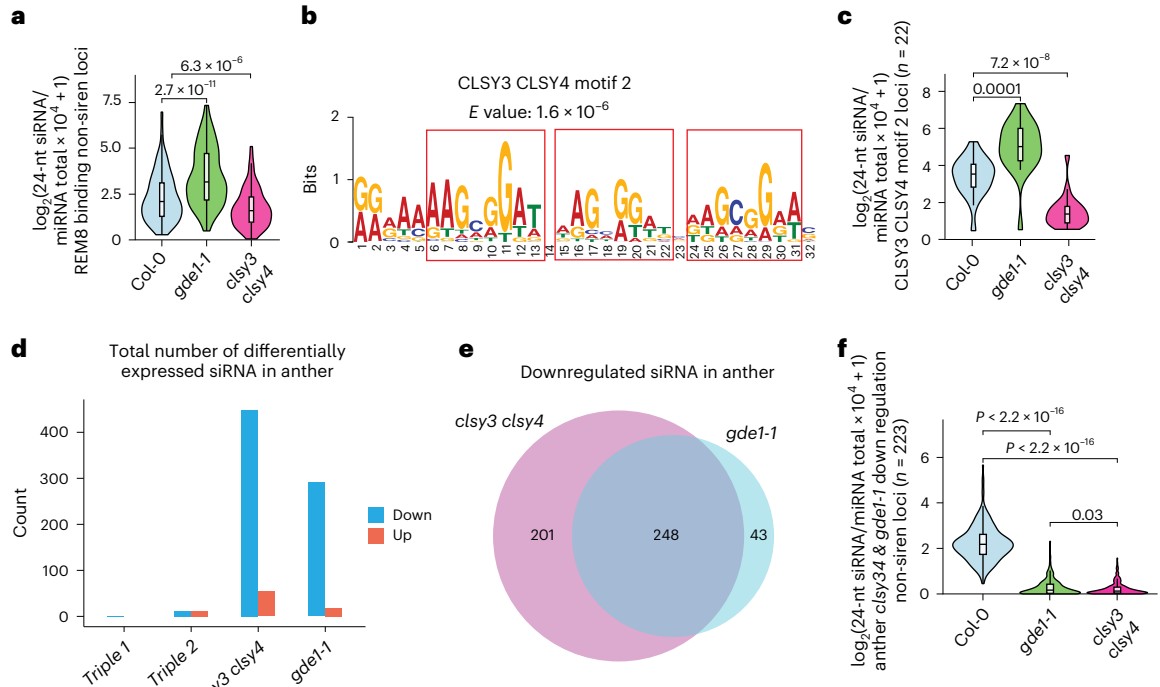

**Fig. 7 | REM8 redistributes Pol IV complex to CLSL3 CLSY4 motif 2 in the absence of GDE1, and GDE1 is required for siRNA in anther. a**, A violin plot showing the 24-nt siRNA levels at REM8-binding non-siren loci ($n = 162$) in the indicated genotypes. $P$ values calculated by pairwise $t$-tests are indicated. The line in the centre of each violin plot represents the median. The thick black bar in the centre represents the interquartile range. The whiskers represent the rest of the distribution. **b**, MEME analysis of REM8-binding non-siren loci. $E$ value is the expected number of false positives in the matches up to this point. **c**, A violin plot showing the 24-nt siRNA levels at CLSL3 CLSY4 motif 2 ($n = 22$) in the indicated genotypes. $P$ values calculated by pairwise $t$-tests are indicated. The line in the centre of each violin plot represents the median. The thick black

bar in the centre represents the interquartile range. The whiskers represent the rest of the distribution. **d**, The total number of differentially expressed siRNA in the anther tissue of indicated genotypes. **e**, A Venn diagram showing the relationship between CLSY3 CLSY4-dependent and GDE1-dependent siRNA loci in anther. **f**, A violin plot showing the 24-nt siRNA levels at anther *clsy3 clsy4* and *gde1-1* downregulation non-siren loci ($n = 223$) in the indicated genotypes. $P$ values calculated by pairwise $t$-tests are indicated. Pairwise $t$-tests showed $P = 2.4074 \times 10^{-120}$ (Col-0an versus *clsy34*an) and $P = 2.0517 \times 10^{-119}$ (Col-0an versus *gde1-1*an). The line in the centre of each violin plot represents the median. The thick black bar in the centre represents the interquartile range. The whiskers represent the rest of the distribution.

sequences. Our work suggests that CLSY proteins have evolved multiple recruitment strategies, some driven by epigenetic features and others by sequence-specific genetic information, allowing flexible and tissue-specific control of siRNA production.

Our study identifies GDE1 as a critical mediator that enhances the recruitment of CLSY3/4 to siren loci. The strong enrichment of CLSY3/4 and Pol IV at siren loci is lost in the *gde1-1* mutant, where these complexes are redistributed to group 2 loci. This redistribution underscores the specificity of GDE1 in guiding siRNA biogenesis at specific CLSY3 CLSY4-dependent loci, reinforcing its central role in the spatial and functional organization of Pol IV recruitment.

VDD and VAL, two REM transcription factors that target siren loci, have been reported as critical for the degeneration of synergid cells, a vital step in ensuring successful fertilization[37]. *vdd-1/+* and *VAL RNAi* mutants exhibit pronounced female gametophytic defects, underscoring the importance of these factors in reproductive success. However, the viability of our *rem46 val rem12* triple mutant in this study suggests that the reduced fertility observed in the *VAL RNAi* mutant may result from off-target silencing of related homologues. The expression of *GAMETOPHYTIC FACTOR 2* (*GFA2*), a key regulator of synergid cell death, can rescue the phenotypic effects observed in the *vdd-1/+* mutant[37]. Notably, none of the *gde1-1*, *clsy3 clsy4* and *pol iv* mutants display any obvious developmental phenotypes, implying that these factors may not directly contribute to developmental processes. Our results also suggest diverse regulatory roles of REM transcription factors using siRNA-dependent and siRNA-independent mechanisms.

Although we identified only 86 CLSY3 CLSY4 motif 1 sites across the genome, over 400 loci were affected in the *rem46 val rem12* triple mutants. This observation suggests that REM transcription factors do not exclusively recognize CLSY3 CLSY4 motif 1. Instead, additional transcription factors such as REM19, REM22 and REM8 are also enriched at siren loci. Given their in vitro association[37], these factors may form distinct combinatory heterodimers, enabling broader recruitment and siRNA regulation at loci that lack the CLSY3 CLSY4 motif 1. This mechanism of cooperative binding and redundancy probably enhances the robustness of siRNA biogenesis, ensuring proper siRNA regulation even when individual motifs or factors are absent.

The recruitment of Pol IV by REM transcription factors with the aid of GDE1 draws striking parallels to transcription factor-mediated DNA methylation and gene silencing in mammals. For example, KRAB-zinc finger proteins (KRAB-ZFPs), the largest transcription factor family in mammalian cells, are known to bind specific DNA sequences and recruit the repressor KRAB-associated protein 1 (KAP-1)[49,50], which recruits diverse complexes to regulate DNA methylation, histone deacetylation, H3K9 trimethylation and transposon and gene silencing[51,52]. In an analogous fashion, our results suggest that REM transcription factors recruit CLSY3/4–Pol IV complexes via GDE1 to direct siRNA production and subsequent DNA methylation at target loci. These parallels suggest that sequence-specific DNA binding proteins are used widely throughout different eukaryotic species to properly pattern DNA methylation.

In summary, our work demonstrates that CLSY proteins use distinct strategies to recruit Pol IV, integrating both epigenetic and genetic cues. REM transcription factors and GDE1 provide a sequence-specific

recruitment mechanism for CLSY3/4–Pol IV complexes, adding an additional layer of regulation to siRNA biogenesis and DNA methylation. At other sites and in other tissues, Pol IV employs a primarily epigenetic mechanism for recruitment, using particular histone marks to recruit the machinery for siRNA biogenesis. These findings broaden our understanding of the diversity and complexity of epigenetic regulation and open additional avenues for exploring the interplay between genetic and epigenetic information in development and genome stability.

## Online content

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

[1]State Key Laboratory of Rice Biological Breeding, Key Laboratory of Molecular Biology of Crop Pathogens and Insects, Institute of Biotechnology, Zhejiang University, Hangzhou, China. [2]Department of Molecular, Cell and Developmental Biology, University of California at Los Angeles, Los Angeles, CA, USA. [3]Howard Hughes Medical Institute, University of California at Los Angeles, Los Angeles, CA, USA. [4]Shandong Laboratory of Advanced Agricultural Sciences at Weifang, Peking University Institute of Advanced Agricultural Sciences, Weifang, China. [5]Institute of Plant Biology, National Taiwan University, Taipei, Taiwan. [6]Ministry of Education Key Laboratory for Bio-Resource and Eco-Environment, College of Life Science, State Key Laboratory of Hydraulics and Mountain River Engineering, Sichuan University, Chengdu, China. [7]Eli and Edythe Broad Center of Regenerative Medicine and Stem Cell Research, University of California at Los Angeles, Los Angeles, CA, USA. [8]Department of Biological Chemistry, University of California at Los Angeles, Los Angeles, CA, USA. ✉e-mail: jacobsen@ucla.edu

## Methods

### Plant materials and growth conditions

All *Arabidopsis* plants used in this paper are Col-0 ecotype, and plants are grown under standard condition with 16 h light/8 h dark at 22 °C. The T-DNA insertion lines used in this study included *gde1-1* (SALK-seq_10069.1), *clsy3-1* (SALK_040366) and *clsy4-1* (SALK_003876). *Agrobacterium* (AGL0 strain)-mediated floral dipping was used to generate all the transgenic plants.

### Plasmid construction

For *pGDE1:GDE1-3FLAG*, *pCLSY3:CLSY3-9myc*, *pCLSY4:CLSY4-9myc*, *pPol IV:Pol IV-9myc*, *pVDD:VDD-9myc*, *pVDD:VDD-3FLAG*, *pVAL:VAL-9myc*, *pREM13:REM13-9myc*, *pREM19:REM19-9myc*, *pREM22:REM22-9myc*, *pREM8:REM8-9myc*, *pVDD:VDD-FLAG-ZF108* and the genomic DNA sequences of each with promoter sequences (around 2 kb upstream from the start codon) were first cloned into *pENTR/D-TOPO* vectors (Invitrogen) and then to the destination vector *pEG302-GW-3FLAG*, *pEG302-GW-9Myc* or *pEG302-GW-3FLAG-ZF108* by LR reaction (LR Clonase II, Invitrogen). For *pIX-HALO-VDD*, *pIX-HALO-VAL* and *pIX-HALO-GDE1*, VDD, VAL and GDE1 were cloned from cDNA into the *pENTR/D-TOPO* vector and then to the destination vector *pIX-HALO* by LR reaction. Primers used in this study are presented in Supplementary Table 3.

### ChIP-seq

Around 2.0 g of floral tissues were used for each ChIP. The plant materials were ground into a fine powder with liquid nitrogen and resuspended with nucleus isolation buffer (50 mM HEPES, 1 M sucrose, 5 mM KCl, 5 mM MgCl$_2$, 0.6% Triton X-100, 0.4 mM phenylmethylsulfonyl fluoride (PMSF), 5 mM benzamidine, 1% formaldehyde (Sigma) and 1× protease inhibitor (Roche)) for 10 min with rotation. Then, 1.7 ml of 2 M glycine solution was added immediately to stop the crosslinking. Lysates were filtered through Miracloth, and the nuclei were collected by centrifugation at 4 °C with 2,880g for 20 min. The pellet was resuspended in 1 ml of extraction buffer 2 (0.25 M sucrose, 10 mM Tris–HCl pH 8.0, 10 mM MgCl$_2$, 1% Triton X-100, 5 mM beta-mercaptoethanol (BME), 0.1 mM PMSF, 5 mM benzamidine and 1× protease inhibitor (Roche)) and centrifuged at 12,000g at 4 °C for 10 min. The nuclei were then resuspended with extraction buffer 3 (1.7 M sucrose, 10 mM Tris–HCl pH 8.0, 2 mM MgCl$_2$, 0.15% Triton X-100, 5 mM BME, 0.1 mM PMSF, 5 mM benzamidine, and 1× protease inhibitor (Roche)), at 4 °C with 12,000g for 60 min. The relative pure nuclei were lysed with 400 µl nucleic lysis buffer (50 mM Tris–HCl pH 8.0, 10 mM EDTA, 1% sodium dodecyl sulfate (SDS), 0.1 mM PMSF, 5 mM benzamidine and 1× protease inhibitor (Roche)) on ice for 10 min and a total of 1.7 ml of ChIP dilution buffer (1.1% Triton X-100, 1.2 mM EDTA, 16.7 mM Tris pH 8.0, 167 mM NaCl, 0.1 mM PMSF, 5 mM benzamidine and 1× protease inhibitor (Roche)) was added to the lysed nuclei. Chromatin was sheared by Bioruptor Plus (Diagenode) for 30 cycles with 30 s on/30 s off per cycle. The lysate was centrifuged twice at 4 °C with 20,000g for 10 min, and the supernatant was incubated with either FLAG epitope (Sigma F1804 1:400 dilution) or myc epitope (Cell Signaling, 71D10 1:200 dilution) at 4 °C overnight. Next, the magnetic Protein A and Protein G Dynabeads (Invitrogen) were added and inoculated at 4 °C for 2 h with rotation. The beads were washed with low-salt solution twice (150 mM NaCl, 0.2% SDS, 0.5% Triton X-100, 2 mM EDTA and 20 mM Tris pH 8.0), high-salt solution (150 mM NaCl, 0.2% SDS, 0.5% Triton X-100, 2 mM EDTA and 20 mM Tris pH 8.0), LiCl solution (250 mM LiCl, 1% IGEPAL, 1% sodium deoxycholate, 1 mM EDTA and 10 mM Tris pH 8.0) and TE solution (1 mM EDTA and 10 mM Tris pH 8.0) for 5 min at 4 °C, respectively. The chromatin was eluted with elution buffer (1% SDS, 10 mM EDTA and 0.1 M NaHCO$_3$) and subjected to reverse crosslinking by adding 20 µl 5 M NaCl and incubated at 65 °C overnight. Then, 1 µl of Protease K (20 mg ml$^{-1}$, Invitrogen), 10 µl of 0.5 M EDTA pH 8.0 and 20 µl of 1 M Tris pH 6.5 were added to deactivate the protein for 4 h at 45 °C, and DNA was purified

through phase lock gel (VWR) and precipitated with 1/10 volume of 3 M sodium acetate (Invitrogen), 2 µl GlycoBlue (Invitrogen) and 1 ml 100% ethanol at −20 °C overnight. The precipitated DNA was used for library construction following the manual of the Ovation Ultra Low System V2 kit (NuGEN), and the libraries were sequenced on Illumina NovaSeq 6000 or NovaSeq X Plus instruments.

### IP–MS

Around 10 g of floral tissues from FLAG-epitope-tagged transgenic plants were used for each IP–MS experiment, and floral tissues of Col-0 plants were used as the control. Flower tissue was ground to a fine powder in liquid nitrogen with a homogenizer. Tissue powder was completely resuspended in 25 ml IP buffer (50 mM Tris–HCl pH 8.0, 150 mM NaCl, 5 mM EDTA, 10% glycerol, 0.1% Tergitol, 0.5 mM dithiothreitol, 1 mg ml$^{-1}$ Pepstatin A, 1 mM PMSF, 50 µM MG132 and cOmplete EDTA-free protease inhibitor (Roche)) at 4 °C for 10 min with rotation. The tissue was further disrupted with a Dounce homogenizer. The lysates were filtered with Miracloth and centrifuged at 20,000g for 10 min at 4 °C. The supernatant was incubated with 250 µl anti-FLAG M2 magnetic beads (Sigma) at 4 °C for 2 h with rotation. The magnetic beads were washed four times with IP buffer and eluted with 250 µg ml$^{-1}$ 3×FLAG peptides. Eluted proteins were used for trichloroacetic acid precipitation and mass spectrometric analysis.

### Quantitative proteomics

Protein pellets were resuspended in 8 M urea and 100 mM Tris pH 8.5, then reduced by adding Tris (2-carboxyethyl) phosphine to a final concentration of 5 mM and incubation for 30 min. Next, the proteins were alkylated by adding iodoacetamide to a final concentration of 10 mM for another 30 min at room temperature. Before protein digestion, the urea concentration was diluted to 2 M with 100 mM Tris pH 8.5. Then, the proteins were digested with LysC (BioLabs) at a 1:100 enzyme/protein ratio at 37 °C for 4 h, followed by the trypsin digestion at 1:100 (trypsin:protein) at 37 °C for 12 h. To stop the digestion, 5% formic acid was added to the samples. Next, the peptides were desalted using C18 pipette tips (Thermo Scientific) and reconstituted in 5% formic acid before being analysed by LC–MS/MS. Tryptic peptide mixtures were loaded onto a 25-cm-long, 75-µm-inner-diameter fused-silica capillary, packed in-house with bulk 1.9 µM ReproSil-Pur beads with 120 Å pores as described[53]. The peptides were delivered by a 140-min water–acetonitrile linear gradient in 6–28% buffer (acetonitrile solution, 0.1% formic acid and 3% dimethyl sulfoxide) using a Dionex Ultimate 3000 nanoflow UHPLC (Thermo Scientific), at a flow rate of 200 nl min$^{-1}$, further increased to 35% and followed by a rapid ramp-up to 85%. The eluted peptides were ionized and the Orbitrap Fusion Lumos Tribrid Mass Spectrometer (Thermo Scientific) was used to acquire the mass spectrometric data. The data-dependent acquisition strategy consisted of a repeating cycle of a full MS1 spectrum (resolution 120,000) followed by sequential MS2 scan (resolution 15,000). Label-free quantification (LFQ) was performed using the MaxQuant software package (v1.6.17.0) with LFQ default setting[54], and the Arabidopsis TAIR 10 proteome database was used for the database search. Trypsin digestion was applied, and a maximum of two missed cleavages were allowed in all searches for tryptic peptides of length 8–40 amino acids. In all, 1% false discovery rate (FDR) was used as a filter at both protein and peptide-spectrum match levels. IP–MS of Col-0 plant tissue was used as the control. The empirical Bayes test performed by LIMMA was used for statistical analysis.

### Co-IP

Ten millilitres of floral tissues was collected from GDE1–3FLAG × VDD–9myc, GDE1–3FLAG × CLSY3–9myc, GDE1–3FLAG × Pol IV–9myc, VDD–9myc, CLSY3–9myc and Pol IV–myc. Tissues were ground into a fine powder with liquid nitrogen, mixed with 10 ml IP buffer (50 mM Tris–HCl pH 7.5, 150 mM NaCl, 2 mM EDTA, 2 mm dithiothreitol, 0.8%

Triton X-100 and 1× protease inhibitor (Roche)) and incubated at 4 °C for 20 min. The lysate was centrifuged at 18,000g for 10 min at 4 °C, and the supernatant was centrifuged one more time. The supernatant was incubated with 30 μl anti-FLAG M2 magnetic beads (Sigma) for 2 h at 4 °C. The beads were washed with IP buffer five times, and proteins were eluted with 40 μl elution buffer (IP buffer containing 100 μg ml$^{-1}$ 3×FLAG peptide as final concentration) by vigorously shaking at 37 °C for 15 min. The elutions were mixed with 2× SDS loading buffer for western blot. Anti-Myc/c-Myc antibody (9E10) HRP (Santa Cruz Biotechnology sc-40 HRP, 1:3,000 dilution) and monoclonal ANTI-FLAG M2 HRP (Sigma-Aldrich A8592, 1:7,500 dilution) were used for western blot.

### Small RNA-seq
Pistils and anther (stage 9 or younger) of each genotype were collected and ground into a fine powder with liquid nitrogen. Total RNA was extracted using the Direct-zol RNA Miniprep kit (Zymo) according to the manufacturer's instructions. Two micrograms of total RNA were mixed with equal volume of the 2× RNA loading dye[43], denatured at 65 °C for 10 min and immediately chilled on ice. Denatured total RNA was separated on 15% TBE urea gel (Invitrogen), and small RNAs from 15 to 30 nucleotides were excised. The excised gel pieces were mashed by pestles. Small RNA was eluted with 400 μl nuclease-free water at 70 °C for 10 min and then precipitated with ethanol. Small RNA libraries were made using NEBNext Small RNA Library Prep Set for Illumina (Multiplex Compatible)[43] according to the manufacturer's instructions. The libraries were sequenced on Illumina NovaSeq 6000 or NovaSeq X Plus instruments.

### RNA-seq
Seedlings, pistils and pollens of Col-0 were collected and ground into a fine powder with liquid nitrogen. Total RNA was extracted using the Direct-zol RNA Miniprep kit (Zymo) according to the manufacturer's instructions. One microgram of total RNA was used to prepare the libraries for RNA sequencing (RNA-seq) following the TruSeq Stranded mRNA kit (Illumina), and the libraries were sequenced on Illumina NovaSeq 6000 or NovaSeq X Plus instruments.

### Whole-genome bisulfite sequencing
DNA from pistils was extracted with Qiagen DNeasy plant mini kit (Qiagen 69106). RNA was removed with PureLink RNase A (Invitrogen). A total of 100 ng of DNA was sheared to 200 bp (duty cycle 10%, intensity 5, cycles per burst 200, treatment time 120 s) with a Covaris S2 (Covaris). Libraries were prepared with the Epitect Bisulfite Conversion kit (Qiagen) and the Ovation Ultralow Methyl-seq kit (NuGEN) following the manufacturer's instructions. The libraries were sequenced on Illumina NovaSeq 6000 or NovaSeq X Plus instruments.

### DAP-seq
The DAP-seq experiment was performed as previously[46]. First, for genomic DNA library preparation, 5 μg of genomic DNA was purified with the DNeasy Plant Mini Kit (Qiagen, 69106) from Col-0 unopened flower buds and the DNA was diluted with water into 55-μl aliquots each containing 1,000 μg of DNA and sheared with the Covaris S2 instrument to an average fragment size of 200 bp (duty cycle 10%, intensity 5, cycles per burst 200, treatment time 120 s). Fifty microlitres of sheared DNA underwent steps 1–3 of library construction using the KAPA Hyper Prep kit (KR0961) (end repair and A-tailing, adapter ligation and post-ligation cleanup). Second, 1 μg of *pIX-HALO-VDD*, *pIX-HALO-VAL* and *pIX-HALO-GDE1* was used to express their corresponding proteins in TNA SP6 Coupled Wheat Germ Extract system (Promega). Magnetic HALO beads were used to enrich proteins and inoculated with genomic DNA library at room temperature for 1 hr. The final PCR amplification of the pulled-down DNA was performed using Fusion High-Fidelity PCR Master Mix. Each sample was individually run on agarose gel, and the smear between 200 bp and 400 bp was cut out

and purified. The libraries were sequenced on Illumina NovaSeq 6000 or NovaSeq X Plus instruments.

### Alphafold3 structure prediction
The three-dimensional models of VDD-VAL-GDE1-motif 1, REM12-REM13-GDE1-motif 1, VDD-REM13-GDE1-motif 1, VDD-REM12-GDE1-motif 1, VAL-REM13-GDE1-motif 1, VAL-REM12-GDE1-motif 1, VDD-VDD-GDE1-motif 1, REM13-REM13-GDE1-motif 1, REM12-REM12-GDE1-motif 1 and VAL-VAL-GDE1-motif 1 were generated by Alphafold3[43] and visualized and analysed by PyMOL 3.0.2.

### Bioinformatic analysis
For ChIP-seq and DAP-seq analysis, raw reads were aligned to the *Arabidopsis* reference genome (TAIR10) with Bowtie2 (v2.3.4.3)[55], allowing only uniquely mapped reads with perfect matches. The Samtools version 1.9 was used to remove duplicated reads[56]. The deeptools version 3.1.3 was used to generate Bigwig tracks[57]. Peaks were called using MACS2 (v2.1.1)[58].

For differential ChIP-seq localization analysis, ChIP-seq levels at the CLSY3 CLSY4-dependent siRNA regions were quantified with the HOMER (v4.11.1) annotatePeaks.pl script using the '-noadj, -size given and -len 1' options. Differentially expressed 24-nt siRNA compared with the WT controls were then identified using DESeq (version 1.42.1) (log$_2$FC ≥1 and FDR ≤0.05). FC, fold change. The data were plotted using the R package ggplot (v3.5.1).

For binding motif analysis, MEME 5.5.0 was used to discover the motifs of the ChIP-seq datasets[59]. FIMO (v5.5.7) was used to scan genome-wide distributions of CLSY3 CLSY4 motif 1 TTTTGCTTAT (single repeat) with one mismatch allowed, TTTTGCTTATNTTTTGCTTAT (double repeats) with one mismatch allowed in each repeat, TTTTGCT-TATNTTTTGCTTATNTTTTGCTTAT (triple repeats) with one mismatch allowed in each repeat, and TTTTGCTTATNNTTTTGCTTAT (double repeats with a two-nucleotide space) with one mismatch allowed in each repeat[60]; CLSY3 CLSY4 motif 2 AAGCGGATNAAGCGGATNAAGCG-GAT had a P value less than $5 × 10^{-9}$ and q value less than 0.025. TomTom (v5.5.7) was used to analyse the similarities between motifs[61].

For whole-genome bisulfite sequencing (WGBS) analysis, WGBS raw reads were aligned to both strands of reference genome TAIR10 using BSMAP (v.2.74)[62] allowing up to two mismatches and one best hit (-v 2 -w 1). Reads with more than three consecutively methylated CHH sites were considered as non-converted reads and removed. Methylation levels were calculated with the ratio of C/(C + T).

For RNA-seq analysis, Col-0 leaf, meiocyte and tapetum RNA seq data were downloaded from NCBI Gene Expression Omnibus (GEO) (https://www.ncbi.nlm.nih.gov/geo/) as accession codes GSM2306324; GSM2306325; GSM2306326 ref. 63, GSM2306313; GSM2306314; GSM2306315 ref. 63 and GSM4911399; GSM4911400; GSM4911401 ref. 10, respectively. All raw reads of RNA-seq data were aligned to reference genome TAIR10 by Bowtie2 (v2.3.4.3)[55], and expression abundance was calculated by RSEM (v1.3.1) with default settings[64]. The bamCoverage of deeptools (version 3.1.351) was used to normalize the data with RPKM (reads per kilobase per million mapped reads)[57].

For small RNA-seq analysis, the adaptor sequence (TGGAATTC TCGG) of small RNA-seq reads was trimmed with trim_galore, and trimmed reads were mapped to the reference genome TAIR10 using Bowtie2 (v2.3.4.3) with only one unique hit and zero mismatches[55]. Small RNA reads that mapped to chloroplast, mitochondrial DNA, tRNA, rRNA, small nucleolar RNAs and small nuclear RNAs were removed using bedtools (v2.26.0)[65]. The deeptools (version 3.1.3) was used to generate Bigwig tracks[57]. The bamCoverage of deeptools (version 3.1.3) was used to normalize the data with RPKM[57].

For differentially expressed 24-nt siRNA clusters analysis, Pol-IV-dependent master siRNA was defined from a previous publication[11]. The 24-nt siRNA levels at the master 24-nt siRNA were quantified with the HOMER (v4.11.1) annotatePeaks.pl script using the '-noadj, -size

given and -len 1' options. The 24-nt siRNA expression levels were normalized by the total miRNA amount, which was defined previously[66]. Comparison of the differentially expressed 24-nt siRNA with the WT controls was then performed using DESeq (version 1.42.1) (log$_2$FC ≤1 and FDR ≤0.05).

### Statistics and reproducibility

Pairwise *t*-tests were used for the siRNA and DNA methylation levels analysis. The two-sided empirical Bayes test performed by LIMMA was used for statistical analysis on IP–MS. Three biological replicates were included for all siRNA analyses, whereas two biological replicates were included for DNA methylation analysis and IP–MS analysis. No data were excluded from all analysis. The western blot experiments shown in Extended Data Fig. 2b were independently repeated three times with similar results.

### Reporting summary

Further information on research design is available in the Nature Portfolio Reporting Summary linked to this article.

### Data availability

All the high-throughput sequencing data generated in this study are accessible at NCBI's GEO via GEO series accession number GSE269181. The mass spectrometry proteomics data generated in this study have been deposited in the ProteomeXchange Consortium via the MassIVE partner repository under accession code MSV000097625. The TAIR10 genome is available at https://www.arabidopsis.org/index.jsp. The Col-0 leaf, meiocyte and tapetum RNA seq data used in this study are available via the National Center for Biotechnology information Gene Expression Omnibus database under accession codes GSM2306324; GSM2306325; GSM2306326, GSM2306313; GSM2306314; GSM2306315 and GSM4911399; GSM4911400; GSM4911401, respectively. All custom scripts are available from the corresponding author upon request. Source data are provided with this paper.

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

### Acknowledgements

We thank J. Law and G. Xu at Salk Institute for discussion and sharing the *clsy* mutants; J. Du at Southern University of Science and Technology for discussion of the Alphafold3 results; C. Picard, M. Wang and Y. He for advice on the manuscript; and J. Deng, Y. Yamamoto, J. Um and Z. Guo for technical support. We also thank M. Akhavan and the UCLA BSCRC BioSequencing Core for sequencing support. This work was supported by NIH grant R35 GM130272 to S.E.J. and a Life Sciences Research Foundation Postdoctoral Fellowship and the Hundred Talents Program at Zhejiang University to Z.W. S.E.J. is an Investigator of the Howard Hughes Medical Institute.

### Author contributions

Z.W. and S.E.J. designed the research, interpreted data and wrote the manuscript; Z.W. performed most of the experiments and bioinformatic data analysis; Y.X. initiated the project and generated GDE1-tagged transgenic lines; S.W. performed siRNA differentially expression analysis and transcription factor binding analysis; Y.-H.S., K.W. and C.A.H. generated transcription factors tagged lines and performed ChIP-seq; Z.Z. performed WGBS bioinformatic data analysis and initial bioinformatic data analysis of small RNA-seq, RNA-seq and ChIP-seq; S.F. performed high-throughput sequencing; Z.W., J.S., L.L. and J.W. performed IP–MS and interpreted the data. J.D., A.L. and C.A.H. provided technical help.

### Competing interests

The authors declare no competing interests.

### Additional information

**Extended data** is available for this paper at https://doi.org/10.1038/s41556-025-01691-0.

**Correspondence and requests for materials** should be addressed to Steven E. Jacobsen.

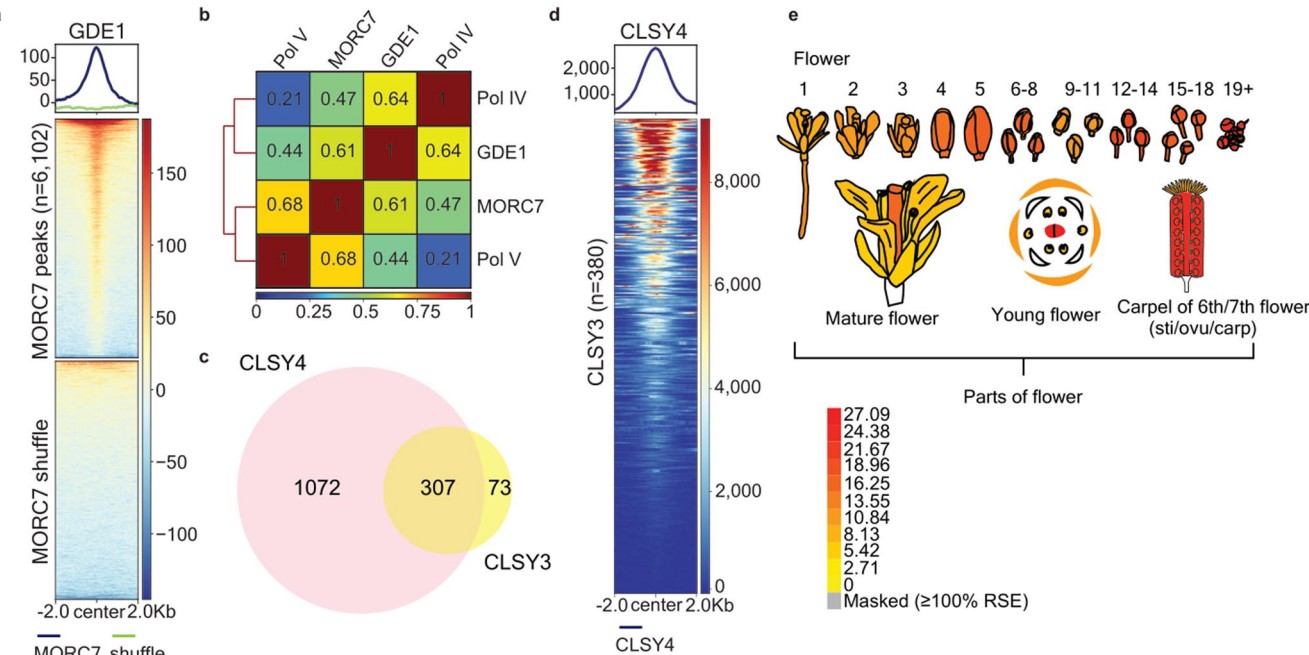

**Extended Data Fig. 1 | GDE1 is a previously uncharacterized protein localizing to RdDM sites. a**, Metaplot and heatmap showing enrichment of GDE1 ChIP-seq signal over MORC7 (n = 6,102). **b**, Genome-wide Spearman correlation analysis between GDE1 ChIP-seq signals and those of MORC7, Pol IV, and Pol V at co-targeted regions. Correlations were calculated in deepTools (the multiBamSummary was followed with plotCorrelation tools). **c**, Venn diagram showing the relationship between CLSY3 and CLSY4 binding sites. **d**, Metaplot and heatmap showing enrichment of CLSY4 ChIP-seq signal over CLSY3 (n = 380). **e**, Relative expression levels of the GDE1 genes in select tissues from ePlant expression viewers.

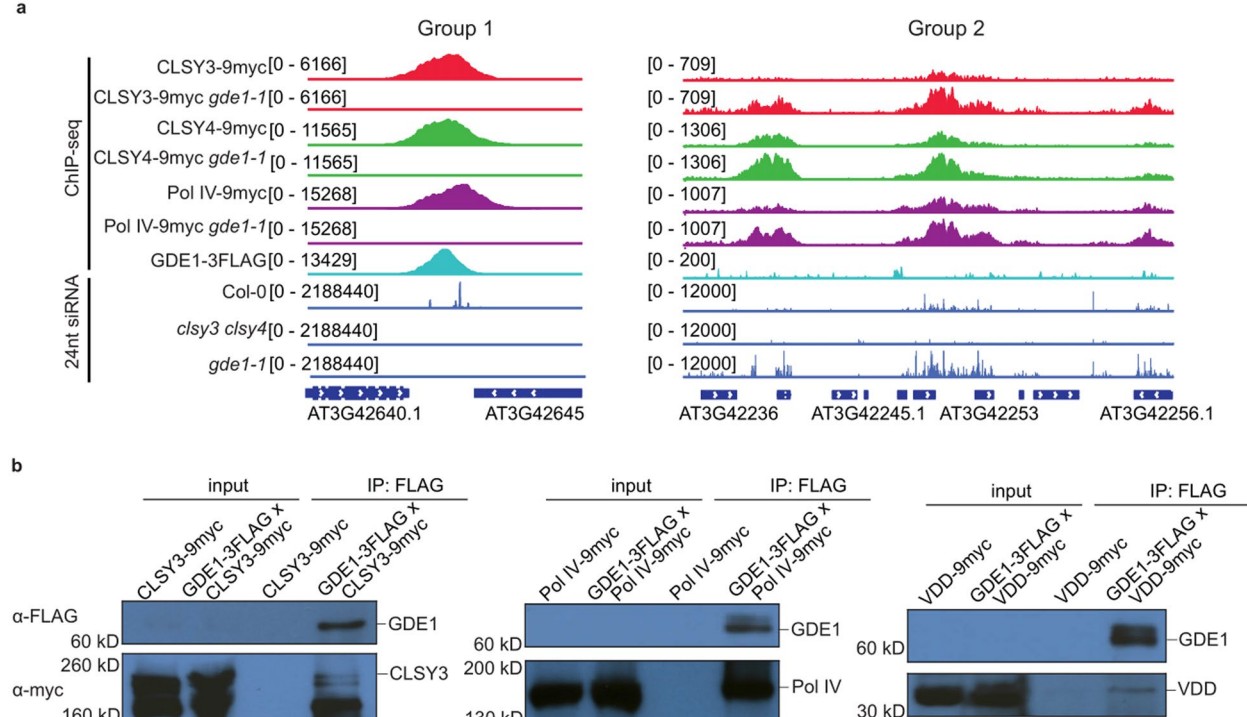

**Extended Data Fig. 2 | GDE1 co-localizes and interacts with Pol IV complex for siRNA production. a**, Screenshot of CLSY3-9myc, CLSY3-9myc *gde1-1*, CLSY4-9myc, CLSY4-9myc *gde1-1*, Pol IV-9myc, Pol IV-9myc *gde1-1* and GDE1-3FLAG ChIP-seq signals with control ChIP-seq signals subtracted over a Group 1 (CLSY3 CLSY4-dependent, GDE1-dependent) representative locus (left) or three Group 2 (CLSY3 CLSY4-dependent, GDE1-upregulated) representative loci (right). Square brackets indicate the range on a bar graph. **b**, Western blot showing a Co-IP assay in GDE1-3FLAG F2 crossed lines with CLSY3-9myc (left), Pol IV-9myc (middle) or VDD-9myc (right), respectively. Similar results were observed in two biological replicates. Source numerical data and unprocessed blots are available in source data.

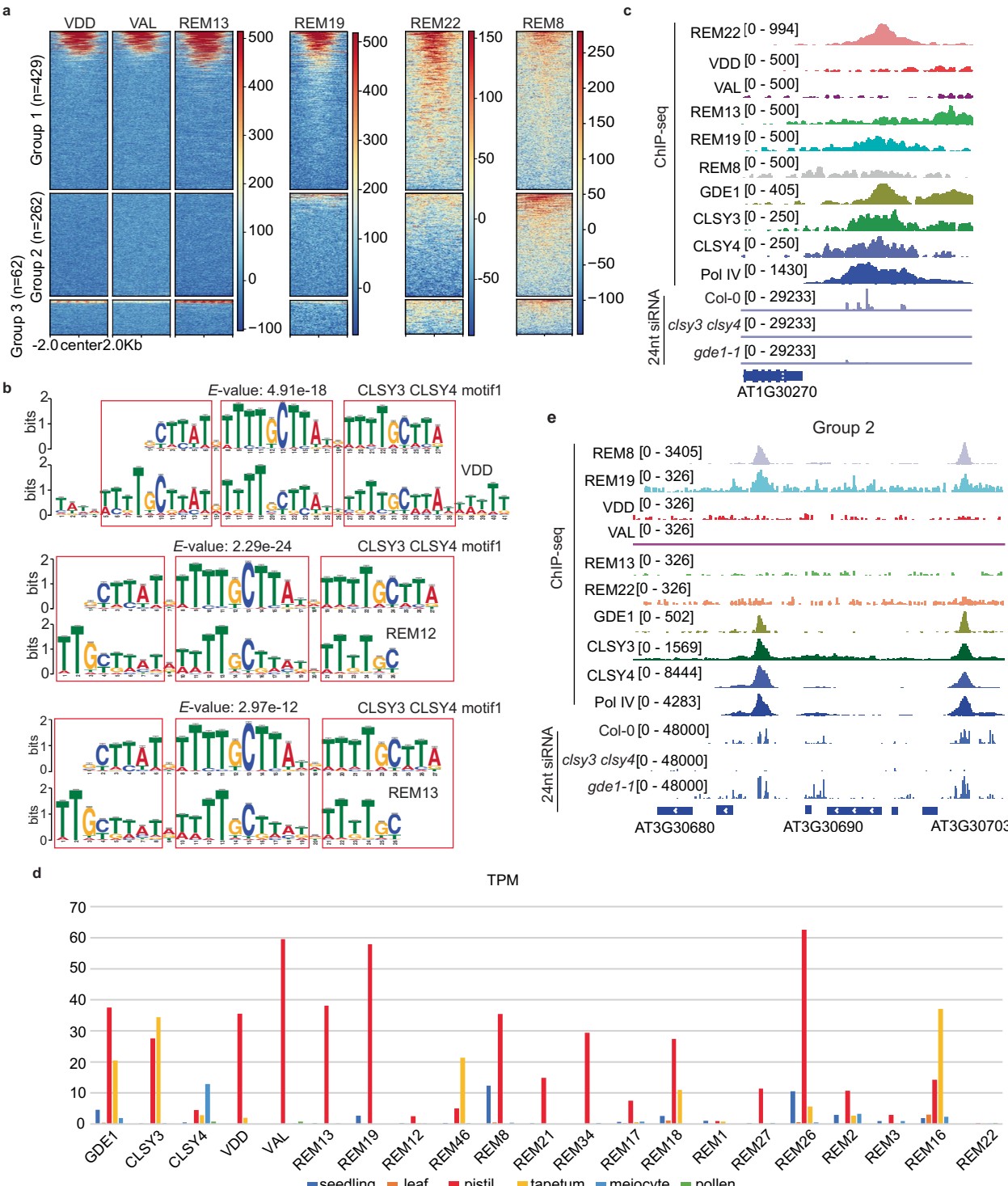

**Extended Data Fig. 3 | Differential localization of REM transcription factors at CLSY3 CLSY4-dependent siRNA sites. a**, Heatmaps showing enrichment of VDD, VAL, REM13, REM19, REM22 and REM8 ChIP-seq signal over three groups of CLSY3 CLSY4-dependent sites. The region is −2Kb to 2Kb from the center. Same regions were used for all metaplots. **b**, TOMTOM analysis showing the similarities of VDD, REM12 and REM13 binding sites with CLSY3 CLSY4 motif 1. *E* value is the expected number of false positives in the matches up to this point. **c**, Screenshot of REM22, VDD, VAL, REM13, REM19, REM8, GDE1, CLSY3, CLSY4 and Pol IV ChIP-seq at a distinct representative Group 1 of CLSY3 CLSY4-dependent siRNA sites, where REM22, but not other REM proteins exhibit enrichment. Square brackets indicate the range on a bar graph. **d**, RNA-seq expression analysis of REM transcription factors, GDE1, CLSY3 and CLSY4 in different tissues. **e**, Screenshot of REM8, REM19, VDD, VAL, REM13, REM22, GDE1, CLSY3, CLSY4 and Pol IV ChIP-seq at two representative Group 2 of CLSY3 CLSY4-dependent siRNA loci. Square brackets indicate the range on a bar graph.

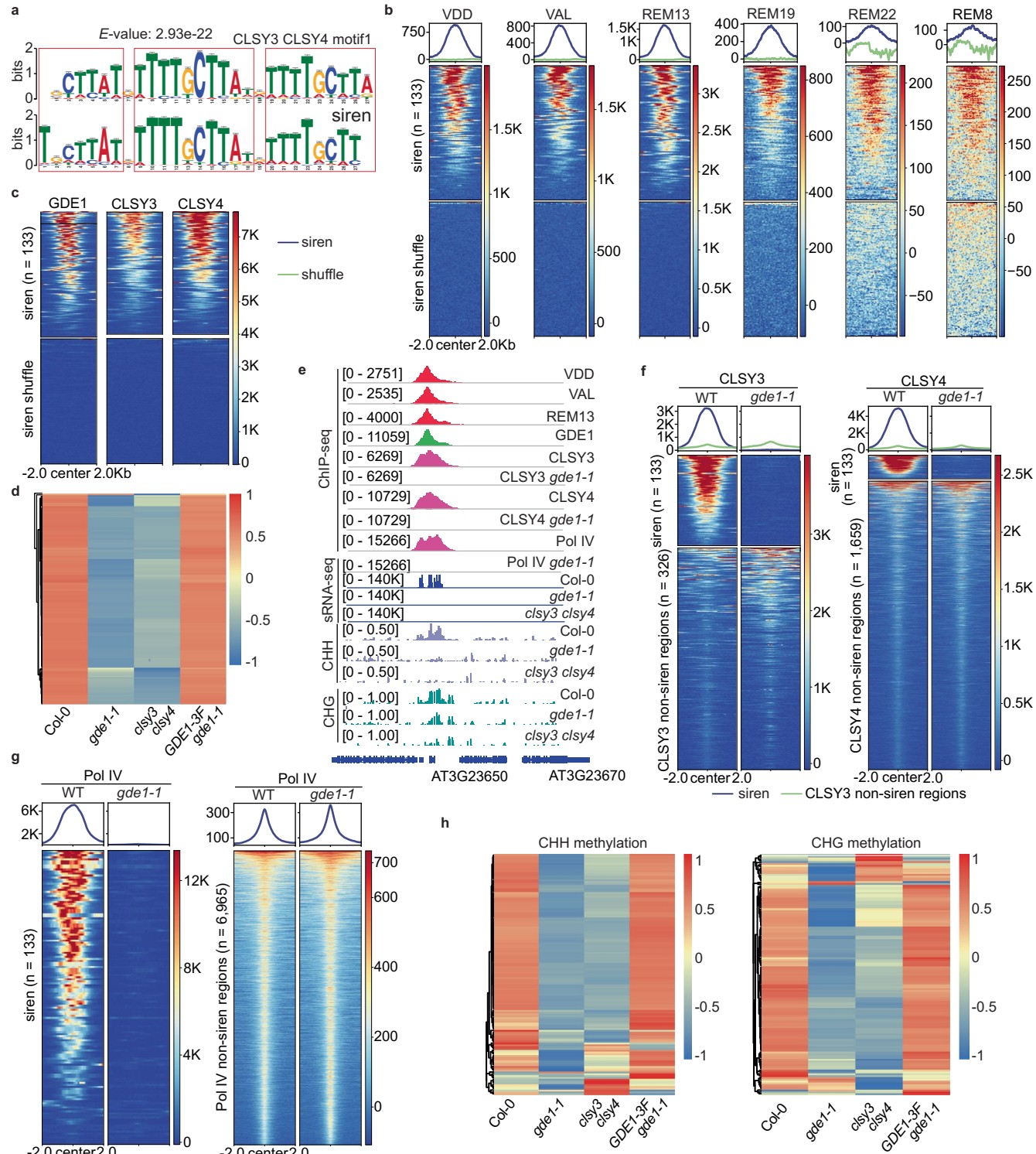

**Extended Data Fig. 4 | Transcription factors-driven Pol IV transcription complex localizes to siren loci for siRNA production and DNA methylation. a**, TOMTOM analysis showing the similarities of siren loci and CLSY3 CLSY4 motif 1. *E* value is the expected number of false positives in the matches up to this point. **b**, Metaplot and heatmap showing enrichment of VDD, VAL, REM13, REM19, REM22 and REM8 ChIP-seq signal over siren loci (n = 133). **c**, Heatmap showing enrichment of GDE1, CLSY3 and CLSY4 ChIP-seq signal over siren loci (n = 133). The region is −2Kb to 2Kb from the center. **d**, Heatmap showing the 24nt-siRNA levels at siren loci (n = 133) in the indicated genotypes. **e**, Screenshots of VDD, VAL, REM13, GDE1, CLSY3, CLSY3 *gde1-1*, CLSY4, CLSY4 *gde1-1*, Pol IV and Pol IV

*gde1-1* ChIP-seq signals with control ChIP-seq signals subtracted and 24nt-siRNA by sRNA-seq and CHG/CHH DNA methylation level by WGBS over a representative siren site in the indicated genotypes. Square brackets indicate the range on a bar graph. **f**, Metaplot and heatmap showing enrichment of CLSY3/4 ChIP-seq signal over CLSY3/4-bound siren loci (n = 133) and CLSY3/4 non-siren regions in WT and *gde1-1* backgrounds. The region is −2Kb to 2Kb from the center. **g**, Metaplot and heatmap showing enrichment of Pol IV ChIP-seq signal over Pol IV-bound siren loci (n = 133) and Pol IV non-siren regions in WT and *gde1-1* backgrounds. The region is −2Kb to 2Kb from the center. **h**, Heatmap showing the CHH and CHG methylation levels at siren loci (n = 133) in the indicated genotypes.

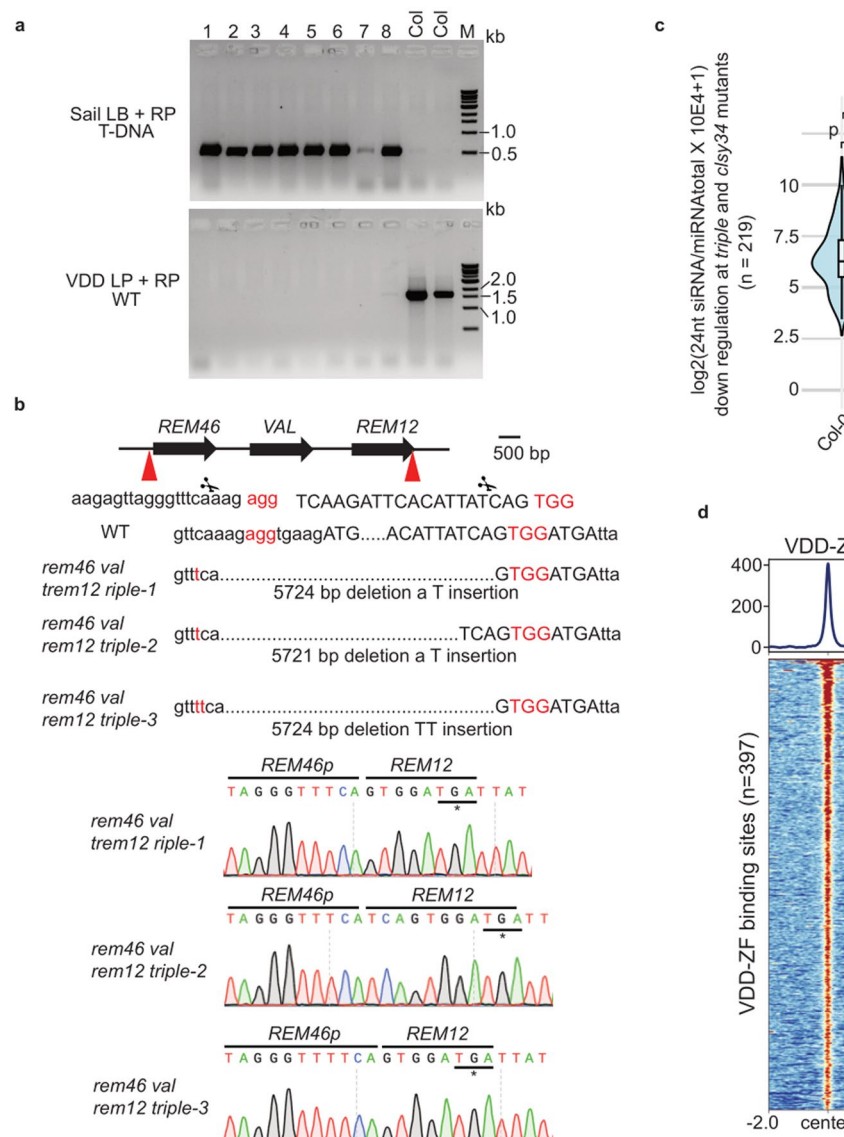

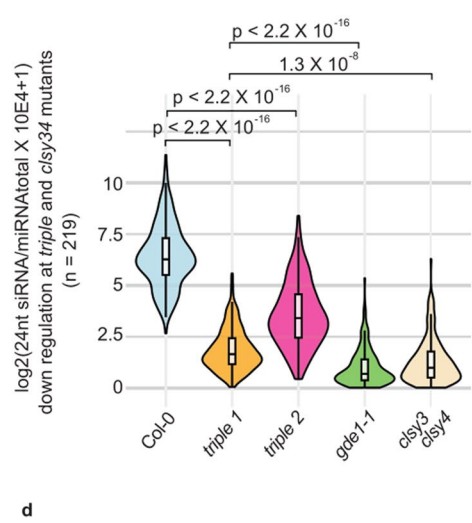

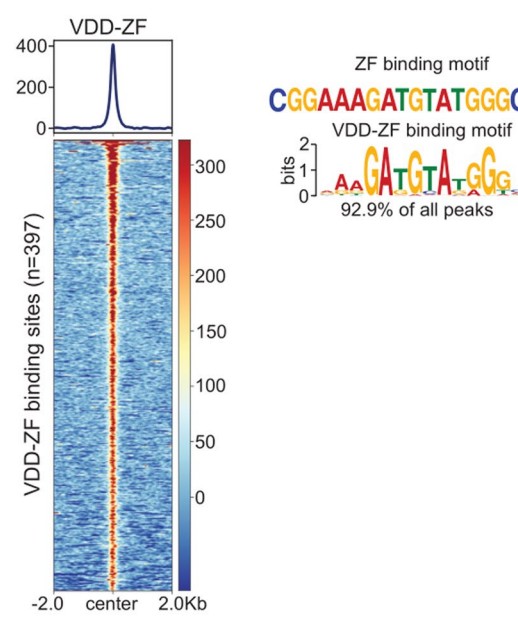

**Extended Data Fig. 5 | REM transcription factors are required for siRNA production at CLSY3 CLSY4-dependent siRNA loci. a**, Genotyping results from the T2 generation confirm the identification of *vdd* null mutants following the transformation of *VDD-9myc* into heterozygous *vdd* knockout mutants. Among the T2 generation, five T1 plants with homozygous *vdd* mutations were validated, all of which exhibit *vdd* null mutations. **b**, Schematic description of CRISPR/ Cas9 construct design for knocking out the tandemly duplicated REM46, VAL, and REM12. Deletion detected in genomic DNA of triple mutants are shown with chromatographs from Sanger sequencing. **c**, Violin plot showing the 24nt-siRNA levels at siRNA down regulation loci at both triple and *clsy3 clsy4* mutants

(n = 219) in the indicated genotypes. *p* values calculated by pairwise t-tests are indicated. Pairwise t-tests showed *p* = 1.0995E-126 (colov vs *rem46 val rem12 triple 1* ov), *p* = 1.82024E-60 (colov vs *rem46 val rem12 triple 2* ov) and *p* = 9.05326E-20 (*gde1-1*ov vs *rem46 val rem12 triple 1 ov*). The line in the center of each violin plot represents the median. The thick black bar in the center represents the interquartile range. The whiskers represent the rest of the distribution. **d**, Metaplot and heatmap showing enrichment of VDD-ZF ChIP-seq signal over ZF fusion targeted sites (n = 397) and motif identified by MEME motif analysis for VDD-ZF. Source numerical data are available in source data.

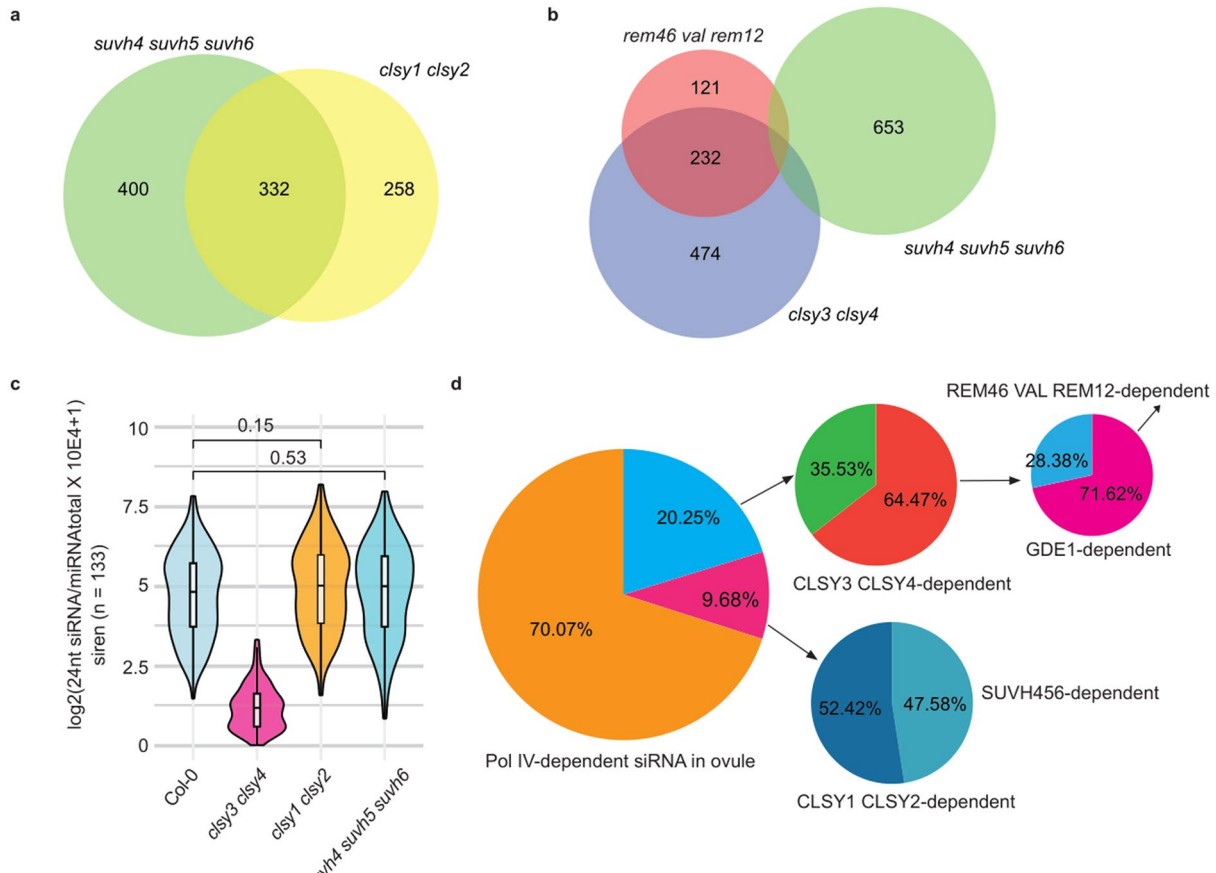

**Extended Data Fig. 6 | REM transcription factors-initiated siRNA biogenesis is independent of H3K9 methylation. a**, Venn diagrams showing the relationships between reduced 24nt-siRNA clusters in *suvh4 suvh5 suvh6* triple and the *clsy1 clsy2* double mutants. **b**, Venn diagrams showing the relationships between reduced 24nt-siRNA clusters in *suvh4 suvh5 suvh6* triple, *clsy3 clsy4* double and the *rem46 val rem12* triple mutants. **c**, Violin plot showing the 24nt-siRNA levels at siren loci (n = 133) in the indicated genotypes. *p* values calculated by pairwise t-tests are indicated. The line in the center of each violin plot represents the median. The thick black bar in the center represents the interquartile range. The whiskers represent the rest of the distribution. **d**, Pie charts showing the proportions of 24nt-siRNAs from all Pol IV-dependent clusters in ovule tissue that are reduced in *clsy1 clsy2* or *clsy3 clsy4* double mutants and the proportions of REM46 VAL REM12-dependent loci in GDE-dependent loci, GDE1-dependent siRNA among CLSY3 CLSY4-dependent group or SUVH456-dependent siRNA among CLSY1 CLSY2-dependent group. Source numerical data are available in source data.

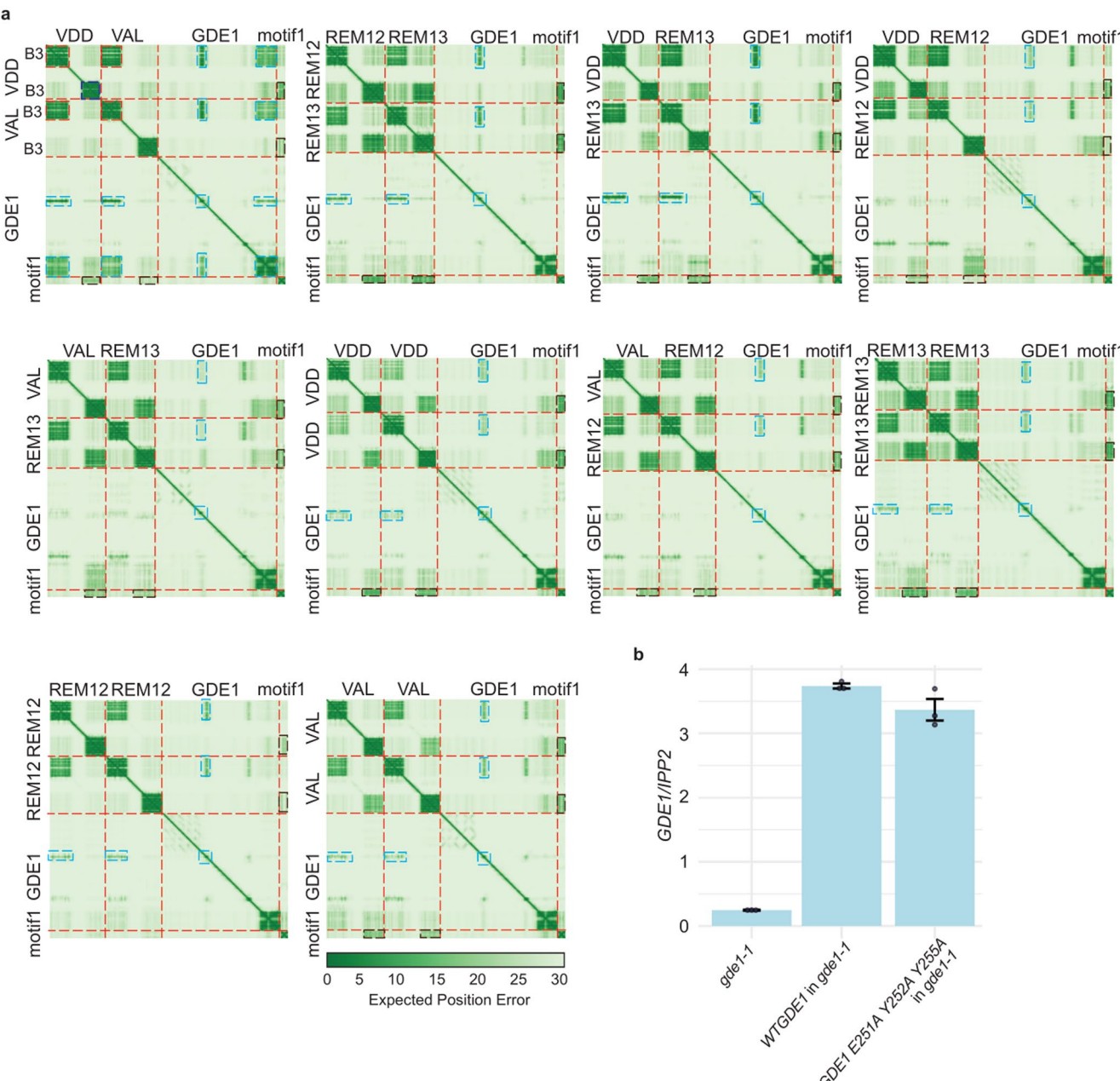

**Extended Data Fig. 7 | AlphaFold3 predicted interactions within the REM transcription factor–GDE1– CLSY3 CLSY4 motif 1 complex. a**, Predicted aligned error (PAE) of VDD-VAL-GDE1-motif 1, REM12-REM13-GDE1-motif 1, VDD-REM13-GDE1-motif 1, VDD-REM12-GDE1-motif 1, VAL-REM13-GDE1-motif 1, VDD-VDD-GDE1-motif 1, VAL-REM12-GDE1-motif 1, REM13-REM13-GDE1-motif 1, REM12-REM12-GDE1-motif 1, VAL-VAL-GDE1-motif 1. **b**, *GDE1* gene expression in ovule tissues of indicated genotypes as determined by qRT-PCR. Error bars represent means ± standard deviation (sd) (n = 3 from technical replicates). Individual data dots represent technical replicates. Three biological replicates were performed with similar results. Source numerical data are available in source data.

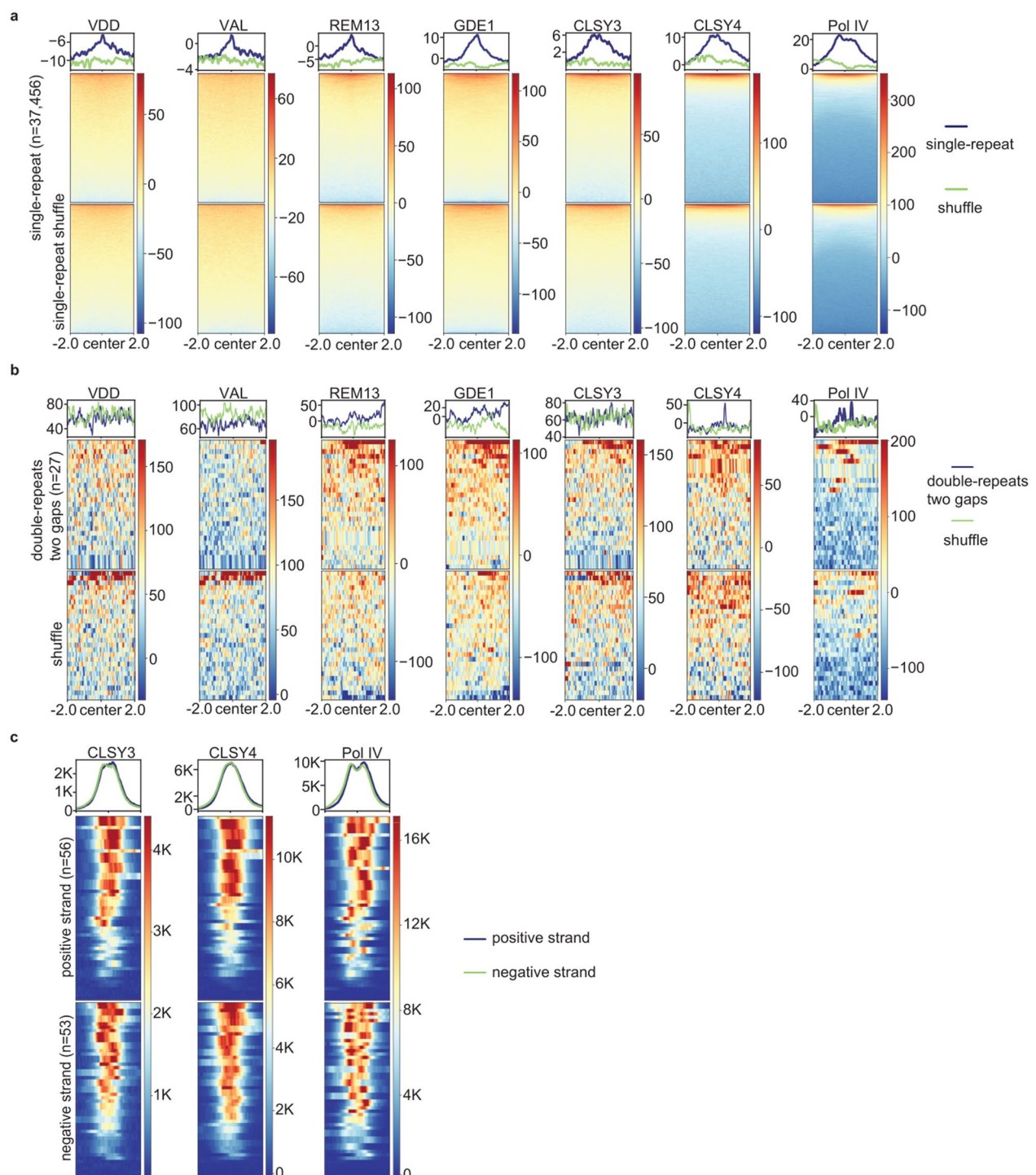

**Extended Data Fig. 8 | Pol IV transcription complex recognizes CLSY3 CLSY4 motif 1. a**, Metaplot and heatmap showing enrichment of VDD, VAL, REM13, GDE1, CLSY3, CLSY4 and Pol IV ChIP-seq signal over single-repeat sites (n = 37,456). **b**, Metaplot and heatmap showing enrichment of VDD, VAL, REM13, GDE1, CLSY3, CLSY4 and Pol IV ChIP-seq signal over double-repeats with two gaps (n = 27). **c**, Metaplot and heatmap showing enrichment of CLSY3, CLSY4 and Pol IV ChIP-seq signal over all double-repeats sites at positive strand (n = 56) or negative strand (n = 53).

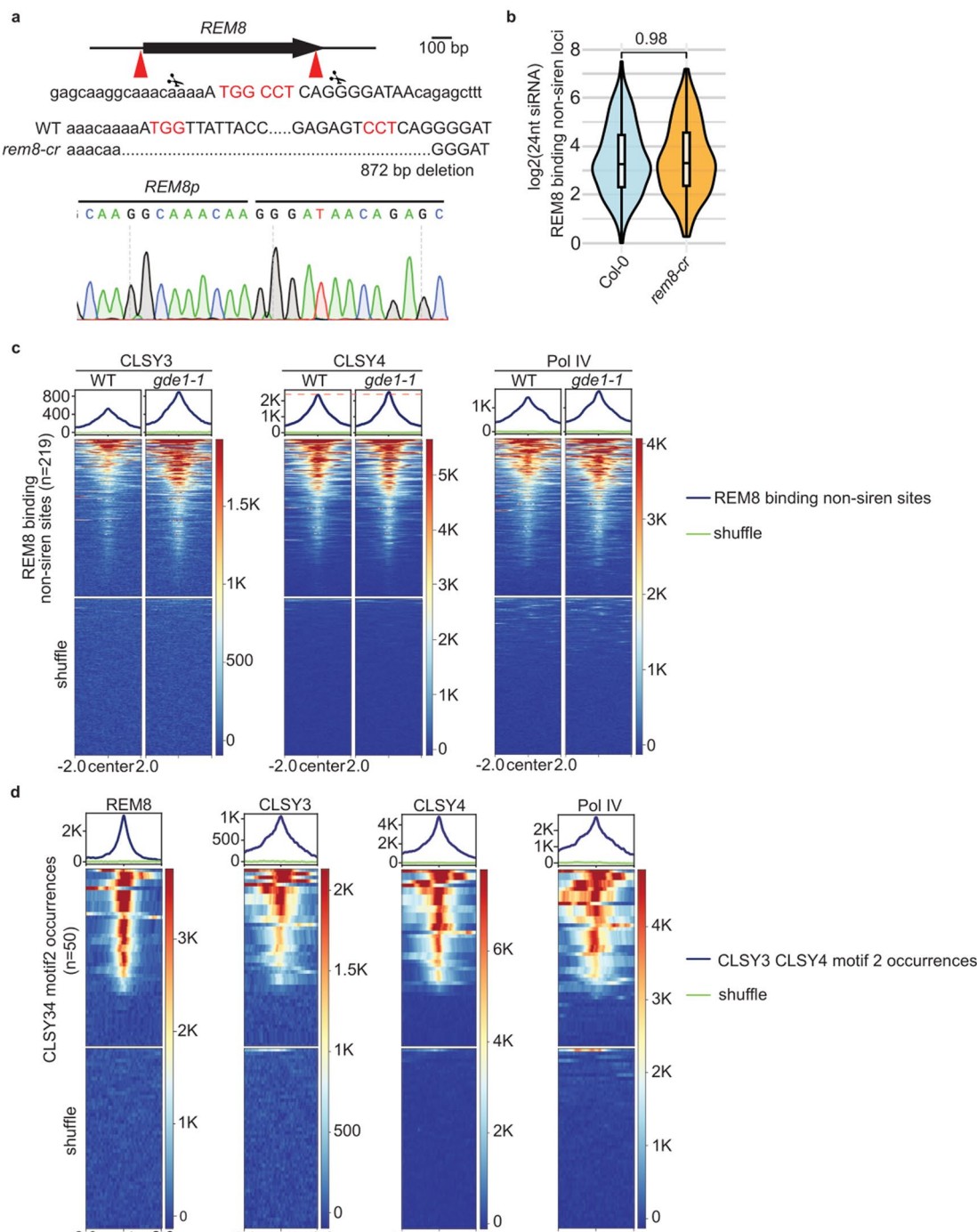

**Extended Data Fig. 9 | Pol IV transcription complex recognizes CLSY3 CLSY4 motif 2. a**, Schematic description of CRISPR/Cas9 construct design for knocking out REM8. Deletion detected in genomic DNA of *rem8-cr* mutants are shown with chromatographs from Sanger sequencing. **b**, Violin plot showing the 24nt-siRNA levels at REM8 binding non-siren loci (n = 162) in the indicated genotypes. *p* values calculated by pairwise t-tests are indicated. The line in the center of each violin plot represents the median. The thick black bar in the center represents the interquartile range. The whiskers represent the rest of the distribution. **c**, Metaplot and heatmap showing enrichment of CLSY3, CLSY4 and Pol IV ChIP-seq signal over REM8 binding non-siren sites and shuffle regions in WT or *gde1-1* backgrounds. The region is −2Kb to 2Kb from the center. **d**, Metaplot and heatmap showing enrichment of REM8, CLSY3, CLSY4, and Pol IV ChIP-seq signal over CLSY3 CLSY4 motif 2 sites (n = 50). Source numerical data are available in source data.

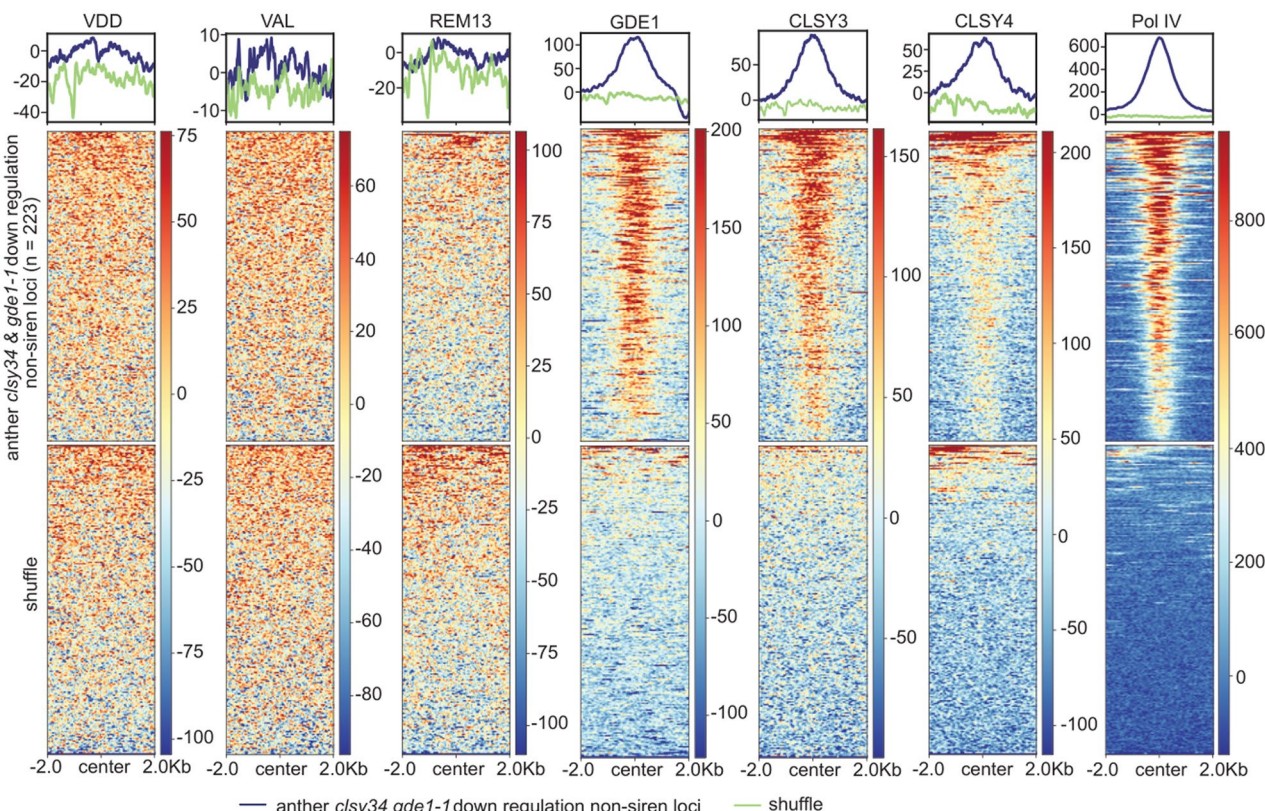

**Extended Data Fig. 10 | GDE1, CLSY3/4 and Pol IV, but not REM transcription factors enriches at *clsy3 clsy4* and *gde1-1* down-regulated siRNA loci in anther.**
Metaplot and heatmap showing enrichment of VDD, VAL, REM13, GDE1, CLSY3, CLSY4 and Pol IV ChIP-seq signal over anther *clsy3 clsy4* and *gde1-1* down regulation non-siren loci (n = 223).

# Reporting Summary

## Statistics

For all statistical analyses, confirm that the following items are present in the figure legend, table legend, main text, or Methods section.

| n/a | Confirmed | |
|---|---|---|
| ☐ | ☒ | The exact sample size (*n*) for each experimental group/condition, given as a discrete number and unit of measurement |
| ☐ | ☒ | A statement on whether measurements were taken from distinct samples or whether the same sample was measured repeatedly |
| ☐ | ☒ | The statistical test(s) used AND whether they are one- or two-sided *Only common tests should be described solely by name; describe more complex techniques in the Methods section.* |
| ☒ | ☐ | A description of all covariates tested |
| ☒ | ☐ | A description of any assumptions or corrections, such as tests of normality and adjustment for multiple comparisons |
| ☐ | ☒ | A full description of the statistical parameters including central tendency (e.g. means) or other basic estimates (e.g. regression coefficient) AND variation (e.g. standard deviation) or associated estimates of uncertainty (e.g. confidence intervals) |
| ☐ | ☒ | For null hypothesis testing, the test statistic (e.g. *F*, *t*, *r*) with confidence intervals, effect sizes, degrees of freedom and *P* value noted *Give P values as exact values whenever suitable.* |
| ☒ | ☐ | For Bayesian analysis, information on the choice of priors and Markov chain Monte Carlo settings |
| ☒ | ☐ | For hierarchical and complex designs, identification of the appropriate level for tests and full reporting of outcomes |
| ☒ | ☐ | Estimates of effect sizes (e.g. Cohen's *d*, Pearson's *r*), indicating how they were calculated |

*Our web collection on statistics for biologists contains articles on many of the points above.*

## Software and code

Policy information about availability of computer code

| Data collection | No software was used for data collection. |
|---|---|
| Data analysis | ChIP-seq analysis and DAP-seq analysis: Raw reads were aligned to the Arabidopsis reference genome (TAIR10) with Bowtie2 (v2.3.4.3), allowing only uniquely mapped reads with perfect matches. The Samtools version 1.9 was used to remove duplicated reads. The deeptools version 3.1.3 was used to generate Bigwig tracks. Peaks were called using MACS2 (v2.1.1). |
| | Differentially ChIP-seq localization analysis: ChIP-seq levels at the clsy3 clsy4-dependent siRNA regions were quantified with the HOMER (v4.11.1) annotatePeaks.pl script using the "-noadj, -size given and -len 1" options. Differentially expressed 24nt-siRNA compared to the WT controls were then identified using DESeq (version 1.42.1)(log2 FC≥1 and FDR≤0.05). The data were plotted using the R package ggplot (v3.5.1). |
| | Binding motif analysis: MEME 5.5.0 was used to discover the motifs of the ChIP-seq data sets. FIMO (v5.5.7) was used to scan genome-wide distributions of clsy3 clsy4 motif1 TTTTGCTTAT (single-repeat) with one mismatch allowed, TTTTGCTTATNTTTTGCTTAT (double-repeats) with one mismatch allowed in each repeat, TTTTGCTTATNTTTTGCTTATNTTTTGCTTAT (triple-repeats) with one mismatch allowed in each repeat, and TTTTGCTTATNNTTTTGCTTAT (double-repeats with a two-nucleotide space) with one mismatch allowed in each repeat; clsy3 clsy4 motif2 AAGCGGATNAAGCGGATNAAGCGGAT with p-value less than 5E-09 and q-value less than 0.025. Tomtom (v5.5.7) was used to analyze the similarities between motifs. |
| | WGBS analysis: WGBS raw reads were aligned to both strands of reference genome TAIR10 using BSMAP (v.2.74) with allowing up to 2 mismatches and 1 best hit (-v 2 -w 1). Reads with more than 3 consecutively methylated CHH sites were considered as non-converted reads and removed. Methylation levels were calculated with the ratio of C/(C + T). |

RNA-seq analysis: Col-0 leaf, meiocyte and tapetum RNA seq data were downloaded from NCBI Gene Expression Omnibus (GEO) as accession GSM2306324; GSM2306325; GSM2306326, GSM2306313; GSM2306314; GSM2306315 and GSM4911399; GSM4911400; GSM4911401(13), respectively. All raw reads of RNA-seq data were aligned to reference genome TAIR10 by Bowtie2 (v2.3.4.3), and expression abundance was calculated by RSEM (v1.3.1) with default settings. The bamCoverage of deeptools version 3.1.3 was used to normalize the data with RPKM .

Small RNA-seq analysis: Adaptor sequence (TGGAATTCTCGG) of small RNA-seq reads were trimmed with trim_galore, and trimmed reads were mapped to the reference genome TAIR10 using Bowtie2 (v2.3.4.3) with only one unique hit and zero mismatches. sRNA reads that mapped to chloroplast, mitochondrial DNA, tRNA, rRNA, small nucleolar RNAs (snoRNAs), and small nuclear RNAs (snRNAs) were removed using bedtools (v2.26.0). The deeptools version 3.1.3 was used to generate Bigwig tracks. The bamCoverage of deeptools version 3.1.3 was used to normalize the data with RPKM.

Differentially expressed (DE) 24nt-siRNA clusters analysis: Pol IV dependent master siRNA were defined from a previous publication. 24nt-siRNA levels at the master 24nt-siRNA were quantified with the HOMER (v4.11.1) annotatePeaks.pl script using the "-noadj, -size given and -len 1" options. 24nt-siRNA expression levels were normalized by total miRNA amount, which were defined from previously. Differentially expressed 24nt-siRNA compared to the WT controls were then identified using DESeq (version 1.42.1)(log2 FC≤1 and FDR≤0.05).

Quantitative proteomics: Label-free quantification was performed using the MaxQuant software package (v1.6.17.0) with LFQ default setting54, and Arabidopsis TAIR 10 proteome database was used for the database search. Trypsin digestion was applied and a maximum of two missed cleavages were allowed in all searches for tryptic peptides of length 8–40 amino acids. In all, 1% false discovery rate was used as a filter at both protein and peptide-spectrum match (PSM) levels. IP-MS of Col-0 plant tissue was used as control. The empirical Bayes test performed by LIMMA was used for statistical analysis.

For manuscripts utilizing custom algorithms or software that are central to the research but not yet described in published literature, software must be made available to editors and reviewers. We strongly encourage code deposition in a community repository (e.g. GitHub). See the Nature Portfolio guidelines for submitting code & software for further information.

## Data

Policy information about availability of data

All manuscripts must include a data availability statement. This statement should provide the following information, where applicable:
- Accession codes, unique identifiers, or web links for publicly available datasets
- A description of any restrictions on data availability
- For clinical datasets or third party data, please ensure that the statement adheres to our policy

All the high-throughput sequencing data generated in this study is accessible at NCBI's Gene Expression Omnibus (GEO) via GEO Series accession number (GSE269181). The mass spectrometry proteomics data generated in this study have been deposited in the ProteomeXchange Consortium via the MassIVE partner repository under accession code MSV000097625. The TAIR10 genome is available at https://www.arabidopsis.org/index.jsp. The Col-0 leaf, meiocyte and tapetum RNA seq data used in this study are available in the National Center for Biotechnology information Gene Expression Omnibus database under accession code GSM2306324; GSM2306325; GSM2306326, GSM2306313; GSM2306314; GSM2306315 and GSM4911399; GSM4911400; GSM4911401, respectively. Source data are provided as a Source Data file.

## Research involving human participants, their data, or biological material

Policy information about studies with human participants or human data. See also policy information about sex, gender (identity/presentation), and sexual orientation and race, ethnicity and racism.

| Reporting on sex and gender | N.A. |
| Reporting on race, ethnicity, or other socially relevant groupings | N.A. |
| Population characteristics | N.A. |
| Recruitment | N.A. |
| Ethics oversight | N.A. |

Note that full information on the approval of the study protocol must also be provided in the manuscript.

# Field-specific reporting

Please select the one below that is the best fit for your research. If you are not sure, read the appropriate sections before making your selection.

☒ Life sciences  ☐ Behavioural & social sciences  ☐ Ecological, evolutionary & environmental sciences

For a reference copy of the document with all sections, see nature.com/documents/nr-reporting-summary-flat.pdf

# Life sciences study design

All studies must disclose on these points even when the disclosure is negative.

| | |
|---|---|
| Sample size | No sample size calculation was performed. Sample sizes are determined on experimental trials and a previous study (Wang et al., Nature Plants, 2023). Sample sizes of all experiments were large enough (e.g. ovule and anther from >=20 plants were collected for sRNA-seq; unopen buds from >=80 plants were harvested for ChIP-seq with two biological replicates and etc.) to reach statistical reproducibility and significance. |
| Data exclusions | No data exclusion in the study. |
| Replication | Two replicates for ChIP-seq and WGBS. Three replicates for sRNA-seq. All replicates were performed independently and produced high reproducible results. |
| Randomization | For all experiments, treatment and control samples were grown side by side. Allocation of samples were not random, because it is not relevant to the study. |
| Blinding | No blinding used because it was largely not relevant to our study. All data were collected based on the genotype of plants, while blinding the samples during the experiments will increases the risk of mislabeling and wrong results. |

# Reporting for specific materials, systems and methods

We require information from authors about some types of materials, experimental systems and methods used in many studies. Here, indicate whether each material, system or method listed is relevant to your study. If you are not sure if a list item applies to your research, read the appropriate section before selecting a response.

### Materials & experimental systems

| n/a | Involved in the study |
|---|---|
| ☐ | ☒ Antibodies |
| ☒ | ☐ Eukaryotic cell lines |
| ☒ | ☐ Palaeontology and archaeology |
| ☒ | ☐ Animals and other organisms |
| ☒ | ☐ Clinical data |
| ☒ | ☐ Dual use research of concern |
| ☐ | ☒ Plants |

### Methods

| n/a | Involved in the study |
|---|---|
| ☐ | ☒ ChIP-seq |
| ☒ | ☐ Flow cytometry |
| ☒ | ☐ MRI-based neuroimaging |

## Antibodies

| | |
|---|---|
| Antibodies used | Antibody for FLAG epitope (for ChIP-seq): M2 antibody, Sigma F1804, 10 ul per ChIP added at a final dilution of 1:400<br>Antibody for MYC epitope (for ChIP-seq): Cell Signaling, 71D10, 20 ul per ChIP added at a final dilution of 1:200<br>HRP conjugated antibody for FLAG epitope (for western blot): Sigma-Aldrich ANTI-FLAG M2-peroxidase A8592, 1:7500 dilution<br>HRP conjugated antibody for MYC epitope (for western blot): Santa Cruz Biotechnology Anti-Myc/c-Myc antibody 9E10 HRP (sc-40 HRP), 1:3000 dilution |
| Validation | Anti-FLAG M2 (Sigma): the antibodies have been validated by the manufacturer, https://www.sigmaaldrich.com/catalog/product/sigma/fl804<br>Anti-FLAG M2-Peroxidase (HRP)(Sigma): the antibodies have been validated by the manufacturer, https://www.sigmaaldrich.com/US/en/product/sigma/a8592<br>Anti-myc (Cell Signaling): the antibodies have been validated by the manufacturer, https://www.cellsignal.com/products/antibody-conjugates/myc-tag-71d10-rabbit-mab-hrp-conjugate/14038<br>HRP conjugated antibody for MYC epitope (Santa Cruz Biotechnology sc-40): the antibodies have been validated by the manufacturer,  https://www.scbt.com/p/c-myc-antibody-9e10?gclid=CjwKCAjwvJyjBhApEiwAWz2nLQpNcYGsOfC7x6jRDD1GtD1Y8eousO7TM84Gg9FKaaHq8gTwQEtgyhoCSMAQAvD_BwE |

# Dual use research of concern

Policy information about dual use research of concern

### Hazards

Could the accidental, deliberate or reckless misuse of agents or technologies generated in the work, or the application of information presented in the manuscript, pose a threat to:

| No | Yes | |
|----|-----|---|
| ☒ | ☐ | Public health |
| ☒ | ☐ | National security |
| ☒ | ☐ | Crops and/or livestock |
| ☒ | ☐ | Ecosystems |
| ☒ | ☐ | Any other significant area |

## Experiments of concern

Does the work involve any of these experiments of concern:

| No | Yes | |
|----|-----|---|
| ☒ | ☐ | Demonstrate how to render a vaccine ineffective |
| ☒ | ☐ | Confer resistance to therapeutically useful antibiotics or antiviral agents |
| ☒ | ☐ | Enhance the virulence of a pathogen or render a nonpathogen virulent |
| ☒ | ☐ | Increase transmissibility of a pathogen |
| ☒ | ☐ | Alter the host range of a pathogen |
| ☒ | ☐ | Enable evasion of diagnostic/detection modalities |
| ☒ | ☐ | Enable the weaponization of a biological agent or toxin |
| ☒ | ☐ | Any other potentially harmful combination of experiments and agents |

# Plants

| | |
|---|---|
| Seed stocks | Col-0 ecotype was obtained from SALK institute. T-DNA lines used in this study are listed as below: gde1-1 (SALKseq_10069.1), clsy3-1 (SALK_040366), and clsy4-1 (SALK_003876). All if from ABRC. |
| Novel plant genotypes | rem46 val rem12 triple mutants were generated using guides: aagagttagggtttcaaagagg and TCAAGATTCACATTATCAGTGG. Guides were cloned into pBEE401 vector and the vector was transformed into Col-0, dge1-1, and clsy3 clsy4 muants.<br>rem8-cr plants were generated using guides:tgagcaaggcaaacaaaaATGG and CCTCAGGGGATAAcagagcttt. Guides were cloned into pBEE401 vector and the vector was transformed into Col-0. |
| Authentication | T-DNA mutants were genotyped by PCR using the primers suggested by SALK (http://signal.salk.edu/tdnaprimers.2.html). CRISPR mutants were Sanger sequenced to confirm the mutations. |

# ChIP-seq

## Data deposition

☒ Confirm that both raw and final processed data have been deposited in a public database such as GEO.

☒ Confirm that you have deposited or provided access to graph files (e.g. BED files) for the called peaks.

| | |
|---|---|
| Data access links<br>*May remain private before publication.* | All the high-throughput sequencing data generated in this study is accessible at NCBI's Gene Expression Omnibus (GEO) via GEO Series accession number (GSE269181). |
| Files in database submission | ChIP-seq-Col-9myc-rep1.bw<br>ChIP-seq-Col-9myc-rep2.bw<br>ChIP-seq-CLSY3-9myc-rep1.bw<br>ChIP-seq-CLSY3-9myc-rep2.bw<br>ChIP-seq-CLSY4-9myc-rep1.bw<br>ChIP-seq-CLSY4-9myc-rep2.bw<br>ChIP-seq-CLSY3-9myc-gde1-rep1.bw<br>ChIP-seq-CLSY3-9myc-gde1-rep2.bw<br>ChIP-seq-CLSY3-9myc-rep1.narrowPeak<br>ChIP-seq-CLSY3-9myc-rep2.narrowPeak<br>ChIP-seq-CLSY4-9myc-rep1.narrowPeak<br>ChIP-seq-CLSY4-9myc-rep2.narrowPeak<br>ChIP-seq-CLSY3-9myc-gde1-rep1.narrowPeak<br>ChIP-seq-CLSY3-9myc-gde1-rep2.narrowPeak<br>ChIP-seq-CLSY4-9myc-gde1-rep1.narrowPeak<br>ChIP-seq-CLSY4-9myc-gde1-rep2.narrowPeak<br>ChIP-seq-PolIV-9myc_rep2.narrowPeak<br>ChIP-seq-PolIV-9myc-gde1-rep1.narrowPeak<br>ChIP-seq-PolIV-9myc-gde1-rep2.narrowPeak |

```
ChIP-seq-REM8-9myc_rep1.narrowPeak
ChIP-seq-REM8-9myc_rep2.narrowPeak
ChIP-seq-REM13-9myc-rep1.narrowPeak
ChIP-seq-REM13-9myc-rep2.narrowPeak
ChIP-seq-REM19-9myc-rep1.narrowPeak
ChIP-seq-REM19-9myc-rep2.narrowPeak
ChIP-seq-REM22-9myc-rep1.narrowPeak
ChIP-seq-REM22-9myc-rep2.narrowPeak
ChIP-seq-VAL-9myc-rep1.narrowPeak
ChIP-seq-VAL-9myc-rep2.narrowPeak
ChIP-seq-VDD-9myc-gde1-rep1.narrowPeak
ChIP-seq-VDD-9myc-gde1-rep2.narrowPeak
ChIP-seq-VDD-9myc-rep1.narrowPeak
ChIP-seq-VDD-9myc-rep2.narrowPeak
ChIP-seq-GDE1_3FLAG-rep1.narrowPeak
ChIP-seq-GDE1_3FLAG-rep2.narrowPeak
ChIP-seq-NPVDDZFcol-rep1.narrowPeak
ChIP-seq-CLSY4-9myc-gde1-rep1.bw
ChIP-seq-CLSY4-9myc-gde1-rep2.bw
ChIP-seq-PolIV-9myc_rep1.bw
ChIP-seq-PolIV-9myc_rep2.bw
ChIP-seq-PolIV-9myc-gde1-rep1.bw
ChIP-seq-PolIV-9myc-gde1-rep2.bw
ChIP-seq-REM8-9myc_rep1.bw
ChIP-seq-REM8-9myc_rep2.bw
ChIP-seq-REM13-9myc-rep1.bw
ChIP-seq-REM13-9myc-rep2.bw
ChIP-seq-REM19-9myc-rep1.bw
ChIP-seq-REM19-9myc-rep2.bw
ChIP-seq-REM22-9myc-rep1.bw
ChIP-seq-REM22-9myc-rep2.bw
ChIP-seq-VAL-9myc-rep1.bw
ChIP-seq-VAL-9myc-rep2.bw
ChIP-seq-VDD-9myc-gde1-rep1.bw
ChIP-seq-VDD-9myc-gde1-rep2.bw
ChIP-seq-VDD-9myc-rep1.bw
ChIP-seq-VDD-9myc-rep2.bw
ChIP-seq-Col_3FLAG-rep1.bw
ChIP-seq-Col_3FLAG-rep2.bw
ChIP-seq-GDE1_3FLAG-rep1.bw
ChIP-seq-GDE1_3FLAG-rep2.bw
ChIP-seq-NPVDDZFcol-rep1.bw
ChIP-seq-NPVDDZFcol-rep2.bw
ChIP-seq-VDD-3FLAG-rep1.bw
ChIP-seq-VDD-3FLAG-rep2.bw
ChIP-seq-VDD-3FLAG-rep1-clsy34.bw
ChIP-seq-VDD-3FLAG-rep2-clsy34.bw
```

| Genome browser session (e.g. UCSC) | Available at GEO |
|---|---|

# Methodology

| Replicates | 2 |
|---|---|

| Sequencing depth | |
|---|---|
| | ChIP-seq-CLSY3-9myc-gde1-rep1 38322895 22468079 50 PE |
| | ChIP-seq-CLSY3-9myc-gde1-rep2 29940352 17188455 50 PE |
| | ChIP-seq-CLSY3-9myc-rep1 41899996 26416716 50 PE |
| | ChIP-seq-CLSY3-9myc-rep2 29741279 17926402 50 PE |
| | ChIP-seq-CLSY4-9myc-gde1-rep1 39976730 28135934 50 PE |
| | ChIP-seq-CLSY4-9myc-gde1-rep2 24208015 16477456 50 PE |
| | ChIP-seq-CLSY4-9myc-rep1 37634222 24080063 50 PE |
| | ChIP-seq-CLSY4-9myc-rep2 21349871 12871954 50 PE |
| | ChIP-seq-Col-9myc-rep1 29647635 18137933 50 PE |
| | ChIP-seq-Col-9myc-rep2 27979739 18669883 50 PE |
| | ChIP-seq-Col_3FLAG-rep1 43742144 31594808 50 PE |
| | ChIP-seq-Col_3FLAG-rep2 36127929 20249874 50 PE |
| | ChIP-seq-GDE1_3FLAG-rep1 29607543 19960847 50 PE |
| | ChIP-seq-GDE1_3FLAG-rep2 33774325 18050851 50 PE |
| | ChIP-seq-NPVDDZFcol-rep1 64890680 52519488 50 PE |
| | ChIP-seq-NPVDDZFcol-rep2 5925518 4265145 50 PE |
| | ChIP-seq-PolIV-9myc-gde1-rep1 37298313 23347234 50 PE |
| | ChIP-seq-PolIV-9myc-gde1-rep2 20151237 11774044 50 PE |
| | ChIP-seq-PolIV-9myc_rep1 33060928 19661423 50 PE |
| | ChIP-seq-PolIV-9myc_rep2 19737587 11729317 50 PE |
| | ChIP-seq-REM13-9myc-rep1 22759340 12182140 50 PE |

ChIP-seq-REM13-9myc-rep2 21816313 11708062 50 PE
ChIP-seq-REM19-9myc-rep1 18032889 9739125 50 PE
ChIP-seq-REM19-9myc-rep2 16623085 8499778 50 PE
ChIP-seq-REM22-9myc-rep1 29047814 17315024 50 PE
ChIP-seq-REM22-9myc-rep2 45598950 35270637 50 PE
ChIP-seq-REM8-9myc_rep1 22709199 11551560 50 PE
ChIP-seq-REM8-9myc_rep2 33970098 13074494 50 PE
ChIP-seq-VAL-9myc-rep1 40593954 24757138 50 PE
ChIP-seq-VAL-9myc-rep2 27496193 17150030 50 PE
ChIP-seq-VDD-3FLAG-rep1-clsy34 81994167 69007097 50 PE
ChIP-seq-VDD-3FLAG-rep1 87821754 74713702 50 PE
ChIP-seq-VDD-3FLAG-rep2-clsy34 8820589 6304238 50 PE
ChIP-seq-VDD-3FLAG-rep2 8883799 6657747 50 PE
ChIP-seq-VDD-9myc-gde1-rep1 43128548 24416059 50 PE
ChIP-seq-VDD-9myc-gde1-rep2 28382558 16987237 50 PE
ChIP-seq-VDD-9myc-rep1 38404573 21434747 50 PE
ChIP-seq-VDD-9myc-rep2 21918605 11938940 50 PE

**Antibodies**

Anti-FLAG M2 (Sigma) 1:400 dilution
Anti-myc Cell Signaling 1:200 dilution

**Peak calling parameters**

-g 119146348 -q 0.01 -f BAM

**Data quality**

All identified peaks in the study were called with a qval threshold of 0.01 (FDR 1%).

**Software**

Trim Galore (v 0.6.7)
bowtie2 (v 2.3.4.3),
samtools (v 1.9)
MACS2 (v 2.1.1)
deeptools (v 3.1.3).
bedtools (v 2.26.0)
DESeq2 (v 1.42.1)
ggplot2 (v 3.5.1)
HOMER (v4.11.1)
FIMO (v5.5.7)
Tomtom (v5.5.7)
MEME 5.5.0

