## [Peer Review File · Nature Cell Biology]

REM transcription factors and GDE1 shape the DNA methylation landscape through the recruitment of RNA Polymerase IV transcription complexes.

Corresponding Author: Dr Steven Jacobsen

Version 0:

Decision Letter:

Revise extended OD

*Please delete the link to your author homepage if you wish to forward this email to co-authors.

Dear Dr Jacobsen,

Your manuscript, "REM transcription factors and GDE1 shape the DNA methylation landscape through the recruitment of RNA Polymerase IV transcription complexes.", has now been seen by 2 referees, who are experts in DNA methylation in plants (referee 1) and plant epigenetics (referee 2). We are still awaiting a report from the third reviewer and will forward this report on once we receive it.

As you will see from their comments (attached below) they find this work of potential interest, but have raised substantial concerns, which in our view would need to be addressed with considerable revisions before we can consider publication in Nature Cell Biology.

Nature Cell Biology editors discuss the referee reports in detail within the editorial team, including the chief editor, to identify key referee points that should be addressed with priority, and requests that are overruled as being beyond the scope of the current study. To guide the scope of the revisions, I have listed these points below. We are committed to providing a fair and constructive peer-review process, so please feel free to contact me if you would like to discuss any of the referee comments further.

In particular, it would be essential to:

- A- Perform additional experiments to validate whether GDE1 co-occupies genomic loci with MORC7, Pol IV, and Pol V at whole-genome level, as well as the predictions of the DNA-binding specificities of the VAL-VDD-GDE1 complex. (Reviewer#1 pts 1,8)
- B- Further characterize the domains and structural features of GDE1 (Reviewer#1 pt 11) and delineate the underlying mechanisms of the role of GDE1 in generation of 24nt-siRNAs (Reviewer#3 last bullet pt)
- C- Perform further experiments to determine whether the REM8-binding motif may facilitate the GDE1-independent recruitment of LSY3/4 to clsy3 clsy4 motifs for the production of 24-nt siRNAs (Reviewer#1 pt 10)
- D- Clarify the discrepancy in the GDE1-FLAG IP (Reviewer#1 pt4, Reviewer#3 first bullet pt)

E- All other referee concerns pertaining to strengthening existing data, providing controls, methodological details, clarifications and textual changes should also be addressed.

Please also note that if we do receive a report from Reviewer#3, we would expect their comments to also be addressed in full in a revised manuscript.

F- Finally please pay close attention to our guidelines on statistical and methodological reporting (listed below) as failure to do so may delay the reconsideration of the revised manuscript. In particular please provide:

We would be happy to consider a revised manuscript that would satisfactorily address these points, unless a similar paper is published elsewhere, or is accepted for publication in Nature Cell Biology in the meantime.

- ensure that it conforms to our format instructions and publication policies (see below and www.nature.com/nature/authors/).

- provide a point-by-point rebuttal to the full referee reports verbatim, as provided at the end of this letter.

- provide the completed Editorial Policy Checklist (found here <https://www.nature.com/authors/policies/Policy.pdf>), and Reporting Summary (found here <https://www.nature.com/authors/policies/ReportingSummary.pdf>). This is essential for reconsideration of the manuscript and these documents will be available to editors and referees in the event of peer review. For more information see <http://www.nature.com/authors/policies/availability.html> or contact me.

Nature Cell Biology is committed to improving transparency in authorship. As part of our efforts in this direction, we are now requesting that all authors identified as 'corresponding author' on published papers create and link their Open Researcher and Contributor Identifier (ORCID) with their account on the Manuscript Tracking System (MTS), prior to acceptance. ORCID helps the scientific community achieve unambiguous attribution of all scholarly contributions. You can create and link your ORCID from the home page of the MTS by clicking on 'Modify my Springer Nature account'. For more information please visit <http://www.springernature.com/orcid>.

Link Redacted

We would like to receive a revised submission within six months. We would be happy to consider a revision even after this timeframe, however if the resubmission deadline is missed and the paper is eventually published, the submission date will be the date when the revised manuscript was received.

We hope that you will find our referees' comments, and editorial guidance helpful. Please do not hesitate to contact me if there is anything you would like to discuss.

Best wishes,

Sabrya Carim

Sabrya Carim, PhD
(she/her/hers)
Associate Editor, Nature Cell Biology
Nature Portfolio

Springer Nature
The Campus, 4 Crinan Street, London N1 9XW, UK
sabrya.carim@springernature.com
<https://orcid.org/0000-0001-9485-1938>

Reviewers' Comments:

Reviewer #1:

Remarks to the Author:

In the plant RNA-directed DNA methylation pathway, Pol IV plays a crucial role in the production of 24-nt siRNAs. Prior research has established that the chromatin remodeling proteins CLSY1 and CLSY2 interact with SHH1, facilitating the recruitment of Pol IV to a specific subset of RdDM target loci. SHH1 recognizes histone modifications, which are involved in the recruitment process. Nonetheless, the question of whether alternative molecular mechanisms can direct Pol IV to RdDM genomic targets remains largely unclear. This study uncovered a novel protein, GDE1, and a family of REM transcription factors that participate in the recruitment of Pol IV, along with CLSY3 and CLSY4, to a distinct subset of RdDM target loci. Notably, the research revealed that GDE1 and REM transcription factors recruit Pol IV to target genomic loci by recognizing specific DNA motifs, uncovering a DNA sequence-dependent recruitment mechanism for Pol IV in RdDM. The results are presented in a concise manner, and they support the primary conclusions. Meanwhile, I have several comments that should be addressed in the revised manuscript. The following points outline these comments:

1. Although the study has utilized metaplots and heatmaps to illustrate the enrichment of GDE1 over MORC7, Pol IV, and Pol V target loci, a genome-wide correlation analysis between GDE1 ChIP-seq signals and those of MORC7, Pol IV, and Pol V would provide further validation. This analysis would ascertain whether GDE1 co-occupy genomic loci with these factors at the whole-genome level.
2. The study identified 753 sites where small interfering RNA (siRNA) levels were diminished in *clsy3 clsy4* double mutants compared to the wild type within ovule tissues. Referring to these sites simply as *clsy3 clsy4*-dependent siRNA sites can be misleading. The synthesis of these siRNAs relies on the function of the CLSY3 and CLSY4 proteins, rather than being a consequence of mutations in *clsy3* or *clsy4*. Therefore, it is more accurate to describe these sites as CLSY3/4-dependent siRNA sites.
3. The IP-MS data for GDE1-3FLAG and VDD-3FLAG are provided in Supplementary Tables 1 and 2, respectively. An appropriate title

should be added to these tables for clarity. Additionally, a clear definition of the numerical values should be presented within the tables.

4. In Extended Data Figure 2e, the GDE1-3FLAG signal was absent in the input samples but present in the immunoprecipitated samples, it is necessary to provide an explanation for this discrepancy. I wonder whether the GDE1 transgene was adequately expressed in the corresponding transgenic plants.

5. REM22 was detected only at 13 *clsy3 clsy4*-dependent siRNA loci (Figure 1f), a number that likely falls in the random overlap between REM22 and *clsy3 clsy4*-dependent siRNA loci. Furthermore, as indicated in Extended Data Figure 3e, CLSY3, CLSY4, and Pol IV seem to show a background ChIP-seq signal, suggesting that these REM22 target loci are unlikely to be RdDM target genomic loci. The related statement should be revised.

6. ChIP-seq experiments were conducted on various REM transcription factors, revealing that some are enriched at group 1 *clsy3 clsy4* siRNA loci, whereas others are enriched at group 2 siRNA loci. The underlying molecular mechanisms responsible for this differentiation need to be investigated. At least, a comparative analysis of the protein sequences or the predicted structures for the two types of REM transcription factors could reveal potential mechanisms.

7. The observation that REMs, CLSYs, GDE1, and Pol IV are enriched at the majority of triple and double repeat sites (Extended Data Figure 6a) is significant as it reveals a possible recruitment mechanism for RdDM components. These findings should be incorporated into the main figure.

8. It is noteworthy that the VAL-VDD-GDE1 complex selectively recognizes double repeat DNA sequences with a one-nucleotide spacer but fails to recognize those with a two-nucleotide spacer. The specificities of this recognition have been predicted by AlphaFold. Validation of these predictions through DNA-protein binding assays would substantially enhance the quality and impact of this research.

9. The data presented in Figure 3 indicate that the signals for CLSY3/4 and Pol IV are predominantly localized to one side of the motif center. The positioning of CLSY3/4 and Pol IV relative to the motif does not depend on the direction of the motif itself, but rather is dependent on uncharacterized factors. I propose that the positioning of accessible regions or nucleosomes in relative to the motif's location is crucial for determining the directionality of Pol IV transcription. I feel that examining accessible regions or nucleosome positions using previous DNase-seq, ATAC-seq, or MNase-seq data could provide valuable insights into the underlying molecular mechanisms.

10. ChIP-seq analysis has revealed a REM8-binding DNA motif, suggesting that it may facilitate the GDE1-independent recruitment of CLSY3/4 to *clsy3 clsy4* motifs for the production of 24-nt siRNAs. Is a *rem8* mutant available for additional investigation? Assessing the effect of the *rem8* mutation on the accumulation of 24-nt siRNAs specifically at REM8-bound *clsy3 clsy4* siRNA loci or at REM8-bound non-siren loci would further confirm these findings.

11. Since GDE1 is a previously unknown protein, it is necessary to delineate any conserved domains or predicted structural features. This information would enhance the reader's comprehension of the GDE1 protein's function.

12. In the materials and methods section, the term "milliliters of floral tissue homogenate" should be used to denote the amount of plant material employed in each experimental procedure. It is necessary to specify the weight of the plant material rather than the volume of the suspension.

Reviewer #2:

Remarks to the Author:

One of the central questions in chromosome biology is the identity of genetic determinants affecting epigenetic modifications. The pathfinder discoveries on this topic are with KRAB-zinc finger transcription factors in mammals and this study provides a clear illustration of a parallel set of mechanisms in Arabidopsis.

The epigenetic phenomenon under investigation is RNA-directed DNA methylation involving the atypical polymerase PolIV for which numerous mutation analyses led to the identification of protein cofactors. The CLSY family of proteins prominent such co-factors and previous studies revealed that they act in a cell type specific manner with CLSY3/4 playing a role in reproductive tissues.

The starting point for the present study was ChIP studies with MORC7 leading to the identification of GDE1 – a protein for which the ChIP DNA peaks were enriched for the DNA motifs that were known to bind the SNF transcription factors CLSY3/4. Other data confirming that GDE1 bound to the same chromosomal sites as MORC7, PolIV and PolIV and CLSY3/4 is evidence that these factors act together in RdDM.

Additional RdDM players revealed in this study are the REM transcription factors most notably VDD and VAL and IP-MS with VDD pulled down GDE1, CLSY3, PolIV and other cofactors of RdDM. VAL and other REM transcription factors were also implicated. Loss of function mutations in various REM genes further supported the involvement of these proteins recruitment of PolIV to the CLSY3/4 target loci (sirens). There is also a very nice gain of function experiment with VDD that adds additional evidence to support the importance of REM factor DNA binding as a determinant of PolIV-mediated RdDM.

Genomic analysis coupled to computation of VDD/VAL DNA binding structures supports the idea that the spacing of the CLSY3/4 motifs is important.

Overall this is a very nice paper and I have just a few minor comments:

- The failure to detect CLSY3/4 PolIV components in the GDE1-FLAG IP is surprising and the explanation that these interactions do not survive the IP-MS protocol is not consistent with the data in Fig 1g that pull downs with VDD included GDE1, CLSY3, PolIV subunits.
- The authors provide helpful information about the % of various *clsy*-dependent siRNA loci that are enriched in various fractions but I

would like to see some more complete book-keeping that tracks all of the PolIV-dependent loci. There are likely several determinant mechanisms associated with PolIV dependent siRNA production and it would be helpful to know where this GDE1-related mechanism fits into the overall picture.

- The identification of Group 2 loci at which GDE-1 is a suppressor of CLSY3/4-dependent sRNAs points to additional complexity in this mechanism. The group 2 and (including other loci with CLSY3/4 motif2 sequence) made me wonder where GDE1 fits into this mechanism. Could it be that GDE1 is a modulator of the process and, if so, under what circumstances would it play a role?

Reviewer #3:
Not yet received.

Methods should be written concisely, but should contain all elements necessary to allow interpretation and replication of the results. As a guideline, Methods sections typically do not exceed 3,000 words. The Methods should be divided into subsections listing reagents and techniques. When citing previous methods, accurate references should be provided and any alterations should be noted. Information must be provided about: antibody dilutions, company names, catalogue numbers and clone numbers for monoclonal antibodies; sequences of RNAi and cDNA probes/primers or company names and catalogue numbers if reagents are commercial; cell line names, sources and information on cell line identity and authentication. Animal studies and experiments involving human subjects must be reported in detail, identifying the committees approving the protocols. For studies involving human subjects/samples, a statement must be included confirming that informed consent was obtained. Statistical analyses and information on the reproducibility of experimental results should be provided in a section titled "Statistics and Reproducibility".

All Nature Cell Biology manuscripts submitted on or after March 21 2016 must include a Data availability statement at the end of the Methods section. For Springer Nature policies on data availability see <http://www.nature.com/authors/policies/availability.html>; for more information on this particular policy see <http://www.nature.com/authors/policies/data/data-availability-statements-data-citations.pdf>. The Data availability statement should include:

- Accession codes for primary datasets (generated during the study under consideration and designated as "primary accessions") and secondary datasets (published datasets reanalysed during the study under consideration, designated as "referenced accessions"). For primary accessions data should be made public to coincide with publication of the manuscript. A list of data types for which submission to community-endorsed public repositories is mandated (including sequence, structure, microarray, deep sequencing data) can be found here <http://www.nature.com/authors/policies/availability.html#data>.
- Unique identifiers (accession codes, DOIs or other unique persistent identifier) and hyperlinks for datasets deposited in an approved repository, but for which data deposition is not mandated (see here for details <http://www.nature.com/sdata/data-policies/repositories>).
- At a minimum, please include a statement confirming that all relevant data are available from the authors, and/or are included with the manuscript (e.g. as source data or supplementary information), listing which data are included (e.g. by figure panels and data types) and mentioning any restrictions on availability.
- If a dataset has a Digital Object Identifier (DOI) as its unique identifier, we strongly encourage including this in the Reference list and citing the dataset in the Methods.

We recommend that you upload the step-by-step protocols used in this manuscript to [protocols.io](https://www.protocols.io). More details can found at <https://www.protocols.io/help/publish-articles>.

All imaging data should be accompanied by scale bars, which should be defined in the legend. Cropped images of gels/blots are acceptable, but need to be accompanied by size markers, and to retain visible background signal within the linear range (i.e. should not be saturated). The boundaries of panels with low background have to be demarked with black lines. Splicing of panels should only be considered if unavoidable, and must be clearly marked on the figure, and noted in the legend with a statement on whether the samples were obtained and processed simultaneously. Quantitative comparisons between samples on different gels/blots are discouraged; if this is unavoidable, it should only be performed for samples derived from the same experiment with gels/blots were processed in parallel, which needs to be stated in the legend.

Unprocessed scans of all key data generated through electrophoretic separation techniques need to be presented in a supplementary figure that should be labelled and numbered as the final supplementary figure, and should be mentioned in every relevant figure legend. This figure does not count towards the total number of figures and is the only figure that can be displayed over multiple pages, but should be provided as a single file, in PDF or TIFF format. Data in this figure can be displayed in a relatively informal style, but size markers and the figure panels corresponding to the presented data must be indicated.

The total number of Supplementary Figures (not including the “unprocessed scans” Supplementary Figure) should not exceed the number of main display items (figures and/or tables (see our Guide to Authors and March 2012 editorial <http://www.nature.com/ncb/authors/submit/index.html#suppinfo>; <http://www.nature.com/ncb/journal/v14/n3/index.html#ed>). No restrictions apply to Supplementary Tables or Videos, but we advise authors to be selective in including supplemental data.

GUIDELINES FOR EXPERIMENTAL AND STATISTICAL REPORTING

REPORTING REQUIREMENTS – To improve the quality of methods and statistics reporting in our papers we have recently revised the reporting checklist we introduced in 2013. We are now asking all life sciences authors to complete two items: an Editorial Policy Checklist (found here <https://www.nature.com/authors/policies/Policy.pdf>) that verifies compliance with all required editorial policies and a reporting summary (found here <https://www.nature.com/authors/policies/ReportingSummary.pdf>) that collects information on experimental design and reagents. These documents are available to referees to aid the evaluation of the manuscript. Please note that these forms are dynamic ‘smart pdfs’ and must therefore be downloaded and completed in Adobe Reader. We will then flatten them for ease of use by the reviewers. If you would like to reference the guidance text as you complete the template, please access these flattened versions at <http://www.nature.com/authors/policies/availability.html>.

We strongly recommend the presentation of source data for graphical and statistical analyses as a separate Supplementary Table, and request that source data for all independent repeats are provided when representative experiments of multiple independent repeats, or averages of two independent experiments are presented. This supplementary table should be in Excel format, with data for different figures provided as different sheets within a single Excel file. It should be labelled and numbered as one of the supplementary tables, titled “Statistics Source Data”, and mentioned in all relevant figure legends.

Version 1:

Decision Letter:

Our ref: NCB-A54594A

21st March 2025

Dear Dr. Jacobsen,

Thank you for submitting your revised manuscript "REM transcription factors and GDE1 shape the DNA methylation landscape through the recruitment of RNA Polymerase IV transcription complexes." (NCB-A54594A) and for your patience with the review process. It has now been seen by the original referees and their comments are below. The reviewers find that the paper has improved in revision, and therefore we'll be happy in principle to publish it in Nature Cell Biology, pending minor revisions to satisfy the referees' final requests and to comply with our editorial and formatting guidelines.

*Please ensure to address the remaining minor points from the reviewers, around textual amendments, clarification and addition of information in the figure legends in the revised manuscript.

*Please ensure that all figures fit into a single standard page and adhere to a maximum page size of roughly 180mm wide x 200mm high, but also please use the full page space to fill the figure. At present several figures are too tiny to be legible once re-sized during the production process. To ensure legibility once figures are re-sized, please use a font size of no smaller than 6pt Arial or Helvetica throughout the figures.

We are now performing detailed checks on your paper and will send you a checklist detailing our editorial and formatting requirements in about 7-10 days. Please *do not upload the final materials and make any revisions* until you receive this additional information from us.

Thank you again for your interest in Nature Cell Biology Please do not hesitate to contact me if you have any questions.

Sincerely,
Sabrya Carim, PhD
(she/her/hers)
Senior Editor, Nature Cell Biology
Nature Portfolio

Springer Nature
The Campus, 4 Crinan Street, London N1 9XW, UK
sabrya.carim@springernature.com
<https://orcid.org/0000-0001-9485-1938>

Reviewer #1 (Remarks to the Author):

Most of my concerns have been well addressed, and the manuscript has been substantially improved. One remaining concern is the description of Figure 6d. The text indicated that "To explore this, we divided the motifs into two additional groups based on the relative positions of Pol IV summits, and observed that the direction of the Pol IV shift closely aligned with more accessible regions." In figure 6d, however, the 24-nt siRNA level and the MNase-seq level showed a positive correlation, suggesting that Pol IV shift closely aligned with more nucleosome-occupied regions, because the MNase-seq signals represent the nucleosome-occupied regions but not the accessible regions. This should be corrected in the revised manuscript.

*Additional Comments in response to Reviewer #2's remaining points:

I have read referee#2's comments and the authors' responses and revisions in the manuscript. I found that the comments have been well addressed in the revised manuscript. Although the CLSY3/4 and Pol IV were not co-purified with GDE1, they were shown to interact with GDE1 determined by co-IP. Moreover, group 1 and 2 siRNAs were characterized by additional experiments and analyses (Extended Data Fig. 6 and Fig. 2f). I feel that the manuscript has been significantly improved, and agree to publication.

Reviewer #3 (Remarks to the Author):

I commend the authors for doing a very professional and complete revision to this manuscript. Overall, this is a strong paper that pushes the field forward into more mammalian-like sequence-based targeting of repressive chromatin marks. One key difference is that this system based on REM TFs is still siRNA based, where in mammals the KRAB proteins directly target DNA methylation without small RNA production.

My remaining comment simply has to do with ChIP performed using tagged versions of proteins from which the tagged transgenic version has not been shown to complement the corresponding mutant. The authors do this for speed and ease of their experiments, and there is a lot of high quality supporting data shown in this paper. However, that was an industry-standard that this paper erodes, so therefore the lead author of this manuscript should not comment on other people's work that lacks this previously-necessary control.

Version 2:

Decision Letter:

Dear Dr Jacobsen,

I am pleased to inform you that your manuscript, "REM transcription factors and GDE1 shape the DNA methylation landscape through the recruitment of RNA Polymerase IV transcription complexes.", has now been accepted for publication in Nature Cell Biology. Congratulations!

Please note that *Nature Cell Biology* is a Transformative Journal (TJ). Authors may publish their research with us through the traditional subscription access route or make their paper immediately open access through payment of an article-processing charge (APC). Authors will not be required to make a final decision about access to their article until it has been accepted. [Find out more about Transformative Journals](https://www.springernature.com/gp/open-research/transformative-journals)

Authors may need to take specific actions to achieve [compliance with funder and institutional open access mandates](https://www.springernature.com/gp/open-research/funding/policy-compliance-faqs). If your research is supported by a funder that requires immediate open access (e.g. according to [Plan S principles](https://www.springernature.com/gp/open-research/plan-s-compliance)) then you should select the gold OA route, and we will direct you to the compliant route where possible. For authors selecting the subscription publication route, the journal's standard licensing terms will need to be accepted, including [self-archiving policies](https://www.springernature.com/gp/open-research/policies/journal-policies). Those licensing terms will supersede any other terms that the author or any third party may assert apply to any version of the manuscript.

If you have not already done so, we strongly recommend that you upload the step-by-step protocols used in this manuscript to protocols.io (<https://protocols.io>), an open online resource that allows researchers to share their detailed experimental know-how. All uploaded protocols are made freely available and are assigned DOIs for ease of citation. Protocols and Nature Portfolio journal papers in which they are used can be linked to one another, and this link is clearly and prominently visible in the online versions of both. Authors who performed the specific experiments can act as primary authors for the Protocol as they will be best placed to share the methodology details, but the Corresponding Author of the present research paper should be included as one of the authors. By uploading your

Protocols onto protocols.io, you are enabling researchers to more readily reproduce or adapt the methodology you use, as well as increasing the visibility of your protocols and papers. You can also establish a dedicated workspace to collect your lab Protocols. Further information can be found at <https://www.protocols.io/help/publish-articles>.

Nature Cell Biology encourages authors presenting evidence for cell, biological, molecular, and genetic interactions to consider communicating these findings using Biofactoid (<https://biofactoid.org/>). This tool helps users share a searchable representation of interactions (e.g. binding, gene expression, post-translational modification) between genes, gene products, or chemicals. Information added to Biofactoid, with author attribution, is shared on social media and public databases, such as Pathway Commons, where it can be discovered and analyzed in the context of a large and growing corpus of knowledge.

With best wishes,

Sabrya Carim, PhD
(she/her/hers)
Senior Editor, Nature Cell Biology
Nature Portfolio

Springer Nature
The Campus, 4 Crinan Street, London N1 9XW, UK
sabrya.carim@springernature.com
<https://orcid.org/0000-0001-9485-1938>

** Visit the Springer Nature Editorial and Publishing website at http://editorial-jobs.springernature.com?utm_source=ejp_NCB_email&utm_medium=ejp_NCB_email&utm_campaign=ejp_NCB for more information about our career opportunities. If you have any questions please click [here](mailto:editorial.publishing.jobs@springernature.com).

We thank the three reviewers for their constructive and helpful suggestions and comments, which helped us to improve the quality and thoroughness of our study. Our response in blue font can be found beneath each original reviewer's comment below.

Reviewers' Comments:

Reviewer #1:

Remarks to the Author:

In the plant RNA-directed DNA methylation pathway, Pol IV plays a crucial role in the production of 24-nt siRNAs. Prior research has established that the chromatin remodeling proteins CLSY1 and CLSY2 interact with SHH1, facilitating the recruitment of Pol IV to a specific subset of RdDM target loci. SHH1 recognizes histone modifications, which are involved in the recruitment process. Nonetheless, the question of whether alternative molecular mechanisms can direct Pol IV to RdDM genomic targets remains largely unclear. This study uncovered a novel protein, GDE1, and a family of REM transcription factors that participate in the recruitment of Pol IV, along with CLSY3 and CLSY4, to a distinct subset of RdDM target loci. Notably, the research revealed that GDE1 and REM transcription factors recruit Pol IV to target genomic loci by recognizing specific DNA motifs, uncovering a DNA sequence-dependent recruitment mechanism for Pol IV in RdDM. The results are presented in a concise manner, and they support the primary conclusions. Meanwhile, I have several comments that should be addressed in the revised manuscript. The following points outline these comments:

Thank you for these positive comments.

1. Although the study has utilized metaplots and heatmaps to illustrate the enrichment of GDE1 over MORC7, Pol IV, and Pol V target loci, a genome-wide correlation analysis between GDE1 ChIP-seq signals and those of MORC7, Pol IV, and Pol V would provide further validation. This analysis would ascertain whether GDE1 co-occupy genomic loci with these factors at the whole-genome level.

Thank you for this suggestion. We conducted a genome-wide pairwise correlation analysis using ChIP-seq signal intensities for GDE1, MORC7, Pol IV, and Pol V. The correlation heatmap, provided as Extended Data Fig 1b (and below), illustrates the degree of genome-wide co-localization among these factors. GDE1 exhibits a high correlation with MORC7 ($r = 0.61$) and Pol IV ($r = 0.64$), supporting their co-localization across genomic loci. In addition, this correlation analysis also suggests that GDE1 works more closely with the Pol IV arm than the Pol V arm of the RdDM pathway. We have also added the following to the text "GDE1 largely co-localized with MORC7 across the genome, as well as with key components of the RdDM pathway, including Pol IV and Pol V (Fig.1a-c and Extended Data Fig.1b). Genome-wide correlation analysis indicates that GDE1 is more closely associated with the Pol IV arm than the Pol V arm of the RdDM pathway. These results suggest that GDE1 is involved in RdDM function, with a stronger connection to the Pol IV arm".

Extended Data Fig. 1b. Genome-wide Spearman correlation analysis between GDE1 ChIP-seq signals and those of MORC7, Pol IV, and Pol V at co-targeted regions.

2. The study identified 753 sites where small interfering RNA (siRNA) levels were diminished in *clsy3 clsy4* double mutants compared to the wild type within ovule tissues. Referring to these sites simply as *clsy3 clsy4*-dependent siRNA sites can be misleading. The synthesis of these siRNAs relies on the function of the CLSY3 and CLSY4 proteins, rather than being a consequence of mutations in *clsy3* or *clsy4*. Therefore, it is more accurate to describe these sites as CLSY3/4-dependent siRNA sites.

Thank you for pointing this out. We have revised all “*clsy3 clsy4*-dependent siRNA sites” to “CLSY3 CLSY4-dependent siRNA sites” through the manuscript.

3. The IP-MS data for GDE1-3FLAG and VDD-3FLAG are provided in Supplementary Tables 1 and 2, respectively. An appropriate title should be added to these tables for clarity. Additionally, a clear definition of the numerical values should be presented within the tables.

Thank you for pointing this out. Titles and definition of the numerical values are now included in the Supplementary Tables. Supplementary Tables 1 and 2 show IP-MS data of GDE1-3FLAG and VDD-3FLAG, respectively. The numerical values are MS/MS counts of each protein from MaxQuant output.

4. In Extended Data Figure 2e, the GDE1-3FLAG signal was absent in the input samples but present in the immunoprecipitated samples, it is necessary to provide an explanation for this discrepancy. I wonder whether the GDE1 transgene was adequately expressed in the corresponding transgenic plants.

Thank you for pointing this out. GDE1 is predominantly expressed in the pistil and tapetum tissues (see Extended Data Fig. 3d). As a result, GDE1-3FLAG is present at very low abundance in flower buds isamples, making it undetectable in the input lanes. However, immunoprecipitation enriches for the GDE1-3FLAG protein, allowing it to be visualized in the immunoprecipitated samples. To further confirm that the GDE1 transgene is indeed expressed in the transgenic plants, we performed qPCR-PCR specifically on ovule tissues, which verified that GDE1-3FLAG is expressed at the expected levels in the parent lines (see Extended Data Fig. 7b). In addition to clarify why GDE1 is likely absent in the input we have added the following to the text “GDE1 proteins were scarcely detectable in input samples, likely due to their restricted expression in flower tissues. However, in the immunoprecipitated samples, GDE1 successfully pulled down CLSY3 and Pol IV (Extended Data Fig. 2b), suggesting that GDE1 forms a complex with CLSY3 and Pol IV.”

Extended Data Fig. 7b. *GDE1* gene expression in ovule tissues of indicated genotype as determined by qRT-PCR. Error bars represent means \pm standard deviation (sd) (n = 3).

5. REM22 was detected only at 13 *clsy3 clsy4*-dependent siRNA loci (Figure 1f), a number that likely falls in the random overlap between REM22 and *clsy3 clsy4*-dependent siRNA loci. Furthermore, as indicated in Extended Data Figure 3e, CLSY3, CLSY4, and Pol IV seem to show a background ChIP-seq signal, suggesting that these REM22 target loci are unlikely to be RdDM target genomic loci. The related statement should be revised.

Thank you for pointing this out. To check REM22 enrichment carefully, we replotted the REM22 ChIP-seq signal over the three groups of CLSY3 CLSY4-dependent sites (see Fig. 3c and Extended Data Fig. 3a). In the previous version of the manuscript, we plotted VDD, VAL, REM13, REM19 and REM22 on the same scale. However, the REM22 ChIP-seq enrichment is not as strong as the others factors, making the signal look weak in metaplots and heatmaps. REM22 showed significant enrichment at group 1. The relatively low overlap in the upset plot is likely due to stringent thresholds for identifying REM22 ChIP-seq peaks (DESeq, padj < 0.01, fold enrichment >2). We also included a better example in Extended Data Fig. 3c, showing co-localization between REM22 with CLSY3, CLSY4, and Pol IV. We also modified the text “Another factor, REM22, was enriched at a distinct subset of CLSY3 CLSY4-dependent siRNA loci within Group 1, but also showed some colocalization with VDD and VAL (Fig. 3b-e and Extended Data Fig. 3a and 3c).”

Fig 3c. Metaplot and heatmap showing enrichment of REM22 ChIP-seq signal over three groups of CLSY3 CLSY4-dependent sites (Left)

Extended Data Fig. 3c. Screenshot of REM22, VDD, VAL, REM13, REM19, REM8, GDE1, CLSY3, CLSY4 and Pol IV ChIP-seq at a distinct representative Group 1 of CLSY3 CLSY4-dependent siRNA sites, where REM22, but not other REM proteins exhibit enrichment. (Right)

6. ChIP-seq experiments were conducted on various REM transcription factors, revealing that some are enriched at group 1 *clsy3 clsy4* siRNA loci, whereas others are enriched at group 2 siRNA loci. The underlying molecular mechanisms responsible for this differentiation need to be investigated. At least, a comparative analysis of the protein sequences or the predicted structures for the two types of REM transcription factors could reveal potential mechanisms.

Thank you for this insightful suggestion. To address the question of how different REM transcription factors recognize distinct motifs, we performed additional AlphaFold3 structural predictions. Our analyses indicate that VDD, VAL, REM12, and REM13, which recognize CLSY3 CLSY4 motif 1, all contain two B3 domains and their C-terminal B3 domain seems to be responsible for DNA recognition. We now present these in Fig. 5a–c, Extended Data Fig. 7a. We also modified the text “All transcription factors (VDD, VAL, REM12, REM13) capable of recognizing the CLSY3 CLSY4 motif 1 feature two B3 domains, located at both ends. AlphaFold3 confidently predicted dimerization of these factors through their N-terminal B3 domain, while their C-terminal B3 domain were predicted to be responsible for DNA recognition across almost all combinations”.

Extended Data Fig. 7a. Predicted aligned error (PAE) of VDD-VAL-GDE1-motif1, REM12-REM13-GDE1-motif1, VDD-REM13-GDE1-motif1. (Top)

Fig. 5a. AlphaFold3 predicted the structure of VDD-VAL-GDE1 bound to dsDNA containing TTTTGCTATGTTTTGCTTAT (sequence below, high-affinity nucleotide acids red bolded). The DNA is shown as a ribbon representation. (Bottom)

In contrast, REM8—the only REM transcription factor identified so far to bind CLSY3 CLSY4 motif 2—possesses a central B3 domain and a C-terminal α -helix. However, we were unable to obtain a reliable structural prediction of how REM8 specifically interacts with motif 2.

Interestingly, the amino acid sequence of the REM8 B3 domain closely resembles that of other B3 domains—including those from VDD, VAL, REM12, REM13, and RAV1—which have been shown to bind a CACCTG motif (See Fig below). Thus, despite structural modeling, we cannot conclusively determine the distinct DNA-binding specificity of REM8 solely from its B3 domain. Perhaps additional factors—such as flanking domains, post-translational modifications, or interactions with co-regulatory proteins—may be involved in differential motif recognition.

Structural alignment of C-terminal B3 domains of RAV1, REM12, REM13, VAL, VDD and a central B3 domain of REM8.

We fully agree that uncovering the molecular mechanisms underlying these differences in motif binding is an important and exciting avenue for future research. However, an in-depth investigation of these structural or biochemical aspects is outside the current scope of our study. We will keep this intriguing question in mind as we refine our methodologies and develop follow-up experiments that can more definitively characterize the molecular determinants of motif specificity in REM transcription factors. Since we could not reach a definitive conclusion, and to avoid confusion, we did not include this comparative structural modeling analysis in the revised manuscript.

7. The observation that REMs, CLSYs, GDE1, and Pol IV are enriched at the majority of triple and double repeat sites (Extended Data Figure 6a) is significant as it reveals a possible recruitment mechanism for RdDM components. These findings should be incorporated into the main figure.

Thank you for your positive comment and suggestion. Figures have been reformatted for improved clarity and to align with the journal's formatting requirements. We now present this in Fig. 6a.

8. It is noteworthy that the VAL-VDD-GDE1 complex selectively recognizes double repeat DNA sequences with a one-nucleotide spacer but fails to recognize those with a two-nucleotide spacer. The specificities of this recognition have been predicted by AlphaFold. Validation of these predictions through DNA-protein binding assays would substantially enhance the quality and impact of this research.

This is a great suggestion. To test this, we performed DNA Affinity Purification sequencing (DAP-seq), which allow to test the ability of VDD-VAL-GDE1 to bind naked genomic DNA *in vitro*. We aligned of all sequences bound to the VDD-VAL-GDE1 complex bound and they all have one-nucleotide spacer. Results have been added to Fig. 6b in the revised manuscript. We have also added the following to the text “In addition, we performed DNA affinity purification sequencing (DAP-seq) by incubating Halo-tagged recombinant proteins with DNA extracted from WT unopened flower buds to test the ability of the VDD-VAL-GDE1 complex to bind Arabidopsis genomic DNA *in vitro*. All sequences that bound by VDD-VAL-GDE1 complex had a one-nucleotide spacer. Overall, these results strongly support the functional significance of this single nucleotide spacer and that the REM-GDE1 complex can directly bind the CLSY3 CLSY4 motif 1.”

Fig. 6b. Sequence logo assessment of nucleotides identified by VDD-VAL-GDE1 DAP-seq.

9. The data presented in Figure 3 indicate that the signals for CLSY3/4 and Pol IV are predominantly localized to one side of the motif center. The positioning of CLSY3/4 and Pol IV relative to the motif does not depend on the direction of the motif itself, but rather is dependent on uncharacterized factors. I propose that the positioning of accessible regions or nucleosomes in relative to the motif's location is crucial for determining the directionality of Pol IV transcription. I feel that examining accessible regions or nucleosome positions using previous DNase-seq, ATAC-seq, or MNase-seq data could provide valuable insights into the underlying molecular mechanisms.

This is an excellent suggestion, and it looks like you are right! We plotted MNase-seq data over the regions where Pol IV shifted either left or right, and observed that the direction of the Pol IV shift closely aligned with more accessible regions. Results have been added in the revised manuscript at Fig. 6d. We have also added the following to the text “An alternative explanation

is that the directionality of Pol IV transcription may be influenced by the accessibility of the regions. To explore this, we divided the motifs into two additional groups based on the relative positions of Pol IV summits, and observed that the direction of the Pol IV shift closely aligned with more accessible regions.”

Fig. 6d. Metaplot and heatmap showing enrichment of CLSY3, CLSY4, Pol IV ChIP-seq, MNase-seq signal and 24nt-siRNA signal over Pol IV left shift (n=23) or Pol IV right shift (n=27).

10. ChIP-seq analysis has revealed a REM8-binding DNA motif, suggesting that it may facilitate the GDE1-independent recruitment of CLSY3/4 to *clsy3 clsy4* motifs for the production of 24-nt siRNAs. Is a *rem8* mutant available for additional investigation? Assessing the effect of the *rem8* mutation on the accumulation of 24-nt siRNAs specifically at REM8-bound *clsy3 clsy4* siRNA loci or at REM8-bound non-siren loci would further confirm these findings.

Thank you for this suggestion. We failed to obtain any good T-DNA insertion lines from the stock center. Therefore, a *REM8* deletion mutant was generated using CRISPR/Cas9 (Extended Data Fig. 9a). However, this mutation did not affect siRNA production at REM8 binding sites (Extended Data Fig. 9b). Given the genetic redundancy of REM transcription factors at CLSY3 CLSY4 motif 1, it is likely that additional factors contribute to the recognition of motif 2. Results have been added to the revised manuscript at Extended Data Fig. 9a-b. We also modified the text “A *REM8* deletion mutant was generated using CRISPR/Cas9, but this mutation did not affect siRNA production at REM8 binding sites. Given the genetic redundancy of REM transcription factors at CLSY3 CLSY4 motif 1, it is likely that additional factors contribute to the recognition of motif 2.”

Extended Data Fig. 9. a, Schematic description of CRISPR/Cas9 construct design for knocking out REM8. Deletion detected in genomic DNA of *rem8-cr* mutants are shown with chromatographs from Sanger sequencing. b, Violin plot showing the 24nt-siRNA levels at REM8 binding non-siren loci (n=162) in the indicated genotypes.

11. Since GDE1 is a previously unknown protein, it is necessary to delineate any conserved domains or predicted structural features. This information would enhance the reader's comprehension of the GDE1 protein's function.

Thank you for this excellent suggestion. We have performed more AlphaFold3 predictions and found an α -helix of GDE1 (amino acids 249-261, hereafter name RBHG (REM-binding helix of GDE1) domain) is confidently predicted to fit into a pocket formed by the transcription factor dimers, establishing extensive electrostatic and hydrophobic interactions (Fig. 5a-b and Extended Data Fig. 7a). To experimentally assess the significance of the α -helix of GDE1's association with REM transcription factor dimers, a triple mutant (E251A/Y252A/Y255A) was generated and introduced into the *gde1-1* mutant. Even though this mutant produced GDE1 transcript levels comparable to the WT GDE1, it did not fully rescue the siRNA deficiency in the *gde1-1* mutant (Fig. 5d and Extended Data Fig. 7b). These results suggest that this α -helix of GDE1 is critical for function, consistent with its proposed role in interacting with REM transcription factors. Results have been added to the revised manuscript at Fig. 5a-b and Extended Data Fig. 7a-b, with the above text included as a section in the Results.

Fig. 5. The RBHG domain of GDE1 is critical for siRNA production.

a, Table showing the residues and interaction formed between RBHG domain of GDE1 and transcription factor dimers.

b, AlphaFold3 predicted the structure of VDD-VAL-GDE1 bound to dsDNA containing TTTTGCTTATGTTTGGCTTAT (sequence below, high-affinity nucleotide acids red bolded). The DNA is shown as a ribbon representation.

c, Interaction of RBHG domain of GDE1 with VDD-VAL. Interacting residues are highlighted in sticks. Hydrogen bonds and pi-bond are highlighted dashed lines.

d, Violin plot showing the 24nt-siRNA levels at siren loci (n=133) in the indicated genotypes. *p* values calculated by pairwise t-tests are indicated.

Additionally, further analysis of the genomic features of the two groups of CLSY3 CLSY4-dependent siRNA loci revealed an intriguing pattern: the majority of Group 2 sites are found in heterochromatic regions, while Group 1 sites are predominantly located in euchromatic regions (Fig. 2f). This suggests that GDE1 primarily regulates siRNA production in euchromatic regions and restricts Pol IV recruitment to heterochromatic regions. Therefore, in *gde1* mutant, the CLSY3/4/Pol IV complex is released from Group 1 sites and it winds up at Group 2 sites, leading

to an increased production of siRNAs at those group 2 sites. These results have been added to Fig. 2f in the revised manuscript, with the above text included.

Fig. 2f Circular genome view showing the enrichment of Group 1 and 2 of CLSY3 CLSY4-dependent siRNA across all five chromosomes, with the pericentromeric heterochromatin marked in red along the inner circle.

In addition, we also include a paragraph in the discussion section to talk about the function of GDE1. “Our study identifies GDE1 as a critical mediator that enhances the recruitment of CLSY3/4 to siren loci. The strong enrichment of CLSY3/4 and Pol IV at siren loci is lost in the *gde1-1* mutant, where these complexes are redistributed to Group 2 loci. This redistribution underscores the specificity of GDE1 in guiding siRNA biogenesis at specific CLSY3 CLSY4-dependent loci, reinforcing its central role in the spatial and functional organization of Pol IV recruitment.”

12. In the materials and methods section, the term "milliliters of floral tissue homogenate" should be used to denote the amount of plant material employed in each experimental procedure. It is necessary to specify the weight of the plant material rather than the volume of the suspension.

Thank you for pointing this out. The weight of the plant material for ChIP-seq and IP-MS has added to the revised manuscript.

Reviewer #2:

Remarks to the Author:

One of the central questions in chromosome biology is the identity of genetic determinants affecting epigenetic modifications. The pathfinder discoveries on this topic are with KRAB-zinc finger transcription factors in mammals and this study provides a clear illustration of a parallel set of mechanisms in Arabidopsis.

The epigenetic phenomenon under investigation is RNA-directed DNA methylation involving the atypical polymerase PolIV for which numerous mutation analyses led to the identification of protein cofactors. The CLSY family of proteins prominent such co-factors and previous studies revealed that they act in a cell type specific manner with CLSY3/4 playing a role in reproductive tissues.

The starting point for the present study was chIP studies with MORC7 leading to the identification of GDE1 – a protein for which the chIP DNA peaks were enriched for the DNA motifs that were known to bind the SNF

transcription factors CLSY3/4. Other data confirming that GDE1 bound to the same chromosomal sites as MORC7, PolV and PolIV and CLSY3/4 is evidence that these factors act together in RdDM.

Additional RdDM players revealed in this study are the REM transcription factors most notably VDD and VAL and IP-MS with VDD pulled down GDE1, CLSY3, PolIV and other cofactors of RdDM. VAL and other REM transcription factors were also implicated. Loss of function mutations in various REM genes further supported the involvement of these proteins recruitment of PolIV to the CLSY3/4 target loci (sirens). There is also a very nice gain of function experiment with VDD that adds additional evidence to support the importance of REM factor DNA binding as a determinant of PolIV-mediated RdDM.

Genomic analysis coupled to computation of VDD/VAL DNA binding structures supports the idea that the spacing of the CLSY3/4 motifs is important.

Overall this is a very nice paper and I have just a few minor comments:

Thank you for the positive comments!

- The failure to detect CLSY3/4 PolIV components in the GDE1-FLAG IP is surprising and the explanation that these interactions do not survive the IP-MS protocol is not consistent with the data in Fig 1g that pull downs with VDD included GDE1, CLSY3, PolIV subunits.

Thank you for pointing this out. We believe that co-IP experiments are much more sensitive than IP-MS procedures which undergo much more stringent washing conditions. To avoid confusion, we have revised our explanation to “To understand how GDE1 influences siRNA production, we performed IP-MS with GDE1-3FLAG transgenic plants to uncover interacting proteins. Unexpectedly, none of the CLSY3/4-Pol IV components were identified in the IP-MS experiments (Fig. 3a and Supplementary Table 1). To further assess potential interactions using a more sensitive approach, we conducted co-immunoprecipitation (co-IP) assays using F2 transgenic plants expressing GDE1-3FLAG with CLSY3-9myc or Pol IV-9myc. GDE1 proteins were scarcely detectable in input samples, likely due to their restricted expression in flower tissues. However, in the immunoprecipitated samples, GDE1 successfully pulled down CLSY3 and Pol IV (Extended Data Fig. 2b), suggesting that GDE1 forms a complex with CLSY3 and Pol IV.”

- The authors provide helpful information about the % of various *clsy*-dependent siRNA loci that are enriched in various fractions but I would like to see some more complete book-keeping that tracks all of the PolIV-dependent loci. There are likely several determinant mechanisms associated with PolIV dependent siRNA production and it would be helpful to know where this GDE1-related mechanism fits into the overall picture.

Thank you for this excellent suggestion to provide a more comprehensive overview of Pol IV-dependent siRNA loci and how GDE1-related mechanisms integrate into the broader framework. We have added a new section in the Results to address this point (see ‘REM transcription factors-initiated siRNA biogenesis is independent of H3K9 methylation.’)

Briefly, we examined H3K9 methylation mutants (*svh4 svh5 svh6*) and observed that these mutations predominantly affect CLSY1 CLSY2-dependent loci, whereas little overlap was found with REM-dependent, CLSY3 CLSY4-regulated loci (Extended Data Fig. 6a-b). Correspondingly, siRNAs at siren loci were unaffected in both *clsy1 clsy2* and *svh4 svh5 svh6* mutants (Extended Data Fig. 6c), indicating that REM transcription factor-initiated siRNA production proceeds largely independently of H3K9 methylation.

More specifically, our analysis revealed that in ovule tissues, 20.25% of Pol IV-dependent 24-nt siRNA clusters are markedly reduced in the *clsy3 clsy4* double mutant, and these loci rely on REM transcription factors plus GDE1 for Pol IV recruitment (Extended Data Fig. 6d). By comparison, 9.68% of Pol IV-dependent 24-nt siRNA clusters require CLSY1/2, where H3K9 methylation plays a critical role (Extended Data Fig. 6d). These data position the GDE1-related mechanism as one of at least two main strategies for Pol IV recruitment in ovule tissues—one driven by REM transcription factors (including GDE1), and another involving H3K9 methylation. Results have been added to the revised manuscript at Extended Data Fig. 6a-d, with the above text included as a section in the Results.

Extended Data Fig. 6. REM transcription factors-initiated siRNA biogenesis is independent of H3K9 methylation.

a, Venn diagrams showing the relationships between reduced 24nt-siRNA clusters in *svh4 svh5 svh6* triple and the *clsy1 clsy2* double mutants.

b, Venn diagrams showing the relationships between reduced 24nt-siRNA clusters in *svh4 svh5 svh6* triple, *clsy3 clsy4* double and the *rem46 val rem12* triple mutants.

c, Violin plot showing the 24nt-siRNA levels at siren loci (n=133) in the indicated genotypes.

d, Pie charts showing the proportions of 24nt-siRNAs from all Pol IV-dependent clusters in ovule tissue that are present in *clsy1 clsy2* or *clsy3 clsy4* double mutants and the proportions of GDE1-dependent siRNA among CLSY3 CLSY4-dependent group or SUVH456-dependent siRNA among CLSY1 CLSY2-dependent group.

- The identification of Group 2 loci at which GDE-1 is a suppressor of CLSY3/4-dependent sRNAs points to additional complexity in this mechanism. The group 2 and (including other loci with CLSY3/4 motif2 sequence) made me wonder where GDE1 fits into this mechanism. Could it be that GDE1 is a modulator of the process and, if so, under what circumstances would it play a role?

Thank you for this suggestion. In looking into this further, we observed that the majority of Group 2 sites are found in heterochromatic regions, while Group 1 sites are predominantly

located in euchromatic regions (Fig. 2f). This suggests that GDE1 primarily regulates siRNA production in euchromatic regions and competes with Pol IV recruitment to heterochromatic regions. Therefore, in *gde1* mutant, the CLSY3/4/Pol IV complexes are released from Group 1 sites and wind up at Group 2 sites, leading to an increased production of siRNAs at those group 2 sites. These results have been added to the revised manuscript at Fig. 2f. We have also added the following to the text “Interestingly, the majority of Group 2 sites were found in heterochromatic regions, while Group 1 sites were predominantly located in euchromatic regions (Fig. 2f). This suggests that GDE1 primarily regulates siRNA production in euchromatic regions and competes with Pol IV recruitment to heterochromatic regions.”

Fig. 2f. Circular genome view showing the enrichment of Group 1 and 2 of CLSY3 CLSY4-dependent siRNA across all five chromosomes, with the pericentromeric heterochromatin marked in red along the inner circle.

Reviewer #3

(Remarks to the Author)

In this manuscript the CLASSY protein recruitment to specific DNA motifs is further characterized and a transcription factor network is elucidated that goes from DNA motif to tissue-specific binding, protein recruitment and RNA Polymerase IV recruitment. A strength of this manuscript is that once introducing a new protein factor, the next sentence is the CHIP data. There is a large amount of data in the manuscript, which helps to define a sequence-specific system of 24-nt siRNA biogenesis. I'm impressed with the sheer volume of information in this manuscript.

Thank you for the positive comments!

Major comments to address

1. The abstract of the paper starts by describing the importance of DNA methylation on transposon and gene regulation. But the rest of the paper focuses on siren loci, which as far as I can tell are not involved in transposon or gene regulation. The manuscript needs to be more honest that the ovule and anther regions under study have no known function or effect on anything. An interesting point here is to try to understand if the VDD strong phenotype is due to the loss of 24-nt siRNAs? Or is this independent, telling us that this transcription factor network also does other important things besides make siRNAs at these select loci?

Thank you for this suggestion. We have revised the abstract to better align with the focus of the manuscript and have toned down the emphasis on the importance of DNA methylation on transposon and gene regulation. In addition, two recently published articles have described the function of siren siRNA in hybrid seed failure in *Capsella* (Dziasek, et al, 2024 Nature Plants) and seed abortion in *Brassica rapa* (Burgess, et al., 2022 The Plant Cell). We have included these discoveries in the introduction section to highlight the broader relevance of siren siRNAs and their potential functional roles as “These siren-derived siRNAs promote DNA methylation both in cis at their originating loci and also in trans at certain protein-coding genes, modifying gene expression and influencing reproductive processes, such as hybrid seed failure in *Capsella* and seed abortion in *Brassica rapa*.”

Thank you also for the comment regarding the VDD strong phenotype. Based on our analysis, we do not believe the lethality phenotype of the VDD knock-out mutant is simply due to the loss of 24-nt siRNAs. For instance, neither the *clsy3 clsy4* double mutant nor the *pol iv* single mutant shows any obvious developmental phenotype. Previous literature has described that the expression of *GAMETOPHYTIC FACTOR 2 (GFA2)*, a key regulator of synergid cell death, can rescue the phenotypic effects observed in the *vdd-1/+* mutant. This suggests that VDD transcription factors may have a critical role in developmental regulation that is independent of siRNA pathways. We have added a paragraph in the discussion section to explore this possibility further and to acknowledge the potential independent functions of VDD transcription factors beyond siRNA production as “VDD and VAL, two REM transcription factors that target siren loci, have been reported as critical for the degeneration of synergid cells, a vital step in ensuring successful fertilization. *vdd-1/+* and *VAL RNAi* mutants exhibit pronounced female gametophytic defects, underscoring the importance of these factors in reproductive success. However, the viability of our *rem46 val rem12* triple mutant in this study suggests that the reduced fertility observed in the *VAL RNAi* mutant may result from off-target silencing of related homologs. The expression of *GAMETOPHYTIC FACTOR 2 (GFA2)*, a key regulator of synergid cell death, can rescue the phenotypic effects observed in the *vdd-1/+* mutant. Notably, none of *gde1-1*, *clsy3 clsy4*, and *pol iv* mutants display any obvious developmental phenotypes, implying that these factors may not directly contribute to developmental processes. Our results also suggest diverse regulatory roles of REM transcription factors using siRNA-dependent and siRNA-independent mechanisms.”

2. The findings of this work characterize a previously-discovered shift in dogma in the production of 24 -nt siRNAs from the idea that it is a chromatin-mark that is drawing Pol IV, to a sequence-specific transcription factor. This was dogma that this lab set up in the field, and I don't believe the two systems are adequately reconciled in this manuscript. For example, does this mean that in cells that do not express Classy 1 and 2, that there is no H3K9me2-based recruitment of Pol IV? This work needs to include a key mutant that cannot place or read H3K9me2, to demonstrate the independence of these transcription factor networks from this mark.

Thank you for this excellent suggestion to provide a more comprehensive overview of these two distinct Pol IV recruitment pathways. As suggested, we have included the H3K9 methylation triple mutant (*svh4 svh5 svh6*), and the results are now included in a new section of the Results, titled “REM transcription factors-initiated siRNA biogenesis is independent of H3K9 methylation.” Briefly, we found that *svh4 svh5 svh6* predominantly affected CLSY1 CLSY2-

dependent loci, whereas little overlap was found with REM-dependent, CLSY3 CLSY4-regulated loci (Extended Data Fig. 6a-b). Correspondingly, siRNAs at siren loci were unaffected in both *clsy1 clsy2* and *suvh4 suvh5 suvh6* mutants (Extended Data Fig. 6c), indicating that REM transcription factor–initiated siRNA production proceeds largely independently of H3K9 methylation. Results have been added to the revised manuscript at Extended Data Fig. 6a-d.

Additionally, previous studies (Zhou et al., 2018 Nature Genetics; 2022 Nature Communications) have demonstrated that the CLSY1 and CLSY2 remodelers help link H3K9 methylation readers, such as SAWADEE DOMAIN HOMOLOG 1 (SHH1), to Pol IV for efficient siRNA production and DNA methylation. In contrast, CLSY3 and CLSY4 recruit Pol IV through an SHH1-independent mechanism. Our findings further establish that the REM transcription factor recruitment mechanism relies on CLSY3 and CLSY4.

Altogether, our current work suggests that CLSY proteins have evolved multiple distinct mechanisms, some based on epigenetic information, and some based on genetic information, for recruiting Pol IV to specific sites.

Extended Data Fig. 6. REM transcription factors-initiated siRNA biogenesis is independent of H3K9 methylation.

a, Venn diagrams showing the relationships between reduced 24nt-siRNA clusters in *suvh4 suvh5 suvh6* triple and the *clsy1 clsy2* double mutants.

b, Venn diagrams showing the relationships between reduced 24nt-siRNA clusters in *suvh4 suvh5 suvh6* triple, *clsy3 clsy4* double and the *rem46 val rem12* triple mutants.

c, Violin plot showing the 24nt-siRNA levels at siren loci (n=133) in the indicated genotypes.

d, Pie charts showing the proportions of 24nt-siRNAs from all Pol IV-dependent clusters in ovule tissue that are present in *clsy1 clsy2* or *clsy3 clsy4* double mutants and the proportions of GDE1-dependent siRNA among CLSY3 CLSY4-dependent group or SUVH456-dependent siRNA among CLSY1 CLSY2-dependent group.

3. This work is highly correlative and misses key controls at the expense of moving quickly. Many of the informatic experiments do not have good negative controls. For example, in Figure 2A, 40 siren loci are identified that do not have VDD/VAL/REM binding because they do not have the DNA sequence motifs. These loci are a key missing control, and should be continued to be studied throughout the analysis of Figure 2. Since they lack motifs and binding, they should not have loss of 24-nt siRNAs in the mutants and loss of CHH methylation. In Figure 2C, split the siren loci into the 93 and 40 from Figure 2A, and only the 93 should have a loss of 24-nt siRNAs, not the 40. Similarly, in Figure 2E there seems to be a significant number of loci that lose siRNA in the triple mutant that are not siren or *clsy3/4* loci. Do these have the motifs? The fact that many of the Venn diagrams mostly overlap is suggestive and provides correlative support, but the fact that there are above-background levels of regions in the Venn diagram that do not overlap, suggest these are not direct / causative mechanistic relationships.

Thank you for these helpful suggestions. We apologize for any confusion caused by the initial presentation of the data. To improve clarity and alignment with the journal's formatting requirements, we have reformatted all figures and text and re-analyzed the data to address your concerns.

In response to your comment regarding the Venn diagram in Figure 2A, the original analysis focused on the overlap between siren loci and common targets of VDD, VAL, and REM13. We have updated this analysis to present a new Venn diagram that separately examines the relationship among the three transcription factors and their overlap with siren loci. We now present this in Fig. 4a. We found that all three transcription factors are highly overlapped. Secondly, only 13 siren loci were not directly targeted by these transcription factors, but we still observed a reduction in siRNA levels at these 13 siren loci (See figure below).

Fig. 4a Venn diagram showing the relationship between siren loci and VDD, VAL, and REM13 ChIP-seq targets. (Left) Violin plot showing the 24nt-siRNA levels at Non-REM target siren loci (n=13) in the indicated genotypes. (Right)

One possibility is that these transcription factors can target these sites naturally but we failed to identify these peaks due to the limit sensitivity of ChIP-seq. Secondly, it is notable that only 86 out of 133 siren loci contain the motif, but 120 siren loci are targeted by these transcription factors, suggesting motif recognition may not be the sole determinant of binding specificity. In addition, we also found that other transcription factors, like REM19, REM22 and REM8, were enriched at siren loci. We speculate that these factors may form heterodimers with VDD, VAL, or REM transcription factors, enabling binding to loci that lack the CLSY3 CLSY4 motif 1. This heterodimerization may explain why additional loci are affected in the *rem46 val rem12* mutants.

To enhance clarity, we have also added the following to the discussion section “Although we identified only 86 CLSY3 CLSY4 motif 1 sites across the genome, over 400 loci were affected in the *rem46 val rem12* triple mutants. This observation suggests that REM transcription factors do not exclusively recognize the CLSY3 CLSY4 motif 1. Instead, additional transcription factors such as REM19, REM22, and REM8 are also enriched at siren loci. Given their *in vitro* association, these factors may form distinct combinatory heterodimers, enabling broader recruitment and siRNA regulation at loci that lack the CLSY3 CLSY4 motif 1. This mechanism of cooperative binding and redundancy likely enhances the robustness of siRNA biogenesis, ensuring proper siRNA regulation even when individual motifs or factors are absent.”

4. The separation of Classy3/Classy4-bound regions into “Group 1” and “Group 2” was confusing to me. What are the genome annotations of this group and how do they differ other than the circular reasoning of having different proteins bound to them? The violin box plots in Figure 2B are circular reasoning, because the groups are defined by having classy3/4 -dependent siRNA production, and then the production of siRNAs are measured for each group. Ex. Fig 2D shows browser shots and these clearly are not genes, but are these transposons or other annotated parts of the genome? What are the loci we are studying??

Thank you for pointing this out and sorry for the confusion. To clarify, we categorized the CLSY3/CLSY4-dependent loci into three groups: Group 1 loci, which are GDE1-dependent; Group 2 loci, which show increased siRNA production in the *gde1-1* mutant; and Group 3 loci, which show no significant change. To enhance clarity, we have also added the following to the text “To determine whether GDE1 affect siRNAs biogenesis, the levels of siRNA in *gde1-1* (*SALKseq_10069.1*) ovule tissue were profiled genome-wide. Strikingly, 57% of these CLSY3 CLSY4-dependent siRNA loci showed reduced siRNAs in *gde1-1*, while 35% exhibited increased siRNAs in *gde1-1* (Fig. 2a-c and Extended Data Fig. 2a), suggestive of overlapping and differential siRNA modulation function of GDE1 with CLSY3 and CLSY4. Within these CLSY3 CLSY4-dependent siRNA loci, we defined GDE1-dependent loci as Group 1 sites, while loci showing increased siRNA in *gde1-1* were defined as Group 2 sites and the rest of the sites were defined as Group 3 sites (Fig. 2a).”

Thank you for your suggestion on annotating Group 1 vs Group 2 to see how they may differ. Interestingly, we found that the majority of Group 2 loci were located in heterochromatic regions, while Group 1 loci were predominantly found in euchromatic regions (Fig. 2f). This indicates that GDE1 primarily regulates siRNA production in euchromatic regions while competing with Pol IV recruitment to heterochromatic regions. These new results have been added to the revised manuscript at Fig. 2f, with the above text included.

Fig. 2f. Circular genome view showing the enrichment of Group 1 and 2 of CLSY3 CLSY4-dependent siRNA across all five chromosomes, with the pericentromeric heterochromatin marked in red along the inner circle.

With regard to the violin box plots, we think the reviewer is likely instead referring to Fig. 1B. We have defined these groups based on thresholds. The purpose of these plots was to quantitatively measure the siRNA levels in the two groups across the *clsy3 clsy4* and *gde1-1* backgrounds, demonstrating that the grouping is logical and meaningful. These plots are now presented in Fig. 2b-c.

Extended Data Fig 2D showed browser shots examples of Group 1 and Group 2 loci. You are correct that these are not genes. Both Group1 and Group2 sites are located in distal intergenic regions and promoters as shown in the pie charts below. To avoid potential confusion, we have decided not to include this promoter/gene annotation in the revised manuscript. We feel the differences in Group 1 vs Group 2 in heterochromatic and euchromatic locations are a more interesting phenomenon to include in the paper.

Pie chart indicating the location of Group 1 and Group 2 CLSY3 CLSY4-dependent siRNA sites.

In addition, to provide better context for the loci described in this study, we have included more details about siren and HyperTE loci in the introduction section as follows: “In leaves, CLSY1 and CLSY2 dominate siRNA biogenesis, while in ovules, CLSY3 and CLSY4 are highly expressed and

direct Pol IV complexes to generate siRNAs at specific loci called siren (small-interfering RNA in the endosperm) loci. These siren-derived siRNAs promote DNA methylation both in cis at their originating loci and also in trans at certain protein-coding genes, modifying gene expression and influencing reproductive processes, such as hybrid seed failure in *Capsella* and seed abortion in *Brassica rapa*. A similar mechanism is observed in the male germ line, where the CLSY3 CLSY4-dependent HyperTE loci in tapetal nurse cells produce siRNAs that move into microspore mother cells, directing DNA methylation and gene regulation.”

4b. There are a small number of loci in these groups, and on line 136 I don't observe any enrichment in group 2. I'm struggling to find the REM12 data on line 139 and don't see the REM19 data on line 141. This paragraph is highly correlative.

We apologize for the confusion in the original presentation of this paragraph. We have revised this section to provide clearer descriptions and better connections between the data and our interpretations. On line 136, we were describing the REM19 ChIP-seq result. On line 141, we claimed that REM19 also shows localization at a subset of Group 2 loci. In the previous version of the manuscript, we plotted VDD, VAL, REM13 and REM19 on the same scale. However, the REM19 ChIP-seq enrichment is not as strong as the others factors, making the signal look weak in metaplots and heatmaps. We have now generated an updated REM19 plot showing enrichment at Group 2 loci to a lesser extent, which is now presented in Fig 3c. In addition, Fig 3h shows the summary of TF binding percentage across the three groups.

Fig. 3c Metaplot showing enrichment of VDD, VAL, REM13, REM19, REM22 and REM8 ChIP-seq signal over three groups of CLSY3/CLSY4-dependent sites. (Left)

Fig. 3h Bar plot showing percentage of each REM transcription factors binding to three distinct groups of CLSY3 CLSY4-dependent siRNA loci. (Right)

We apologize that we did not reference the figures properly on line 139. We have added the corresponding figure reference in the revised manuscript. As shown in Fig 3g, REM12 DAP-seq locations highly overlapped with the same set of CLSY3/CLSY4-dependent siRNA loci bound by VDD and VAL. In addition, REM12 displayed a similar motif with CLSY3 CLSY4 motif1 in Extended Data Fig. 3b.

Fig. 3g Venn diagram showing the similarities between VDD/VAL common targeted siRNA loci with REM12 targeted siRNA loci.

4c. In Ex. Figure 3A, I see strong enrichment for a very small number of loci in Group 1, not the whole group that is assumed in the paper. Not the 429 loci, but likely just 50. I just think these groupings are artificial and don't hold up well. I suggest focusing on genome annotations or other tangible features rather than the seemingly arbitrary Group 1 and 2.

Thank you for making these points. To clarify, the 429 loci in Group 1 shown in extended Figure 3A were defined as those in which the siRNAs were dependent on GDE1. The 50 or so strong signals the reviewer is referring to are the strong signals from the binding of VDD, VAL, etc, and we agree that this is only a subset of the Group 1 loci. We also appreciate the reviewer's comment about annotation. As it turns out, this strongly bound subset of Group 1 primarily corresponds to the siren loci, extensively focused later in the manuscript. We acknowledge that we have very likely not identified all the transcription factors involved at these loci, which is likely why the strong signals in this figure are only 50 or so of the 429 Group 1 loci, and the purpose of this figure was to differentiate the function of GDE1 from CLSY3/4 and narrow down the interesting siren loci rather than to claim exhaustive identification of transcription factor binding.

To address this concern and improve clarity, we have rephrased the relevant text as follows: "These transcription factors were enriched at more than 77% of the siren loci (Fig. 4a and Extended Data Fig. 4b) and the strongly bound subset of Group 1 primarily corresponds to the siren loci (Fig. 4b)." Results have been added to the revised manuscript in Fig. 4b.

Fig. 4b Metaplot showing enrichment of VDD, VAL, and REM13 ChIP-seq signal over siren and Group1 non-siren region.

The other point addressing annotation of the different groups, is the localization in euchromatin (Group 1) and heterochromatin (Group 2) as mentioned above, which has also been added to the manuscript as Fig. 2f.

Minor fixes to make

1. There is so much critical data in the Extended Data section that is necessary to understand and interpret this manuscript. If the data is a critical step for the rest of the paper, that needs to be pulled into the main figures. For example, Ex. Data 5D.

Thank you for your suggestion. Figures have been reformatted for improved clarity and to align with the journal's formatting requirements. We now present Extended Fig 5D in Fig. 4j.

2. I was confused by line 211. The occurrence of 3 repeats is almost the same level as the occurrence of 2 repeats, and it likely a statistically over-represented compared to the expected rate. But this is said to be "rare".

We apologize for this sloppy language, and you are right. We have rephrased this as "Throughout the genome, the occurrence of the three repeats (n=38) and two repeats (n=48) is rare, with 37,456 loci having a single repeat. "

3. In Figures such as 4D, please write out the full genotype rather than writing "triple" mutant.

Thank you for pointing this out and we revised as suggested.

4. I did not see data on whether the epitope-tagged versions of the proteins used for CHIP complemented the mutants? Was this assumed? Please show this data as Extended Data.

Thank you for pointing this out. We analyzed siRNA and CHH/CHG methylation levels at siren loci in *GDE1-3FLAG* complementation lines. As shown in Fig 4c-d, the introduction of *GDE1-3FLAG* successfully complement the siRNA and DNA methylation phenotypes of the *gde1-1* mutant.

The complementation of *CLSY3* and *CLSY4* transgenic lines has been previously reported in Zhou et al., 2022 (Supplementary Fig. 12b of their paper). These transgenes were cloned using the same destination construct as described in that study. Law et al., 2011 (*PLoS Genetics*) demonstrated that Pol IV-3FLAG complements the DNA methylation defects in the *nRPD1-4* mutant (see Fig. 1A of their paper).

Most single REM transcription factor mutants (e.g., *rem8*) do not exhibit an siRNA phenotype, likely due to genetic redundancy, so we had nothing to complement. siRNA phenotypes are observed when *REM46*, *VAL*, and *REM12* are simultaneously deleted. Complementation experiments with all three genes are complex due to this redundancy and the experimental scale required. The *vdd* knockout mutant is lethal, although this lethality may not be related to siRNA production. We introduced our *VDD-9myc* construct into *vdd1-/+* heterozygous mutants, and it successfully rescued the lethality phenotype. This data has been included in Extended Data Fig. 5a.

These results collectively demonstrate that most of epitope-tagged constructs used in our study are functional and capable of complementing the respective mutants.

Thank you again for all of these detailed comments!

We thank all reviewers for their constructive and helpful suggestions and comments. Our response in blue font can be found beneath each original reviewer's comment below.

Reviewer #1:

Remarks to the Author:

Most of my concerns have been well addressed, and the manuscript has been substantially improved. One remaining concern is the description of Figure 6d. The text indicated that "To explore this, we divided the motifs into two additional groups based on the relative positions of Pol IV summits, and observed that the direction of the Pol IV shift closely aligned with more accessible regions." In figure 6d, however, the 24-nt siRNA level and the MNase-seq level showed a positive correlation, suggesting that Pol IV shift closely aligned with more nucleosome-occupied regions, because the MNase-seq signals represent the nucleosome-occupied regions but not the accessible regions. This should be corrected in the revised manuscript.

Thank you for catching this! We have revised the description to reflect the positive correlation between the 24-nt siRNA level and MNase-seq level indicates that the Pol IV shift aligns more closely with nucleosome-occupied regions, rather than accessible regions. We are not exactly sure why this is the case. Perhaps Pol IV transcription is inducing silencing and higher nucleosome occupancy. We have added to the text that we are not sure if the higher nucleosome occupancy is a cause or a consequence of Pol IV transcription.

*Additional Comments in response to Reviewer #2's remaining points:

I have read referee#2's comments and the authors' responses and revisions in the manuscript. I found that the comments have been well addressed in the revised manuscript. Although the CLSY3/4 and Pol IV were not co-purified with GDE1, they were shown to interact with GDE1 determined by co-IP. Moreover, group 1 and 2 siRNAs were characterized by additional experiments and analyses (Extended Data Fig. 6 and Fig. 2f). I feel that the manuscript has been significantly improved, and agree to publication.

Thank you for these positive comments.

Reviewer #2:

None

Reviewer #3:

Remarks to the Author:

I commend the authors for doing a very professional and complete revision to this manuscript. Overall, this is a strong paper that pushes the field forward into more mammalian-like sequence-based targeting of repressive chromatin marks. One key difference is that this system based on REM TFs is still siRNA based, where in mammals the KRAB proteins directly target DNA methylation without small RNA production.

My remaining comment simply has to do with CHIP performed using tagged versions of proteins from which the tagged transgenic version has not been shown to complement the corresponding mutant. The authors do this for speed and ease of their experiments, and there is a lot of high quality supporting data shown in this paper. However, that was an industry-standard that this paper erodes, so therefore the lead author of this manuscript should not comment on other people's work that lacks this previously-necessary control.

Thank you for your thoughtful and positive feedback on our revised manuscript. We appreciate your recognition of the work's contribution to the field. Regarding your comment on the use of tagged proteins for CHIP analysis, we understand the importance of complementing transgenic versions with corresponding mutants as an industry standard. We will be careful in our discussions of related work and ensure that our conclusions are clearly supported by our data. Your perspective is valuable, and we will take it into account in our future work and presentations.